# Compact Proofs of Model Performance via Mechanistic Interpretability

**Jason Gross**[*]     **Rajashree Agrawal**     **Thomas Kwa**[†]     **Euan Ong**[†]     **Chun Hei Yip**[†]

**Alex Gibson**[‡]                 **Soufiane Noubir**[‡]                 **Lawrence Chan**

## Abstract

We propose using mechanistic interpretability – techniques for reverse engineering model weights into human-interpretable algorithms – to derive and compactly prove formal guarantees on model performance. We prototype this approach by formally proving accuracy lower bounds for a small transformer trained on Max-of-$K$, validating proof transferability across 151 random seeds and four values of $K$. We create 102 different computer-assisted proof strategies and assess their length and tightness of bound on each of our models. Using quantitative metrics, we find that shorter proofs seem to require and provide more mechanistic understanding. Moreover, we find that more faithful mechanistic understanding leads to tighter performance bounds. We confirm these connections by qualitatively examining a subset of our proofs. Finally, we identify compounding structureless errors as a key challenge for using mechanistic interpretability to generate compact proofs on model performance.

## 1 Introduction

One approach to ensuring the safety and reliability of powerful AI systems is via formally verified proofs of model performance [48, 11]. If we hope to deploy formal verification on increasingly large models [24, 27] with powerful emergent capabilities [56] across more diverse and broader domains [5, 46], we will need *compact* proofs of generalization bounds on *specific* models that certify *global* robustness. However, existing approaches tend to use proof strategies that suffer from bad asymptotic complexity, while verifying either generalization properties of training procedures or local robustness properties of specific models.

One key challenge to verification is that neural network architectures are highly expressive [51, 58], and models with similar training procedure and performance may still have learned significantly different weights [38, 9]. This expressivity makes it difficult to adequately *compress* explanations of global model behavior in ways that *correspond* closely enough to the model's actual mechanisms to be useful for efficient verification without being too *lossy*, especially when using only knowledge of the architecture or training procedure. We propose verifying model performance using understanding derived from *mechanistic interpretability* (Section 2) – that is, reverse engineering the specific implementation of the algorithm from the learned weights of particular models. Knowledge of the specific implementation allows us to construct less lossy simplifications of the model, and more efficiently reason about model performance over possible inputs.

In this work, we provide a case study of translating mechanistic interpretations into compact proofs. We train an attention-only transformer on a Max-of-$K$ task with 151 random seeds (Section 3), and

---

[*]Corresponding author. Please direct correspondence to `jgross@mit.edu`.
[†]These authors contributed equally to this work.
[‡]These authors contributed equally to this work.

38th Conference on Neural Information Processing Systems (NeurIPS 2024).

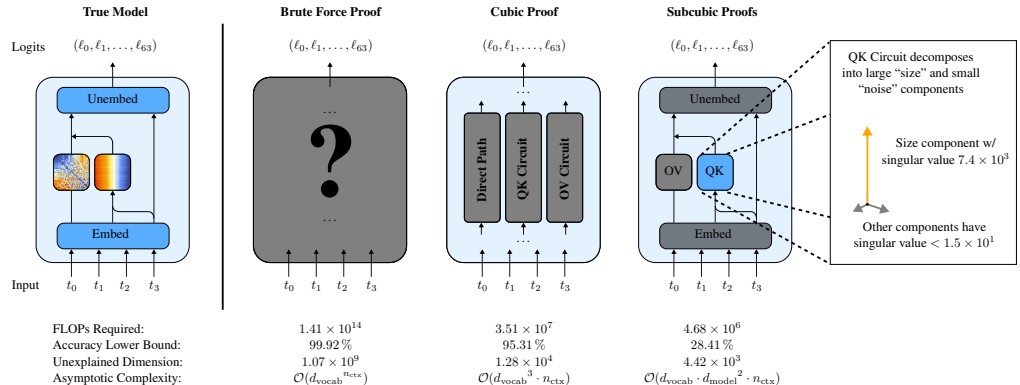

**Figure 1:** We construct proofs using different degrees of mechanistic interpretation. (Left) The models we consider in this paper are one-layer attention-only transformers, and so contain three "paths": the OV circuit, the QK circuit, and the direct path. (Right) For the brute-force proof (Section 4.3.1), we treat the model as a black box and thus need to check all possible combinations of inputs. For the cubic proof (Section 4.3.1), we decompose the model into its three corresponding paths, but still check the correctness of each path via brute force. Finally, in some subcubic proofs (Section 4.3), we use all parts of the mechanistic interpretation presented in Section 3. (Bottom) For each of the three categories of proof, we report the number of FLOPs used in computing the certificate (lower=better, Appendix A.6), lower bound on model accuracy (higher=better), effective dimension of the unexplained parts of the model (lower=better, Appendix A.5), and asymptotic complexity of the proof strategy as we scale the inputs and model (lower=better). Significantly more compact proofs have vacuous accuracy bounds by default. Using more mechanistic understanding allows us to recover some, but not all, of the accuracy bounds on these more compact proofs, as our understanding is not fully faithful to the model internals.

then reverse engineer the models using standard mechanistic interpretability techniques. We use our understanding to define a set of 102 different computer-assisted proof strategies with varying tightness of bound and with different asymptotic complexity and number of required FLOPs (Section 4).[4] We validate our technique against an additional 604 models for varying values of $K$ (Appendix A.2.1).

We define a quantitative metric to assess the mechanistic understanding used in a proof strategy by the dimensionality of the function space that the proof strategy must consider, which we deem the *unexplained dimensionality* of the proof strategy (Sections 5.1, and A.5). Using this metric, we find a negative relationship between proof length and degree of understanding. We qualitatively examine proof strategies to confirm and explain this relationship, finding that more compact proofs both require and provide more mechanistic understanding. We also find suggestive evidence that the trade-off between proof length and tightness of bound is modulated by the faithfulness of the mechanistic understanding used to derive the proof (Section 5.2).[5]

However, we also identify compounding structureless error terms as a key challenge for generating compact proofs on model behavior (Sections 5.3, and G.2.5). The implementation of algorithms inside of neural networks may contain components that defy mechanistic understanding and appear to us as "noise". When we don't know how noise composes across model components, establishing a bound requires pessimizing over the ways the composition could occur. Worst-case noise can quickly grow even when the empirical noise is small, leading to vacuous performance bounds.

## 2 Mechanistic interpretability for proofs

**Generalization bounds on global performance** In the style of prior mechanistic interpretability evaluation work [6], we target theorem templates that establish bounds on the expected global performance of the model. Let $\mathcal{M} : X \to Y$ be a model (here assumed to be a neural network), $\mathcal{D}$ be a probability distribution over input-label pairs $(l, \mathbf{t}) \in L \times X$, notated as $\mathcal{D}|_X$ when marginalized over labels, and $f : L \times Y \to \mathbb{R}$ be a scoring function for evaluating the performance of the model. Then, we seek to establish lower bounds $b$ on the expected $\bar{s}$ as the form:

$$\bar{s} := \mathbb{E}_{(l,\mathbf{t}) \sim \mathcal{D}} \left[ f(l, \mathcal{M}(\mathbf{t})) \right] \geq b. \tag{1}$$

---

[4]Our 102 proof strategies are can be broken up as $1 + 1 + 10 \times 5 \times 2$: two standalone strategies, and a class of strategies parameterized on three axes of cardinality 10, 5, and 2 (Appendix H).

[5]Code: https://github.com/JasonGross/guarantees-based-mechanistic-interpretability/

As $f$ can be any metric, this is a fully general template for theorems that can capture any aspect of model performance for which we have a formal specification. However, in this work we restrict $f$ to be the accuracy and $\mathcal{D}|_X$ to be uniform, so our theorems lower bound the accuracy of the model. Our proof methodology generalizes straightforwardly to other input distributions (Appendix A.8), and only a little work is required to generalize from accuracy to log-loss (Appendix A.11).

**Proof template** The proofs of model performance in this work have two components: a computational component $C$ : model weights $\rightarrow \mathbb{R}$ and a non-computational component $Q$ arguing that for any model $\mathcal{M}'$, $C(\mathcal{M}') \leq \mathbb{E}_{(l,\mathbf{t}) \sim \mathcal{D}} f(l, \mathcal{M}'(\mathbf{t}))$, thus implying that $C$ generates a valid lower bound for the performance of $\mathcal{M}$. The whole proof is $Q$ paired with a trace of running $C$ that certifies its output on $\mathcal{M}$.[6] Here, $b = C(\mathcal{M})$. As even the size of the model parameters is much larger than any reasonable $Q$, we approximate the length of a proof pair $C, Q$ by the length of a trace of $C(\mathcal{M})$.

**Proof compactness vs. tightness of bound** Different proof strategies make different tradeoffs between compactness and tightness of bound. For example, consider two extreme proof strategies: We can "prove" a vacuous bound using a null proof. On the other hand, in the brute-force proof, we simply run the model on the entirety of $\mathcal{D}$ to achieve $b = \bar{s}$, albeit with a very long proof.

We quantify the length of $C(\mathcal{M})$ using two metrics: the *asymptotic time complexity* of $C$ as we scale the size of the model and the input $\mathbf{t}$, as well as the empirical average *number of floating point operations* required to evaluate $C(\mathcal{M}')$ over a given set of models $\{\mathcal{M}_i\}$. We measure *tightness of bound* of $C(\mathcal{M})$ using the ratio of the bound to the true accuracy: $b/\bar{s}$.

**Proof as pessimal ablation** A standard way of assessing the faithfulness of mechanistic interpretability is by ablating the parts of the model that your interpretation does not explain [54, 6, 23]. In this framework, proofs can be thought of as performing a *pessimal ablation* over the unexplained parts of the model – we set the remaining components of the model (the "noise" or error terms) to values over $X$ that minimize the performance of the model. However, the number of ablations required for a complete argument might be quite high. Thus, we construct *relaxations* (Appendix A.4) over input sequences, such that performing pessimal ablations on a smaller number of relaxed input sequences is sufficient to lower bound the performance on $\mathcal{D}$.

## 3   Experimental setting

We study our approach to generating compact proofs in a simple toy setting: Max-of-$K$.

**Model Architecture** We study one-layer, one-head, attention-only transformers with no biases but with learned positional embeddings, with vocabulary size $d_{\text{vocab}}$, model and head dimension $d = d_{\text{model}} = d_{\text{head}}$, and context length $n_{\text{ctx}} := k$. The model parameters consist of the $n_{\text{ctx}} \times d_{\text{model}}$ positional embedding $P$; the $d_{\text{vocab}} \times d_{\text{model}}$ token embed $E$; the $d_{\text{model}} \times d_{\text{model}}$ query, key, value, and output matrices of the attention head $Q$, $K$, $V$, and $O$; as well as the $d_{\text{model}} \times d_{\text{vocab}}$ unembed matrix $U$. We assume (as is standard in language modeling) that $d_{\text{model}} < d_{\text{vocab}}$.

For an $n_{\text{ctx}} \times d_{\text{vocab}}$ one-hot encoding $\mathbf{x} = [x_0, x_1, \ldots, x_{n_{\text{ctx}}-1}]$ of an input sequence $\mathbf{t} = [t_0, t_1, \ldots, t_{n_{\text{ctx}}-1}]$, we compute the logits of the model as follows:

$$h^{(0)} = \mathbf{x}E + P \qquad\qquad \text{Initial residual stream } (n_{\text{ctx}} \times d_{\text{model}})$$

$$\alpha = h^{(0)}QK^T h^{(0)T}/\sqrt{d} \qquad\qquad \text{Attention matrix } (n_{\text{ctx}} \times n_{\text{ctx}})$$

$$h^{(1)} = \sigma^*(\alpha) \cdot h^{(0)}VO + h^{(0)} \qquad\qquad \text{Final residual stream } (n_{\text{ctx}} \times d_{\text{model}})$$

$$\mathcal{M}(\mathbf{t}) = \ell = h^{(1)}_{n_{\text{ctx}}-1}U \qquad\qquad \text{Final seq. position logits } (d_{\text{vocab}})$$

where $\sigma^*$ is the masked softmax function used in causal attention. Because we only look at outputs of the model above the final sequence position $i = n_{\text{ctx}} - 1$, we also denote this position as the "query position" and the value of the token in this position as $t_{\text{query}}$, one-hot encoded as $x_{\text{query}}$. The model's prediction is the token corresponding to the max-valued logit $\ell_{\text{max}}$.

**Task** Specifically, we study the setting with $n_{\text{ctx}} = k = 4$ because it is the largest sequence length for which we can feasibly evaluate the brute-force proof. We set hidden dimension $d_{\text{model}} = 32$

---

[6]Other components of the proof to account for the difference between floating point numbers and reals are described in Appendix A.7. Note that all proofs explicitly given in this paper are of $Q$ only; we do not include any traces of running $C$.

and a vocabulary of size $d_{\text{vocab}} = 64$ comprising integers between 0 and 63 inclusive. For an input sequence $\mathbf{t}$, we denote the *true* maximum of the sequence by $t_{\max}$. Outputting the correct behavior is equivalent to outputting logits $\ell$ such that $\Delta\ell_{t^*} := \ell_{t^*} - \ell_{\max} < 0$ for all $t^* \neq t_{\max}$. We trained 151 models on this task. Models achieved average accuracy $0.9992 \pm 0.0015$ over the entire distribution.

**Path decomposition** Following prior work [13], we expand the logits of the model and split the paths through the model into three components – the QK circuit, the OV circuit, and the direct path:

$$
\mathcal{M}(\mathbf{t}) = \sigma^*\Big( \underbrace{\left(x_{\text{query}}E + P_{\text{query}}\right) QK^T \left(\mathbf{x}E + P\right)^T / \sqrt{d}}_{\text{QK circuit}} \Big) \cdot \underbrace{\left(\mathbf{x}E + P\right) VOU}_{\text{OV circuit}} + \underbrace{\left(x_{\text{query}}E + P_{\text{query}}\right) U}_{\text{direct path}}
$$

(2)

Intuitively, the QK circuit determines *which* tokens the model attends to from a particular query token and sequence position, while the OV circuit *processes* the tokens and sequence positions the model attends to. The direct path is simply the skip connection around the attention head.

We further divide the QK and OV circuits into token (position-independent) and position-dependent components. Let $P_{\text{avg}} = \sum_i P_i / n_{\text{ctx}}$ be the average position embeds across positions (of size $d_{\text{model}}$), and let $\bar{\mathbf{P}}$ denote either $\mathbf{1}_{n_{\text{ctx}}} \otimes P_{\text{avg}}$ or $\mathbf{1}_{d_{\text{vocab}}} \otimes P_{\text{avg}}$ depending on context, the result of broadcasting $P_{\text{avg}}$ back into the shape of $P$ or $E$ (that is, $n_{\text{ctx}} \times d_{\text{model}}$ or $d_{\text{vocab}} \times d_{\text{model}}$). Similarly, let $\mathbf{P}_q = \mathbf{1}_{d_{\text{vocab}}} \otimes P_{\text{query}}$ be the result of broadcasting $P_{\text{query}}$. Then for one-hot encoded $\mathbf{x}$, we can rewrite the QK and OV circuits, as well as the direct path, as follows:

$$
\text{QK circuit} = x_{\text{query}}\Big( \underbrace{\mathbf{E}_q QK^T \bar{\mathbf{E}}^T}_{\text{EQKE}} \mathbf{x}^T + \underbrace{\mathbf{E}_q QK^T \hat{\mathbf{P}}^T}_{\text{EQKP}} \Big)
$$

$$
\text{OV circuit} = \mathbf{x} \underbrace{\bar{\mathbf{E}}VOU}_{\text{EVOU}} + \underbrace{\hat{\mathbf{P}}VOU}_{\text{PVOU}} \qquad \text{Direct Path} = x_{\text{query}} \underbrace{\mathbf{E}_q U}_{\text{EU}}
$$

where $\hat{\mathbf{P}} = P - \bar{\mathbf{P}}$ and $\bar{\mathbf{E}} = E + \bar{\mathbf{P}}$ and $\mathbf{E}_q = E + \mathbf{P}_q$ (since $h^{(0)} = \mathbf{x}\bar{\mathbf{E}} + \hat{\mathbf{P}}$).

## 3.1 Mechanistic interpretation of learned models

Using standard empirical mechanistic interpretability techniques, we interpret one of our learned models (our "mainline" model) by independently examining the QK and OV circuits and the direct path.[7] We find that the model outputs the largest logit on the true max token $t_{\max}$ by attending more to larger tokens via the QK circuit and copying the tokens it attends to via the OV circuit. We then quantitatively confirm that these interpretations hold for all 151 models by reporting the mean plus minus standard deviation for various summary statistics. Plots for this section are available in Appendix B.2.

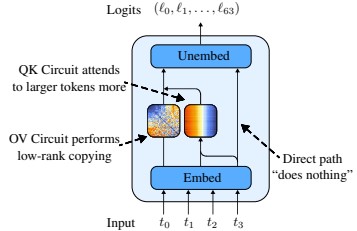

**Figure 2:** The models in our setting implement Max-of-$K$ by attending exponentially more to larger tokens and copying the attended-to tokens (Section 3.1).

**QK circuit** By qualitatively examining the position-independent QK component EQKE, we find the amount of pre-softmax attention paid to a key token is approximately independent of the value of the query token $t_{\text{query}}$, and increases monotonically based on the size of the key token. We confirm this hypothesis by performing a singular-value decomposition (SVD) of the EQKE matrices (Appendix G.2.3), and find that it contains a single large rank-one component with singular value around $7800 \pm 380$, around $620 \pm 130$ times larger than the second largest component with singular value $13 \pm 3$. The left (query-side) singular vector is approximately constant in all dimensions, with value $0.1243 \pm 0.0003 \approx \frac{1}{8} = \frac{1}{\sqrt{d_{\text{vocab}}}}$. The right (key-side) singular vector of this component is monotonically increasing as we increase the size of the key token, with ($1/\sqrt{d}$-scaled) pre-softmax attention increasing by an average of $1.236 \pm 0.056$ when the key token increases by 1.[8]

In comparison, each $1/\sqrt{d}$-scaled entry of the position-dependent QK component EQKP has negligible size (average $0.31 \pm 0.18$), suggesting that EQKP is unimportant to the functioning of the model.

---

[7] All of our trained models behave similarly; see Appendix B.3.

[8] This implies that the ratio of attention paid to token $t$ and $t-1$ is approximately $\exp(1.236) \approx 3.442$.

We confirm this by zero ablating EQKP, which changes the models' accuracies from $0.9992 \pm 0.0015$ to $0.9993 \pm 0.0011$. Combined with our interpretation of EQKE, this implies that the attention pattern of the model depends only on the token values and not the ordering of the sequence.

**OV circuit** Then, by qualitatively examining the position-independent OV component EVOU, we see that it has large positive entries along the diagonal. In fact, the entry along the diagonal is the largest in the row for all rows corresponding to $t > 6.6 \pm 1.2$. Since each entry in the sequence is uniformly sampled and $d_{\text{vocab}} = 64$, this means that EVOU is a good approximation for the identity matrix for all but $\approx (7/64)^4 \approx 1.2 \times 10^{-2} \%$ of the sequences.

As with the position-dependent QK component, the position-dependent OV component PVOU also has negligible size and is unimportant to model performance. Taken together with the above results on EVOU, this suggests that the attention head copies the tokens it attends to.

**Direct path** As with the two position-dependent components, the entries in EU have small absolute magnitude $2.54 \pm 0.20$,[9] and contribute negligibly to model performance.

# 4 Proofs of model performance

In this section we describe intuitions for three categories of proof that are developed around different mechanistic interpretations and methods for using the interpretations. The strategies result in proofs of different complexities with varying bound tightness (Table 1). We provide detailed theorem statements, proofs, algorithms, and explanations of proof search in Appendices C, D, E, F, and G.

Our theorem statements for $Q$ will all be of the form

$$\forall \mathcal{M}', C_{\textit{specific strategy}}(\mathcal{M}') \leq \mathbb{E}_{\mathbf{t} \sim \mathcal{D}|_X} f(t_{\max}, \mathcal{M}'(\mathbf{t})).$$

We leave implicit the traces of running $C_{\textit{specific strategy}}$ on our specific models to give the overall theorem. We report the computational complexity or estimated FLOPs of running $C_{\textit{specific strategy}}$ as approximations for our proof lengths.

## 4.1 The brute-force baseline

We start by considering the brute-force proof (Appendix D), which treats the model as a black box and evaluates it on all possible sequences.[10] However, this proof strategy has bad asymptotic complexity and is untenable for larger models and larger input distributions. So in subsequent sections, we use knowledge of the model drawn from the interpretation in Section 3.1 to derive more compact proofs.

## 4.2 A cubic proof

Next, we use the fact that the model is composed of the direct path and the QK and OV circuits (Section 3) to decrease the number of sequences that we need to consider, and the fact that only the position-independent components EQKE and EVOU contribute meaningfully to performance (Section 3.1) to pessimize over sequence ordering.

First, let a pure sequence $\xi$ be a sequence with at most three distinct tokens: the max token $t_{\max}$, the final token $t_{\text{query}} \leq t_{\max}$, and optionally a third token $t' < t_{\max}$, and let $\Xi^{\text{pure}}$ be the set of all pure sequences in $X$.[11] For a given input sequence $\mathbf{t}$, define the adjacent pure sequences $\text{Adj}(\mathbf{t})$ as the set of sequences that share the same max and query token, and only take on values in $\mathbf{t}$:

$$\text{Adj}(\mathbf{t}) = \left\{ \xi \in \Xi^{\text{pure}} \,\middle|\, \max_i \xi_i = t_{\max}, \, \xi_{\text{query}} = t_{\text{query}}, \forall i < n_{\text{ctx}}, \, \xi_i \in \mathbf{t} \right\}$$

Using the convexity of softmax and the fact that the model contains three paths, we can show that one-layer attention-only transformers satisfies a variant of the following convexity property: for a given $\mathbf{t}$, if $\mathcal{M}(\xi)$ is correct for all $\xi \in \text{Adj}(\mathbf{t})$, then $\mathcal{M}(\mathbf{t})$ is correct. That is, for these transformers, we can bound the accuracy on all sequences by evaluating $\mathcal{M}$ on only the $O(d_{\text{vocab}}^3(n_{\text{ctx}} - 1)!)$

---

[9]For comparison, the average off-diagonal element of EVOU is $21.68 \pm 0.83$ below the corresponding diagonal element.

[10]Appendix A.10 discusses how to compute the "brute-force" accuracy of a model on an infinite distribution.

[11]In Section 4.3, we will consider a smaller set of "pure sequences".

**Table 1:** We report the proof complexity, accuracy bound, and estimated flops required (Equation 2), as well as unexplained dimensionality (Section 5). We round the FLOP and unexplained dimension counts to the closest power of 2, and report the mean/standard deviation of the bound averaged across all 151 models. As we include more aspects of the mechanistic interpretation (reflected by a lower number of unexplained dimensions), we get more compact proofs (in terms of both asymptotic complexity and FLOPs), albeit with worse bounds. For space reasons, we use $k := n_{\text{ctx}}$, $d := d_{\text{model}}$, and $v := d_{\text{vocab}}$.

| Description of Proof | Complexity Cost | Bound | Est. FLOPs | Unexplained Dimensions |
|---|---|---|---|---|
| Brute force | $\mathcal{O}(v^{k+1}kd)$ | $0.9992 \pm 0.0015$ | $2^{47}$ | $2^{30}$ |
| Cubic | $\mathcal{O}(v^3k^2)$ | $0.9531 \pm 0.0087$ | $2^{25}$ | $2^{14}$ |
| Sub-cubic | $\mathcal{O}(v^2 \cdot k^2 + v^2 \cdot d)$ | $0.702 \pm 0.033$ | $2^{21}$ | $2^{13}$ |
| w/o mean+diff | | $0.349 \pm 0.080$ | $2^{21}$ | $2^{13}$ |
| Low-rank QK | $\mathcal{O}(v^2k^2 + \underbrace{vd^2}_{\text{QK}} + \underbrace{v^2d}_{\text{EU\&OV}})$ | $0.675 \pm 0.035$ | $2^{22}$ | $2^{12}$ |
| SVD only | | $0.284 \pm 0.072$ | $2^{22}$ | $2^{12}$ |
| Low-rank EU | $\mathcal{O}(v^2k^2 + \underbrace{vd}_{\text{EU}} + \underbrace{v^2d}_{\text{QK\&OV}})$ | $0.633 \pm 0.062$ | $2^{21}$ | $2^{13}$ |
| SVD only | | $(3.38\pm0.06)\times10^{-6}$ | $2^{21}$ | $2^{13}$ |
| Low-rank QK\&EU | $\mathcal{O}(v^2k^2 + \underbrace{vd^2}_{\text{QK}} + \underbrace{vd}_{\text{EU}} + \underbrace{v^2d}_{\text{OV}})$ | $0.610 \pm 0.060$ | $2^{21}$ | $2^{13}$ |
| SVD only | | $(3.38\pm0.06)\times10^{-6}$ | $2^{22}$ | $2^{13}$ |
| Quadratic QK | $\mathcal{O}(v^2k^2 + \underbrace{vd}_{\text{QK}} + \underbrace{v^2d}_{\text{EU\&OV}})$ | $0.316 \pm 0.037$ | $2^{21}$ | $2^{12}$ |
| Quadratic QK\&EU | $\mathcal{O}(v^2k^2 + \underbrace{vd}_{\text{QK\&EU}} + \underbrace{v^2d}_{\text{OV}})$ | $0.283 \pm 0.036$ | $2^{21}$ | $2^{13}$ |

pure sequences. This allows us to bound the accuracy of our actual $\mathcal{M}$ on all $d_{\text{vocab}}{}^{n_{\text{ctx}}}$ sequences, while evaluating it on $O(d_{\text{vocab}}{}^3(n_{\text{ctx}} - 1)!)$ sequences.

We can reduce the number of sequences that we need to evaluate by pessimizing over the order of a sequence. For a given tuple of $(t_{\max}, t_{\text{query}}, t')$, there are $(n_{\text{ctx}} - 1)!$ pure sequences, corresponding to the permutations of the tuple. Pessimizing over the order of sequences reduces the number of sequences to consider for each $(t_{\max}, t_{\text{query}}, t')$ tuple to the number of $t'$ in the pure sequence, and the total number of sequences to $O(d_{\text{vocab}}{}^3 n_{\text{ctx}})$. By precomputing the five component matrices EU, EQKE, EQKP, EVOU, PVOU and cleverly caching intermediate outputs, we can reduce the additional work of each sequence to the $O(n_{\text{ctx}})$ required to compute the softmax over $n_{\text{ctx}}$ elements, resulting in asymptotic complexity $O(d_{\text{vocab}}{}^3 n_{\text{ctx}}{}^2)$ (Theorem 12, additional details in Appendix E).

## 4.3 Sub-cubic proofs

We now consider proofs that are more compact than $O(d_{\text{vocab}}{}^3)$. These require avoiding iteration over any set of size $O(d_{\text{vocab}}{}^3)$ (e.g. the set of pure sequences) and performing operations that take $O(d_{\text{vocab}})$ time on each of $O(d_{\text{vocab}}{}^2)$ combinations. Unfortunately, some methods of avoiding these operations can lead to vacuous bounds (i.e. accuracy lower bounds near 0%). In order to recover non-vacuous bounds, we introduce two tricks: the "mean+diff trick" to better approximate the sum of two components with unequal variance, and the "max row diff trick" to improve upon the low-rank approximations for EU and EQKE. We consider applying variants of these tricks at different locations in the naïve subcubic proof, leading to 100 distinct subcubic proof strategies. See Appendix G.2 for a formal description of these strategies.

### 4.3.1 Removing cubic-time computations

**Reducing the number of cases by pessimizing over sufficiently small tokens** Previously, we considered $\Theta(d_{\text{vocab}}{}^3 n_{\text{ctx}})$ pure sequences $\xi$, with $\xi$ parameterized by $(t_{\max}, t_{\text{query}}, t', c)$. Recall from our mechanistic interpretation in Section 3.1 that the pre-softmax attention paid from $t_{\text{query}}$ to a key token $t'$ is broadly invariant in $t_{\text{query}}$ and increases roughly linearly with the size of $t'$. This allows us to pessimize over the OV circuit over all "sufficiently small" tokens.

More formally, suppose we are given some gap $g \in \mathbb{N}$. For each pure sequence $\xi$ with max token $t_{\max}$, query token $t_{\text{query}}$, such that $t_{\text{query}} \leq t_{\max} - g$, and $c$ copies of the third token type $t' \leq t_{\max} - g$, we pessimally ablate the OV circuit over the set $\Xi^{\text{pure}}(t_{\max}, t_{\text{query}}, c; g)$ of pure sequences $\xi'$ with the same max and query tokens and $c$ copies of the third token type $t'$. If the model gets all sequences in $\Xi^{\text{pure}}(t_{\max}, t_{\text{query}}, c; g)$ correct, then we can conclude that it gets $\xi$ correct, otherwise, we treat the model as having gotten $\xi$ wrong. This means that it suffices to only consider the $O(d_{\text{vocab}}{}^2 n_{\text{ctx}})$ pessimal pure sequences of each of the $O(d_{\text{vocab}}{}^2 n_{\text{ctx}})$ sets of the form $\Xi^{\text{pure}}(t_{\max}, t_{\text{query}}, c; g)$.

**Decoupling and pessimizing computations that require $O(d_{\text{vocab}}{}^3)$ computations** Many parts of our cubic certificate require iterating through $O(d_{\text{vocab}}{}^2)$ cases parameterized by $(t_{\max}, t_{\text{query}})$ or $(t_{\max}, t')$. For example, as part of the pessimization procedure over pure sequences, for each of the $d_{\text{vocab}}$ possible values of $t_{\max}$, we need to consider the relative effects on the $d_{\text{vocab}}$-sized logits of attending to each of the $O(d_{\text{vocab}})$ other tokens $t' < t_{\max}$, and for each $t_{\max}$ and $t_{\text{query}}$, we need to check that the contribution of the direct path on logits $x_{\text{query}}$EU is not sufficiently large as to overwhelm the contribution from $x_{\max}$EVOU. We independently pessimize over each of these components over one of the $d_{\text{vocab}}$-sized axes: for example, instead of computing $x_{\max}$EVOU $+ x_{\text{query}}$EU for each $t_{\max}$, $t_{\text{query}}$ pair, we first pessimally ablate the direct path along the query token (which takes $O(d_{\text{vocab}}{}^2)$ time as it does not depend on the $t_{\max}$, and then consider the sum $x_{\max}$EVOU $+ \max_{x'} x'$EU. Since this sum no longer depends on $t_{\text{query}}$, we only need to perform it $O(d_{\text{vocab}})$ times, for a total cost of $O(d_{\text{vocab}}{}^2)$.

**Low rank approximations to** EQKE **and** EU Recall from Section 3.1 that EQKE is approximately rank 1, where the sole direction of variation is the size of the key token. By computing only the low rank approximation to EQKE, we can more cheaply compute the most significant component of the behavior in the QK circuit. To bound the remaining error, we can use the fact that after pulling off the first principal component from each of the four matrices we multiply, very little structure remains.

We can find the rank 1/2 approximations by performing SVD on EQKE. We can efficiently compute the SVD in $\mathcal{O}(d_{\text{vocab}} d_{\text{model}}{}^2)$ time by using the fact that EQKE can be written as the product of a $d_{\text{vocab}} \times d_{\text{model}}$ matrix and a $d_{\text{model}} \times d_{\text{vocab}}$ matrix. This allows us to avoid performing the $\mathcal{O}(d_{\text{vocab}}{}^2 d_{\text{model}})$-cost matrix multiplications to explicitly compute EQKE.

Similarly, we can more efficiently check that the direct path EU contributes negligibly to the model outputs, by using SVD to decompose EU into a sum of rank 1 products (which we can evaluate exactly) and a high-rank error term that we can cheaply bound.

### 4.3.2 Additional subcubic proof strategies

**Tighter bounds for sums of variables with unequal variance via the "mean+diff trick"** Suppose we want to lower bound the minimum of the sum of two functions over three variables $h(x, y, z) = f(x, y) + g(y, z)$, while only iterating over two variables at a time. The naïve way is to minimize $f(x, y)$ and $g(x, y)$ independently:

$$\min_{x,y,z} h(x, y, z) \geq \min_{x,y} f(x, y) + \min_{y,z} g(y, z)$$

Here, the error comes from setting the $y$s in $f$ and $g$ to different values. But in cases where $g(y, z)$ varies significantly with $y$ and only slightly with $z$, rewriting $g$ as a sum of a component that is independent of $z$ (only varying along $y$), and one that depends on $z$, yields a better lower bound:

$$\min_{x,y,z} h(x, y, z) \geq \min_{x,y} \left( f(x, y) + \mathbb{E}'_z g(y, z') \right) + \min_{y,z} (g(y, z) - \mathbb{E}'_z g(y, z'))$$

This estimate will have error at most $\varepsilon$, while the naïve estimator can have arbitrarily large error. We refer to this rewrite as the "mean+diff trick".[12] From the mechanistic interpretation in Section 3.1, we know that some of the components barely vary among one or more axes. So we can apply the mean+diff trick to get tighter lower bounds.

**Avoiding matrix multiplications using the "max row-diff trick"** Using properties of linear algebra, we derive a cheap approximation to the max row-diff for the product of matrices $AB$ in terms of the product of the max row-diff of $B$ and the absolute value of $A$, which we deem the "max row-diff"

---

[12]In fact, this is the motivation behind the standard rewrites of QK and OV into position-independent and position-dependent components (Section 3).

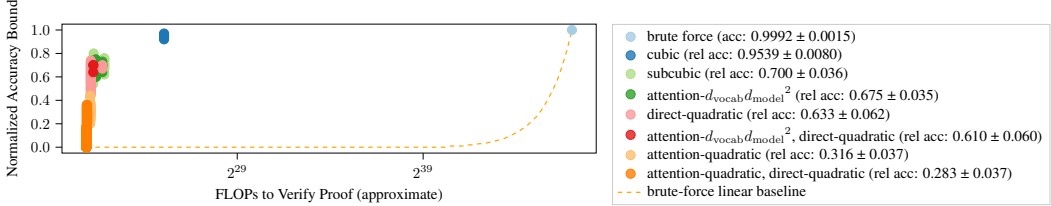

**Figure 3:** For each of the proofs in Section 4, we plot the number of FLOPs used to compute the certificate, as well as the normalized accuracy lower-bound ($b/\bar{s}$). The brute-force proof (Section 4.1) computes the exact performance uses orders of magnitude more compute than other approaches. The cubic proof (Section 4.3) uses a small amount of mechanistic understanding and less compute, while still retaining good accuracy lower bounds. Finally, subcubic proofs (Section 4.3) require the entirety of the mechanistic interpretation of the model to attain non-vacuous bounds; this understanding allows us to further reduces compute costs, but we still achieve worse bounds. See Appendix H.2.1 for a detailed description of the various proof strategies.

trick. We apply this trick to get a better cheap bound on the error terms of low-rank approximations, without having to multiply out the full matrices. See Appendices G.2.2, and F for more details.

## 5 Results

We run each of 151 transformers on the various proof strategies of different asymptotic complexity, and analyze these proofs to empirically examine the relationship between proof length, bound tightness, and degree of understanding. For each proof on each transformer, we approximate the length of the proof by estimating the number of FLOPs used, and plot this against the ratio of certified bound the true accuracy $b/\bar{s}$ (Equation 2) in Figure 3. There exists a clear trade-off between bound tightness and compactness of the proof – more compact proofs yield looser bounds, and tighter bounds are associated with more expensive proofs.

### 5.1 Compact proofs both require and provide mechanistic understanding

**Quantifying mechanistic understanding using unexplained dimensionality** We first quantify the amount of mechanistic understanding used in a proof by measuring its **unexplained dimensionality** – the number of free parameters required to fully describe model behavior, assuming the structural assumptions of the proof are correct. More detailed interpretations leave fewer free parameters needing to be filled in via empirical observation (Appendix A.5). In Figure 5, we plot these axes and find a suggestive correlation: proofs based on less mechanistic understanding are longer.

**More mechanistic understanding allows for more compact proofs** In addition to the constructions in Section 4, the parts of proofs we were unable to compact seem to correspond to components that we do not mechanistically understand. For example, we could not cheaply bound the behavior of EVOU without multiplying out the matrices, and this seems in part because we do have a mechanistic understanding of how EVOU implements low-rank copying.

**Compact proofs seem to provide understanding** By examining compact proofs, we can extract understanding about the model. For example, the fact that replacing each row of EU with its average across rows has little effect on the bound implies that EU does not vary much based on $t_{\text{query}}$.

### 5.2 Proof length vs. bound tightness trade-off is modulated by faithfulness of interpretation

**Compact proofs are less faithful to model internals** To derive more compact proofs, we use our mechanistic understanding to simplify the model computation in ways that diverge from the original model internals. For example, in some subcubic proofs (Section 4.3), we approximate EQKE with a rank-1 approximation corresponding to the "size direction". However, while other components are small, they're nonzero; this approximation harms model internals.

**Less faithful interpretations lead to worse bounds on performance** To confirm that faithfulness of understanding affects the tightness of bound independent of proof length, we plot the normalized accuracy bound of subcubic proofs that perform a rank-1 approximation to EQKE, versus the ratio of the first two singular components. A larger ratio between the components implies that the rank-1 approximation is more faithful. In Figure 4, we see a positive correlation between the two axes: when the interpretation is more faithful, the bounds are tighter, even at a fixed proof length.

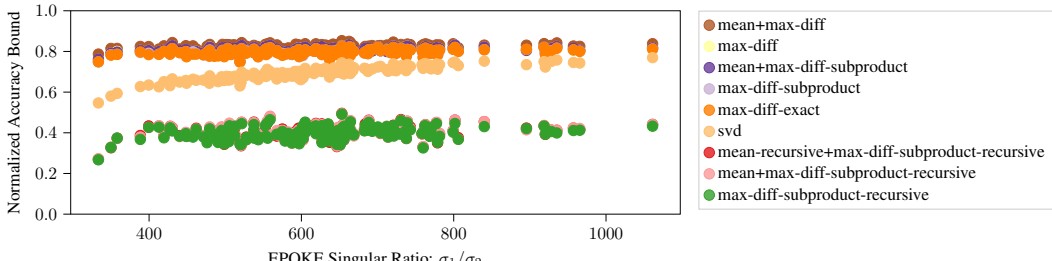

**Figure 4:** We plot the normalized accuracy bound versus the ratio of first and second singular values of EQKE, for various types of subcubic proofs that depend on a rank-1 approximation EQKE. For each class of proof, the closer EQKE is to rank-1, the tighter the accuracy bound. This suggests that more faithful interpretations lead to tighter bounds even holding proof length fixed. See Appendix H.2.2 for a detailed description of proof strategies.

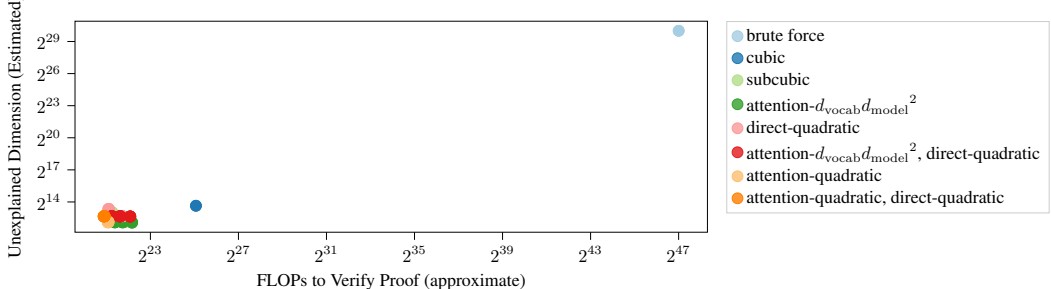

**Figure 5:** We plot, for each proof, the approximate number of flops required to evaluate the proof, versus the unexplained dimensionality (Section 5.1). More mechanistic understanding leaves fewer dimensions unexplained. We observe that more compact proofs seem to leave fewer unexplained dimensions, which is indicative of the relationship of mechanistic understanding and compact proofs. See Appendix H.2.1 for more detail.

### 5.3 Compounding structureless noise is a big challenge for compacting global-behavior proofs

**Pessimal error terms compound in the absence of known structure** The rank-1 approximation of EQKE has small error. However, when making rank-1 approximations of each of the constituent matrices $E, Q, K$, pessimizing over the worst way to composing the individual small error terms leads to a bound on the error term of EQKE that is orders of magnitude larger than the actual error term. Because we don't understand how the matrices compose in a way that doesn't cause errors to compound (without just multiplying out the matrices), this approximation leads to a trivial bound on performance (Appendix G.2.5). We speculate that in many cases, there is no short human-interpretable description for why random noise or approximation errors do not compound across layers of neural networks (e.g., see the error correction results on *randomly initialized* neural networks from Hänni et al. [21]), and thus that compounding structureless errors may be an issue in practice.

## 6 Related Work

**Generalization Bounds** Prior work in the PAC-Bayes framework [58, 36, 12] proves generalization bounds over learning procedures, which are similar to the global performance bounds we consider in this work. These proofs tend to provide statistical guarantees [25, 26] about the outputs of a known stochastic training procedure, while we seek to bound the performance of particular trained models.

**Formally verifying neural networks** Most prior work formally verifies neural networks either via model checking [28, 7] or by relaxing the problem setting and taking an automated theorem proving approach [17, 50, 18, 35, 43] to verify *local* robustness properties. These proof strategies tend to be derived by examining only the network architecture. We take an approach more akin to interactive theorem proving [22] and verify *global* performance properties by reverse-engineering the weights.

**Mechanistic Interpretability** Finally, mechanistic interpretability is the subfield of the broader field of understanding model internals [45], which is too large to faithfully summarize. Our work takes most direct inspiration from efforts to deeply understand how either toy models [38, 9, 53, 2] or small pretrained text transformers [54, 20] implement algorithmic tasks, generally by performing ablations and SVD. In contrast, we formally prove that a transformer implements an algorithm.

Nichani et al. [39] proves that, in a significantly simplified 2-layer, 1-head attention-only transformer model and for the task of in-context bigram statistics, gradient descent will create induction heads [40]. Our results concern transformers with fixed weights. In concurrent work, Michaud et al. [34] use techniques inspired by mechanistic interpretability to perform automated program synthesis on 2-dimensional RNNs, while our work works with significantly larger transformer models.

## 7    Conclusion and Future Work

**Summary** In this work, we used a Max-of-$K$ setting to prototype the use of mechanistic interpretability to derive compact proofs of model behavior. Using varying amounts of understanding, we derived more efficient proof computations lower bounding model accuracy. We found preliminary evidence that mechanistic understanding can compactify proofs. Moreover, we observed that the tightness of the lower bound offered by various proof strategies can be used to grade the faithfulness our mechanistic interpretation. Finally, we identified compounding structureless errors as a key obstacle to deriving compact proofs of model behavior.

**Limitations and future work** We study one-layer attention-only transformers on a toy algorithmic task. Future work should explore the viability of deriving proofs via interpretability using larger models featuring MLPs or layernorm on more complex domains. In addition, we were unable to significantly compact the part of the proof involving the OV circuit, which future work can explore. The proofs we explored in this work also did not lead to qualitatively novel insights; future work may be able to derive such insights with improved techniques. Finally, future work can address the problem of compounding structureless errors, perhaps by relaxing from worst-case pessimal ablations to typical-case heuristic guarantees [8].

## Acknowledgments and Disclosure of Funding

We are immensely grateful to Paul Christiano for providing the initial support for this project and for his invaluable research advice, encouragement, and feedback throughout its duration.

Additionally, we are thankful for clarifying discussions and feedback from Jacob Hilton, Matthew Coudron, Adrià Garriga-Alonso, Aryan Bhatt, Leo Gao, Jenny Nitishinskaya, Somsubhro Bagchi, Gabriel Wu, Erik Jenner, Ryan Greenblatt, Ronak Mehta, Louis Jaburi, and many others. Louis Jaburi in particular contributed the text of the final proof of Theorem 11 in Appendix E.

We are indebted to various organizations for their support: **Alignment Research Center** for funding this project and making it possible at all; **Mentorship for Alignment Research Students** (MARS) program of the **Cambridge AI Safety Hub** (CAISH) for setting up the collaboration between a subset of authors, and providing funding for compute and in-person research sprints; **Constellation** and **FAR Labs** for hosting a subset of the authors and providing an excellent research environment, including as part of the Visiting Fellows Program and Astra Fellowship.

### Author Contributions

**Jason Gross** led the project, including managing the team and conceptualizing the proofs approach. He ran the Max-of-4 experiments, devised the proof strategies, and wrote up the formal proofs. He worked on various case studies and developed general methodology for computing complexity and length bounds for proofs. He also developed the particular convex relaxations presented in the paper.

**Rajashree Agrawal** was invaluable in steering the direction of the project, including contributing to the preliminary experiment on Max-of-2 and developing the pessimal ablation approach. She worked on framing the results, and contributed text to the paper.

**Thomas Kwa** and **Euan Ong** extended the preliminary experiments to larger values of $k$ and contributed substantially to the cubic proof. **Chun Hei Yip**, **Alex Gibson**, and **Soufiane Noubir** worked on case studies other than the Max-of-$K$ task and informed discussion on proof complexity.

**Lawrence Chan** spearheaded the writing of the paper, including turning informal claims into formal theorem statements, creating figures, and writing the core text. He also developed the unexplained dimensionality metric for clarifying the takeaway of the paper.

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

# A   Subtleties of our approach

In this section, we address some subtleties and frequently asked questions about our approach.

## A.1   Why study this simple task?

Formal reasoning is computationally expensive; very few large software projects have ever been verified [31, 30], none of them comparable to large transformer models [10, 19]. Separately, there is a high fixed cost to taking on any verification project, regardless of computational efficiency of the verification itself. Thus, we picked the simplest setting to study the question of interest: Is it even possible to formally reason more efficiently than by brute force about model behavior?

## A.2   Scalability

In this section, we address concerns about the scalability of our approach.

### A.2.1   Larger input spaces

We demonstrate that our proof strategies can be reused on larger input spaces while scaling better than the brute force approach does.

We applied our proof strategies to models trained for Max-of-5, Max-of-10, and Max-of-20. While running the brute force proof on Max-of-20 would require approximately $2^{148}$ FLOPs, which is about $2^{70}\times$ the cost of training GPT-3 [32], our cubic proof achieves bounds of $(94.1 \pm 1.1)\,\%$ (Max-of-5), $(91.4 \pm 2.1)\,\%$ (Max-of-10), and $(88.4 \pm 4.0)\,\%$ (Max-of-20) in under two minutes. See Tables 2, 3, 4, and 5 for more detailed numbers, and Figures 6, and 7 for visualizations. These results demonstrate that proof strategies can be reused on larger input spaces while scaling better than the brute force approach does.

### A.2.2   Different tasks

In this paper, we worked on highly optimizing our relaxation to make our bounds as tight as possible when incorporating as little understanding as possible. This is not necessary for deriving proofs. Our general formalization of mechanistic interpretability is replicable: (1) theorem statements are exact expressions for the difference between the actual behavior of the model and the purported behavior, and (2) proofs are computations that bound the expression. Furthermore, our convexity theorems and proofs are applicable much more generally generally to element retrieval tasks.

### A.2.3   More complicated architectures

We worked on a simple model studied in *A Mathematical Framework for Transformer Circuits* [13]. In follow-up work, we will extend this approach to proving bounds on 1L transformers with ReLU MLP trained on modular addition.

### A.2.4   Larger models

It is an open question whether or not the mechanistic interpretability approach to proofs can scale to larger models. However, a large part of this question lies in the feasibility of deriving a high degree of faithful mechanistic understanding from large models — that is, whether mechanistic interpretability itself will scale. This is widely recognized in the field, and scaling interpretability approaches while getting both a high degree of mechanistic understanding and assurances that said understanding is faithful to the model is an active area of research. Broadly, we see the compact proofs approach as a metric on the quality of mechanistic understanding — we are not purporting to have a general solution to the problem of scaling interpretability, but instead claim that the challenges in proofs are in fact challenges in understanding networks.

**Table 2:** Version of Table 1 from Section 4.1 with $n_{\text{ctx}} = 5$, $d_{\text{vocab}} = 64$. We report the proof complexity, accuracy bound, and estimated flops required (Equation 2), as well as unexplained dimensionality (Section 5). Unlike Table 1, which computes the brute force bound exactly, we instead use importance sampling to estimate the bound; estimated FLOPs are reported for what the full brute force proof would take. We round the FLOP and unexplained dimension counts to the closest power of 2, and report the mean/standard deviation of the bound averaged across all 151 models. For space reasons, we use $k := n_{\text{ctx}}$, $d := d_{\text{model}}$, and $v := d_{\text{vocab}}$.

| Description of Proof | Complexity Cost | Bound | Est. FLOPs | Unexplained Dimensions |
|---|---|---|---|---|
| Brute force | $\mathcal{O}(v^{k+1}kd)$ | $0.9990 \pm 0.0018$ | $2^{54}$ | $2^{36}$ |
| Cubic | $\mathcal{O}(v^3k^2)$ | $0.941 \pm 0.011$ | $2^{26}$ | $2^{14}$ |
| Sub-cubic | $\mathcal{O}(v^2 \cdot k^2 + v^2 \cdot d)$ | $0.705 \pm 0.031$ | $2^{22}$ | $2^{13}$ |
|   w/o mean+diff | | $0.405 \pm 0.073$ | $2^{22}$ | $2^{13}$ |
| Low-rank QK | $\mathcal{O}(v^2k^2 + \underbrace{vd^2}_{\text{QK}} + \underbrace{v^2d}_{\text{EU\&OV}})$ | $0.682 \pm 0.033$ | $2^{22}$ | $2^{12}$ |
|   SVD only | | $0.335 \pm 0.066$ | $2^{22}$ | $2^{12}$ |
| Low-rank EU | $\mathcal{O}(v^2k^2 + \underbrace{vd}_{\text{EU}} + \underbrace{v^2d}_{\text{QK\&OV}})$ | $0.649 \pm 0.055$ | $2^{21}$ | $2^{13}$ |
|   SVD only | | $(4.8 \pm 0.1) \times 10^{-8}$ | $2^{21}$ | $2^{13}$ |
| Low-rank QK&EU | $\mathcal{O}(v^2k^2 + \underbrace{vd^2}_{\text{QK}} + \underbrace{vd}_{\text{EU}} + \underbrace{v^2d}_{\text{OV}})$ | $0.628 \pm 0.053$ | $2^{22}$ | $2^{13}$ |
|   SVD only | | $(4.8 \pm 0.1) \times 10^{-8}$ | $2^{22}$ | $2^{13}$ |
| Quadratic QK | $\mathcal{O}(v^2k^2 + \underbrace{vd}_{\text{QK}} + \underbrace{v^2d}_{\text{EU\&OV}})$ | $0.354 \pm 0.034$ | $2^{21}$ | $2^{12}$ |
| Quadratic QK&EU | $\mathcal{O}(v^2k^2 + \underbrace{vd}_{\text{QK\&EU}} + \underbrace{v^2d}_{\text{OV}})$ | $0.335 \pm 0.033$ | $2^{21}$ | $2^{13}$ |

**Table 3:** Version of Table 1 from Section 4.1 with $n_{\text{ctx}} = 10$, $d_{\text{vocab}} = 64$. We report the proof complexity, accuracy bound, and estimated flops required (Equation 2), as well as unexplained dimensionality (Section 5). Unlike Table 1, which computes the brute force bound exactly, we instead use importance sampling to estimate the bound; estimated FLOPs are reported for what the full brute force proof would take. We round the FLOP and unexplained dimension counts to the closest power of 2, and report the mean/standard deviation of the bound averaged across all 151 models. For space reasons, we use $k := n_{\text{ctx}}$, $d := d_{\text{model}}$, and $v := d_{\text{vocab}}$.

| Description of Proof | Complexity Cost | Bound | Est. FLOPs | Unexplained Dimensions |
|---|---|---|---|---|
| Brute force | $\mathcal{O}(v^{k+1}kd)$ | $0.9988 \pm 0.0013$ | $2^{86}$ | $2^{66}$ |
| Cubic | $\mathcal{O}(v^3k^2)$ | $0.914 \pm 0.021$ | $2^{28}$ | $2^{14}$ |
| Sub-cubic | $\mathcal{O}(v^2 \cdot k^2 + v^2 \cdot d)$ | $0.674 \pm 0.028$ | $2^{23}$ | $2^{13}$ |
|   w/o mean+diff | | $0.539 \pm 0.061$ | $2^{23}$ | $2^{13}$ |
| Low-rank QK | $\mathcal{O}(v^2k^2 + \underbrace{vd^2}_{\text{QK}} + \underbrace{v^2d}_{\text{EU\&OV}})$ | $0.657 \pm 0.028$ | $2^{23}$ | $2^{12}$ |
|   SVD only | | $0.469 \pm 0.059$ | $2^{23}$ | $2^{12}$ |
| Low-rank EU | $\mathcal{O}(v^2k^2 + \underbrace{vd}_{\text{EU}} + \underbrace{v^2d}_{\text{QK\&OV}})$ | $0.639 \pm 0.032$ | $2^{23}$ | $2^{13}$ |
|   SVD only | | $(0 \pm 100) \times 10^{-12}$ | $2^{22}$ | $2^{13}$ |
| Low-rank QK&EU | $\mathcal{O}(v^2k^2 + \underbrace{vd^2}_{\text{QK}} + \underbrace{vd}_{\text{EU}} + \underbrace{v^2d}_{\text{OV}})$ | $0.625 \pm 0.031$ | $2^{23}$ | $2^{13}$ |
|   SVD only | | $(2.9 \pm 0.1) \times 10^{-17}$ | $2^{23}$ | $2^{13}$ |
| Quadratic QK | $\mathcal{O}(v^2k^2 + \underbrace{vd}_{\text{QK}} + \underbrace{v^2d}_{\text{EU\&OV}})$ | $0.392 \pm 0.030$ | $2^{22}$ | $2^{12}$ |
| Quadratic QK&EU | $\mathcal{O}(v^2k^2 + \underbrace{vd}_{\text{QK\&EU}} + \underbrace{v^2d}_{\text{OV}})$ | $0.390 \pm 0.028$ | $2^{22}$ | $2^{13}$ |

**Table 4:** Version of Table 1 from Section 4.1 with $n_{\text{ctx}} = 10$ and $d_{\text{vocab}} = 128$. We report the proof complexity, accuracy bound, and estimated flops required (Equation 2), as well as unexplained dimensionality (Section 5). Unlike Table 1, which computes the brute force bound exactly, we instead use importance sampling to estimate the bound; estimated FLOPs are reported for what the full brute force proof would take. We round the FLOP and unexplained dimension counts to the closest power of 2, and report the mean/standard deviation of the bound averaged across all 151 models. For space reasons, we use $k := n_{\text{ctx}}$, $d := d_{\text{model}}$, and $v := d_{\text{vocab}}$.

| Description of Proof | Complexity Cost | Bound | Est. FLOPs | Unexplained Dimensions |
|---|---|---|---|---|
| Brute force | $\mathcal{O}(v^{k+1}kd)$ | $0.9972 \pm 0.0031$ | $2^{96}$ | $2^{77}$ |
| Cubic | $\mathcal{O}(v^3k^2)$ | $0.882 \pm 0.012$ | $2^{31}$ | $2^{16}$ |
| Sub-cubic | $\mathcal{O}(v^2 \cdot k^2 + v^2 \cdot d)$ | $0.622 \pm 0.031$ | $2^{24}$ | $2^{15}$ |
| w/o mean+diff | | $0.390 \pm 0.070$ | $2^{24}$ | $2^{15}$ |
| Low-rank QK | $\mathcal{O}(v^2k^2 + \underbrace{vd^2}_{QK} + \underbrace{v^2d}_{EU\&OV})$ | $0.594 \pm 0.035$ | $2^{24}$ | $2^{14}$ |
| SVD only | | $0.320 \pm 0.053$ | $2^{25}$ | $2^{14}$ |
| Low-rank EU | $\mathcal{O}(v^2k^2 + \underbrace{vd}_{EU} + \underbrace{v^2d}_{QK\&OV})$ | $0.607 \pm 0.031$ | $2^{24}$ | $2^{15}$ |
| SVD only | | $(5.4 \pm 0.2) \times 10^{-20}$ | $2^{24}$ | $2^{15}$ |
| Low-rank QK&EU | $\mathcal{O}(v^2k^2 + \underbrace{vd^2}_{QK} + \underbrace{vd}_{EU} + \underbrace{v^2d}_{OV})$ | $0.595 \pm 0.030$ | $2^{24}$ | $2^{14}$ |
| SVD only | | $(5.4 \pm 0.2) \times 10^{-20}$ | $2^{25}$ | $2^{14}$ |
| Quadratic QK | $\mathcal{O}(v^2k^2 + \underbrace{vd}_{QK} + \underbrace{v^2d}_{EU\&OV})$ | $0.350 \pm 0.029$ | $2^{24}$ | $2^{14}$ |
| Quadratic QK&EU | $\mathcal{O}(v^2k^2 + \underbrace{vd}_{QK\&EU} + \underbrace{v^2d}_{OV})$ | $0.384 \pm 0.025$ | $2^{24}$ | $2^{14}$ |

**Table 5:** Version of Table 1 from Section 4.1 with $n_{\text{ctx}} = 20$, $d_{\text{vocab}} = 64$. We report the proof complexity, accuracy bound, and estimated flops required (Equation 2), as well as unexplained dimensionality (Section 5). Unlike Table 1, which computes the brute force bound exactly, we instead use importance sampling to estimate the bound; estimated FLOPs are reported for what the full brute force proof would take. We round the FLOP and unexplained dimension counts to the closest power of 2, and report the mean/standard deviation of the bound averaged across all 151 models. For space reasons, we use $k := n_{\text{ctx}}$, $d := d_{\text{model}}$, and $v := d_{\text{vocab}}$.

| Description of Proof | Complexity Cost | Bound | Est. FLOPs | Unexplained Dimensions |
|---|---|---|---|---|
| Brute force | $\mathcal{O}(v^{k+1}kd)$ | $0.995 \pm 0.015$ | $2^{148}$ | $2^{126}$ |
| Cubic | $\mathcal{O}(v^3k^2)$ | $0.884 \pm 0.040$ | $2^{29}$ | $2^{14}$ |
| Sub-cubic | $\mathcal{O}(v^2 \cdot k^2 + v^2 \cdot d)$ | $0.561 \pm 0.043$ | $2^{24}$ | $2^{13}$ |
| w/o mean+diff | | $0.486 \pm 0.060$ | $2^{24}$ | $2^{13}$ |
| Low-rank QK | $\mathcal{O}(v^2k^2 + \underbrace{vd^2}_{QK} + \underbrace{v^2d}_{EU\&OV})$ | $0.547 \pm 0.043$ | $2^{24}$ | $2^{12}$ |
| SVD only | | $0.431 \pm 0.060$ | $2^{24}$ | $2^{12}$ |
| Low-rank EU | $\mathcal{O}(v^2k^2 + \underbrace{vd}_{EU} + \underbrace{v^2d}_{QK\&OV})$ | $0.538 \pm 0.043$ | $2^{24}$ | $2^{13}$ |
| SVD only | | $(1.0 \pm 6.0) \times 10^{-4}$ | $2^{24}$ | $2^{13}$ |
| Low-rank QK&EU | $\mathcal{O}(v^2k^2 + \underbrace{vd^2}_{QK} + \underbrace{vd}_{EU} + \underbrace{v^2d}_{OV})$ | $0.526 \pm 0.041$ | $2^{24}$ | $2^{13}$ |
| SVD only | | $(1.0 \pm 5.0) \times 10^{-4}$ | $2^{24}$ | $2^{13}$ |
| Quadratic QK | $\mathcal{O}(v^2k^2 + \underbrace{vd}_{QK} + \underbrace{v^2d}_{EU\&OV})$ | $0.322 \pm 0.035$ | $2^{24}$ | $2^{12}$ |
| Quadratic QK&EU | $\mathcal{O}(v^2k^2 + \underbrace{vd}_{QK\&EU} + \underbrace{v^2d}_{OV})$ | $0.321 \pm 0.035$ | $2^{24}$ | $2^{13}$ |

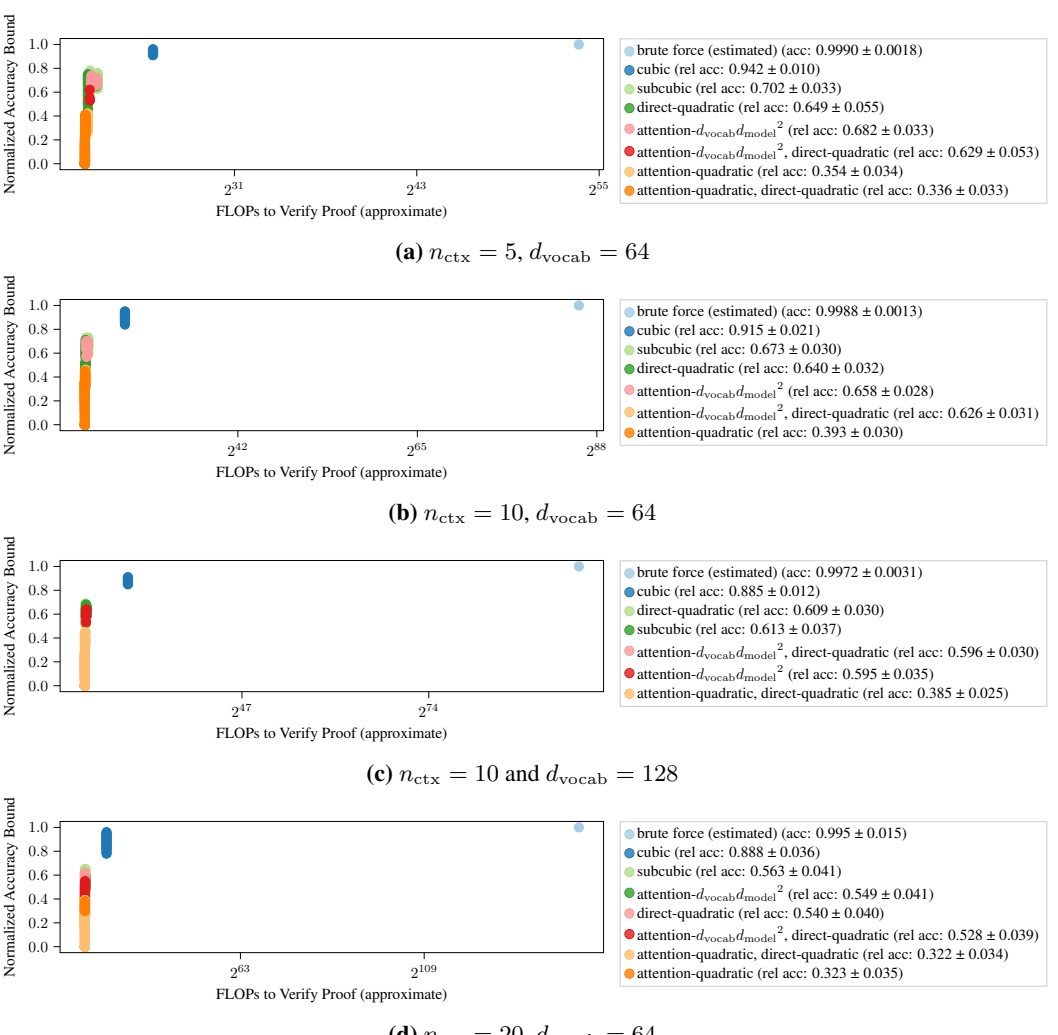

**(a)** $n_{\mathrm{ctx}} = 5$, $d_{\mathrm{vocab}} = 64$

**(b)** $n_{\mathrm{ctx}} = 10$, $d_{\mathrm{vocab}} = 64$

**(c)** $n_{\mathrm{ctx}} = 10$ and $d_{\mathrm{vocab}} = 128$

**(d)** $n_{\mathrm{ctx}} = 20$, $d_{\mathrm{vocab}} = 64$

**Figure 6:** Version of Figure 3 from page 8 with varying $n_{\mathrm{ctx}}$ and $d_{\mathrm{vocab}}$. The brute-force proof (Section 4.1) computes the exact performance uses orders of magnitude more compute than other approaches; unlike in Figure 3, here we use importance sampling to estimate the bound.

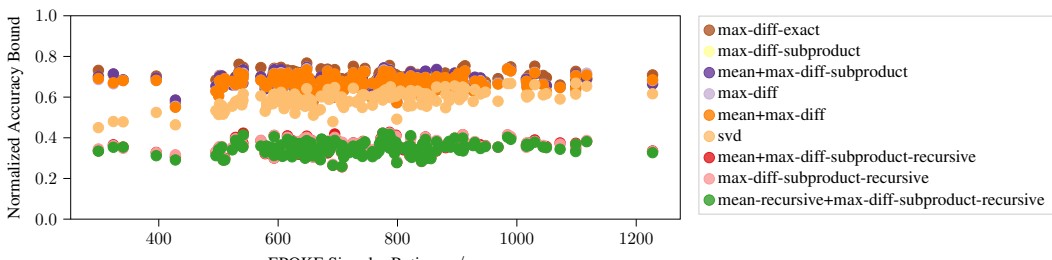

**(a)** $n_{\text{ctx}} = 5$, $d_{\text{vocab}} = 64$. The "svd" proof strategy best-fit line has equation $b/\bar{s} = 0.000\,15(\sigma_1/\sigma_2) + 0.48$, $R^2 = 0.37$.

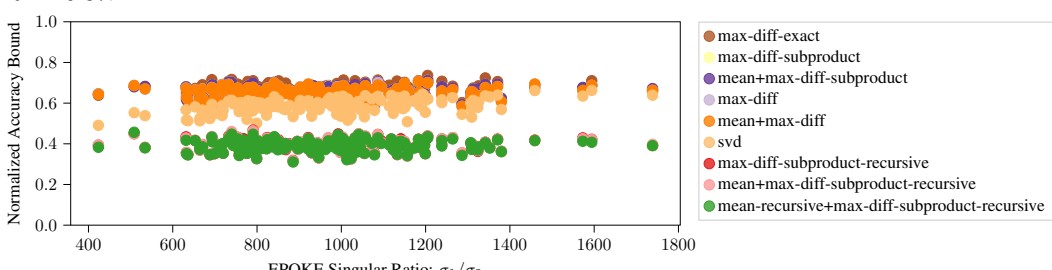

**(b)** $n_{\text{ctx}} = 10$, $d_{\text{vocab}} = 64$. The "svd" proof strategy best-fit line has equation $b/\bar{s} = 0.000\,074(\sigma_1/\sigma_2) + 0.51$, $R^2 = 0.23$.

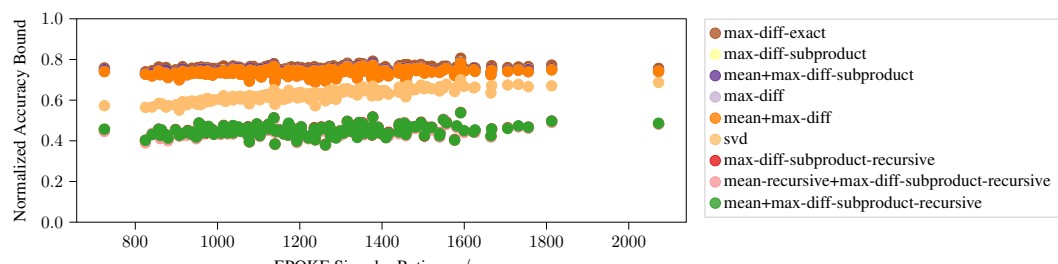

**(c)** $n_{\text{ctx}} = 10$ and $d_{\text{vocab}} = 128$. The "svd" proof strategy best-fit line has equation $b/\bar{s} = 0.000\,085(\sigma_1/\sigma_2) + 0.40$, $R^2 = 0.42$.

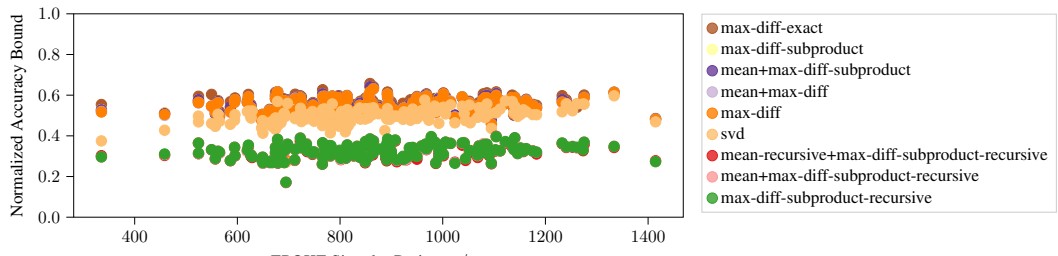

**(d)** $n_{\text{ctx}} = 20$, $d_{\text{vocab}} = 64$. The "svd" proof strategy best-fit line has equation $b/\bar{s} = 0.000\,098(\sigma_1/\sigma_2) + 0.41$, $R^2 = 0.22$.

**Figure 7:** Version of Figure 4 from page 9 with varying $n_{\text{ctx}}$ and $d_{\text{vocab}}$. Note that the "svd" proof strategy has a clear upward trend, especially on early points.

## A.3 Why is more mechanistic understanding correlated with worse bounds?

Figure 3 exhibits Simpson's Paradox: although more faithful mechanistic understanding is correlated with better bounds within each class of proof (and moreover the most extensive mechanistic understanding results in the greatest improvement in bound tightness over baseline), when we aggregate across all proof strategies, we find that more mechanistic understanding is correlated with worse bounds.

This relationship is summarized in Figure 8.

From the compression perspective, more mechanistic understanding is about having more compression. Unless the model is losslessly compressible, we should expect that more compression will inherently be more lossy, no matter how good our compression scheme is. Correspondingly, using more understanding to get more compression will often result in a weaker bound, no matter how good our understanding is.

Conversely, we can think of the quality of proofs (the combination of tightness of bound, and length of proof) as a metric for how good our mechanistic understanding is. From this perspective, the fact that mechanistic-interpretability-derived bounds are bad suggests gaps in our mechanistic understanding. As the field matures and we develop tools that enable more faithful and complete understanding of model behavior, we expect that the quality of bounds we derive from mechanistic understanding will improve.

## A.4 Convex relaxation

In this work, we construct convex relaxations to perform the pessimal ablations for our proofs.

In what sense are we using "convexity"? The intuition is that we are attempting to optimize a function $f$ over its domain $X$ by incrementally making local changes to the input sequence, such as replacing one token by another, or by changing the order of tokens. The reason that convex optimization problems are easy to solve is that all local extrema are global extrema. This is not the case for our optimization problem, so we find a relaxation of $f$ and its domain such that all local extrema are in fact global extrema.

Furthermore, most convex optimizers perform optimization at runtime by repeatedly stepping towards extrema. In this work, we "optimize by hand", performing the optimization in the proof of our general theorems. The computation of the bound then only needs to instantiate the precomputed possible extrema with the actual values of the model's parameters to determine the the extrema actually are.

We now give a formal description of what we mean by "convex relaxation".

For a set of inputs $X_i$, we define a set of "relaxed inputs" $X_i^{\text{relaxed}}$ with an injection $T_i : X_i \hookrightarrow \mathcal{P}(X_i^{\text{relaxed}})$ mapping input to the model to the set of corresponding relaxed inputs. On the relaxed input, we define a function $h_i : X_i^{\text{relaxed}} \to \mathbb{R}$ such that for all $\mathbf{t} \in X_i$ and all labels $l$ for which $(l, \mathbf{t})$ is supported by (has non-zero probability in) $\mathcal{D}$, we can find $\mathbf{t}^{\text{relaxed}} \in T_i(\mathbf{t})$ with $f(l, \mathcal{M}(\mathbf{t})) \geq h_i(\mathbf{t}^{\text{relaxed}})$. We proceed by finding a small subset of "boundary" examples $B_i \subset X_i^{\text{relaxed}}$, proving that if $h_i(\mathbf{t}^{\text{relaxed}}) \geq b_i$ for all $\mathbf{t}^{\text{relaxed}} \in B_i$ then $h_i(\mathbf{t}^{\text{relaxed}}) \geq b_i$ for all $\mathbf{t}^{\text{relaxed}} \in X_i^{\text{relaxed}}$.

Then, the computational component $C$ of the proof validates that that $h_i(\mathbf{t}^{\text{relaxed}}) \geq b_i$ for some $b_i$ for all $\mathbf{t}^{\text{relaxed}} \in X_i^{\text{relaxed}}$. This allows us to conclude that $f(l, \mathcal{M}(\mathbf{t})) \geq b_i$ for all $\mathbf{t} \in X_i$.

## A.5 Computing unexplained dimensionality

We claim in Figure 5 that we can use unexplained dimensionality as a metric for understanding. Here we describe how we compute the unexplained dimensionality of a proof strategy.

As in Figure 1, for any given proof, we can separate our treatment of transformer components into "black-box" (e.g., matrix multiplication) and "white-box" components (e.g., specifying that the QK circuit is approximately rank one; pessimizing over non-max tokens). Considering the performance score as a large white-box component which may reference black-boxes internally, we define the unexplained dimensionality of a single black-box computation as the log-cardinality of it function space (so, e.g, $2 \cdot 64$ for a function $\underline{64} \to \mathbb{R}^2$, whose cardinality is $(\mathbb{R}^2)^{\underline{64}}$, where $\underline{64}$ denotes the finite

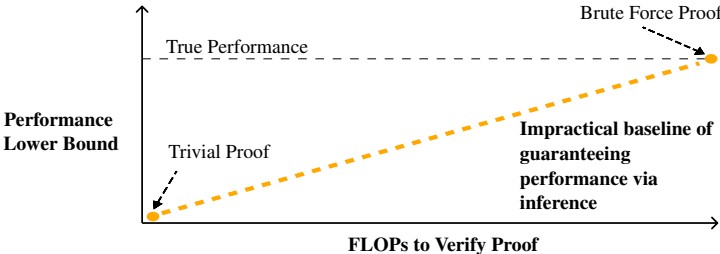

**(a)** The baseline of using inference to generate proofs.

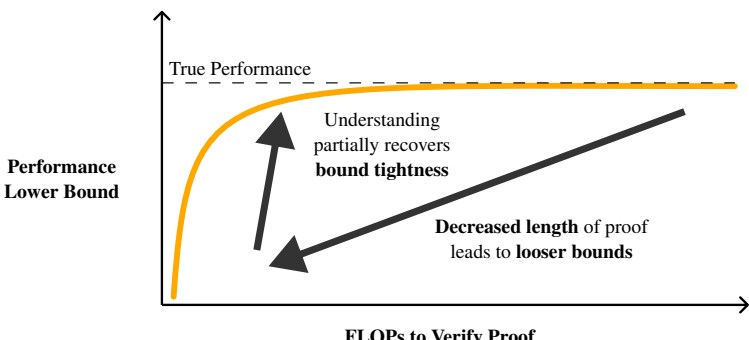

**(b)** Shorter proofs by default have a worse performance bound $b$. Faithful understanding allows us to recover significant — but not complete — bound tightness with minimal proof-length overhead.

**Figure 8:** The theoretical relationship between proof length and bound tightness.

set on 64 elements). The unexplained dimensionality of the entire proof is the sum of the unexplained dimensions of all black-box components.

Intuitively speaking, unexplained dimensionality tries to capture the degrees of freedom that we have to check via brute enumeration over black-box computations. Proofs with less unexplained dimensionality contain more mechanistic understanding, and vice versa.

### A.6  Computing approximate FLOPs

In Figure 3 and Table 1 on page 6 and on page 8, we display approximate floating point operations. We instrument our code to execute on phantom tensors that track their shape and accumulate an approximate count of floating point operations. We compute matrix additions and multiplications in the obvious way. We take the instruction count of SVD to be the cost of verifying that the output of SVD is a valid decomposition: that we have a pair of orthonormal bases which when multiplied out give the original basis.

### A.7  IEEE 754 vs. $\mathbb{R}$

In Section 2 we defined $C$ and $Q$ and glossed over whether we were reasoning over reals or floats. Here we clarify this point that we've so far been sweeping under the rug.

Let $\mathbb{F}$ denote the set of the relevant flavor of IEEE 754 Floating Point numbers (generally 32-bit for our concrete models, but everything would hold just as well for 64-bit). Let $\mathbb{F}^*$ denote $\mathbb{F}$ restricted to finite numbers (that is, without NaNs and without $\pm\infty$).

We parameterize $C$, $\mathcal{M}$, and $\mathcal{D}$ over the real field[13] they operate on, so that, e.g., $C_F$ : model weights $\to F$. Then we have $Q$ establishing that for any model $\mathcal{M}'$, $C_{\mathbb{R}}(\mathcal{M}'_{\mathbb{R}}) \le \mathbb{E}_{(l,\mathbf{t})\sim\mathcal{D}_{\mathbb{R}}} f_{\mathbb{R}}(l, \mathcal{M}'_{\mathbb{R}}(\mathbf{t}))$, and we have a trace demonstrating that $C_{\mathbb{F}}(\mathcal{M}_{\mathbb{F}}) = b$.

---

[13]Technically the floating point numbers are not a field. We gloss over this point here, since they define all of the field operations, even if those operations do not satisfy the field axioms.

Let $i : \mathbb{F}^* \to \mathbb{R}$ be any injection that maps each floating point number to some real number that it is "closest to". Supposing that $b \in \mathbb{F}^*$ and thus $b \in \mathbb{R}$, we need two additional components of the proof. We need to find $\varepsilon, \varepsilon' \in \mathbb{R}^+$ prove that

$$|C_{\mathbb{R}}(\mathcal{M}_{\mathbb{R}}) - i\left(C_{\mathbb{F}}(\mathcal{M}_{\mathbb{F}})\right)| < \varepsilon \qquad \text{and} \qquad \left|\left(\mathbb{E}_{(l,\mathbf{t}) \sim \mathcal{D}_{\mathbb{R}}} f_{\mathbb{R}}(l, \mathcal{M}_{\mathbb{R}}(\mathbf{t}))\right) - i\left(\mathbb{E}_{(l,\mathbf{t}) \sim \mathcal{D}_{\mathbb{F}}} f_{\mathbb{F}}(l, \mathcal{M}_{\mathbb{F}}(\mathbf{t}))\right)\right| < \varepsilon'$$

Then we can chain these proofs to prove that

$$i\left(\mathbb{E}_{(l,\mathbf{t}) \sim \mathcal{D}_{\mathbb{F}}} f_{\mathbb{F}}(l, \mathcal{M}_{\mathbb{F}}(\mathbf{t}))\right) \geq b - \varepsilon - \varepsilon'$$

Such $\varepsilon$-ball robustness proofs should be well within the scope of existing approaches to formal methods on neural nets, see, e.g., [44, 3, 4, 29, 1, 57]. We leave actually dealing with the gap between floating point numbers and real numbers to future work.

### A.8 Non-uniform distributions

In Equation 1 in Section 2 we defined the expected model performance as the expectation of the distribution $\mathcal{D}$:

$$\bar{s} := \mathbb{E}_{(l,\mathbf{t}) \sim \mathcal{D}} \left[f(l, \mathcal{M}(\mathbf{t}))\right] \geq b.$$

We then immediately specialized to the case where the marginalization $\mathcal{D}|_X$ of $\mathcal{D}$ over labels is uniform. As we'll see in Theorem 1 in Appendix D and Algorithm 3 in Appendix E, the bound computation is modularized between a function that bounds the performance $f(l, \mathcal{M}(\mathbf{t}))$ over a restricted collection of inputs, and a much simpler function that combines the bounds on individual cases into a bound on the expectation over the entire distribution. The per-input bound computation is CORRECTNESS in Algorithm 1 and RELAXED-CORRECTNESS-PESSIMIZING-OVER-POSITION in Algorithm 3; the expectation computation is BRUTE-FORCE in Algorithm 1 and CUBIC in Algorithm 3.

Since the expectation computation is modularized, it is straightforward to extend our approach to non-uniform distributions simply by adjusting the weighting of each region of inputs. However, if the distribution is too far off from the uniform training distribution, the bound we get may not be very good, as we may not be allocating adequate computation to the high-probability regions of the input space.

### A.9 Adversarial robustness via flexibility in $\mathcal{D}$

There is flexibility inherent in Equation 1. Normally, by out-of-distribution (OOD) or adversarial inputs, we suppose that there is a distribution $\mathcal{D}_{\text{in}}$ that's used for training and (in-distribution) validation, and another distribution $\mathcal{D}'$ that is the deployment distribution or generated by an adversary. If we had knowledge of $\mathcal{D}'$, we could compute the expected performance from inputs sampled from $\mathcal{D}'$. Even if we don't have exact knowledge of $\mathcal{D}'$, we can still define a very broad distribution $\mathcal{D}$ that covers possible $\mathcal{D}'$s.

In this work, $\mathcal{D}$ is the distribution of all $64^4$ possible valid input sequences. In addition, as our proofs partition $\mathcal{D}$ into subdistributions, and bound the performance on each subdistribution, we can bound the model's performance on any possible distribution over valid input sequences.

### A.10 Infinite distributions

In the brute force proof in Section 4.1, we run the model on the entirety of $\mathcal{D}$. This operation is straightforward when $X$ is finite. Perhaps surprisingly, we can do this even if $X$ is infinite as long as the PDF $L \times X \to \mathbb{R}$ of $\mathcal{D}$ is computable and the natural computational topology of $X$ is compact [16, 15, 14], because integration of computable functions on computable reals is computable [49].

### A.11 Using alternate loss functions

Building on the point from Appendix A.8, it is also relatively straightforward to extend our approach from bounding expected accuracy to bounding log-loss. We will see in Figure 12 that the accuracy and log-loss share a subterm $\Delta \ell_i$. Since we compute this subterm in all of our algorithms, we can

easily extend our approach to log-loss by combining $\Delta \ell_i$ directly rather than merely checking that the value is negative as we currently do in RELAXED-CORRECTNESS-PESSIMIZING-OVER-POSITION in Algorithm 3. Although this is sufficient for the brute-force and cubic proofs, for the subcubic proof using Algorithm 6 in Appendix F, we would additionally have to compute a log-loss bound for the sequences where the largest non-max token is "too close" to the max token, which we currently neglect by considering the model to get them wrong in the worst case.

### A.12  Proving upper bounds

In this work, we focus on proving lower bounds on model performance. Most of our theorems, for example in Appendices E, and F, prove two-sided bounds. Most of the other theorems can be straightforwardly adapted to proving upper bounds by swapping uses of min and max. Therefore, we expect that proving upper bounds on model performance should be straightforward.

### A.13  What proof system?

Length of proof depends on what proof system we use. We permit any proof system where proof-checking time is linear in the length of the proof. This excludes dependently typed proof systems such as Martin-Löf type theory, but such proof systems can easily be accommodated by considering a proof-checking-trace rather than the proof object itself. Alternatively, a more conventional proof system like ZF, ZFC, or the proof system underlying Isabelle/HOL should suffice.

## B  Experimental details

### B.1  Training details

To train each model, we generate 384,000 random sequences of 4 integers picked uniformly at random, corresponding to less than 2.5% of the input distribution. We use AdamW with batch_size $= 128$, lr $= 0.001$, betas $= (0.9, 0.999)$, weight_decay left at the default 0.01. We train for 1 epoch (3000 steps). Over our 151 seeds, models trained with this procedure achieve $(99.92 \pm 0.15)\,\%$ train accuracy and a loss of $(4 \pm 8) \times 10^{-3}$.[14] When qualitatively examining a single model (for example in Section 3.1 or Appendix H.1), we use the model with config seed 123, model seed[15] 613947648.

As our models as sufficiently small, we did not have to use any GPUs to accelerate training our inference. Each training run takes less than a single CPU-hour to complete. In total, the experiments in this paper took less than 1000 CPU-hours.

We use the following software packages in our work: Paszke et al. [41], Plotly Technologies Inc. [42], Nanda and Bloom [37], Rogozhnikov [47], Virtanen et al. [52], McKinney [33], Waskom [55]

### B.2  Additional details supporting our mechanistic interpretation of the model

We provide heatmaps of the matrices corresponding to the five components described/defined in Section 3, for the mainline model.

---

[14]Numbers reported as mean across training runs $\pm$ std dev across training runs of mean accuracy and loss.

[15]The model seed is deterministically pseudorandomly derived from the seed 123.

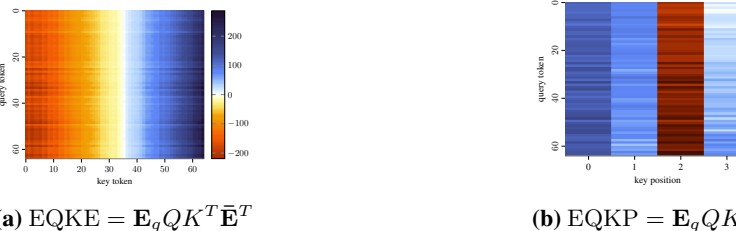

**(a)** $\mathrm{EQKE} = \mathbf{E}_q Q K^T \bar{\mathbf{E}}^T$            **(b)** $\mathrm{EQKP} = \mathbf{E}_q Q K^T \hat{\mathbf{P}}^T$

**Figure 9:** The QK circuit can be decomposed into the position-independent and position-dependent components EQKE and EQKP. It computes the pre-softmax attention score for the model. The positional contribution to the attention score, as shown in Figure (b), is minimal. In Figure (a), the gradient from left to right along the key axis indicates that the single attention head pays more attention to larger tokens. The uniformity along the query axis suggests that this behavior is largely independent of the query token. Further, the light and dark bands imply that some queries are better than others at focusing more on larger tokens.

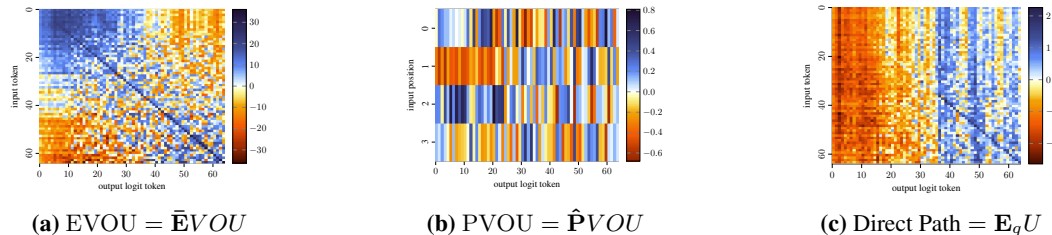

**(a)** $\mathrm{EVOU} = \bar{\mathbf{E}} VOU$     **(b)** $\mathrm{PVOU} = \hat{\mathbf{P}} VOU$     **(c)** Direct Path $= \mathbf{E}_q U$

**Figure 10:** The OV circuit is a sum of EVOU and PVOU. In Figure (a) we see that EVOU "copies" — with the exception of input tokens $\leq 5$ ($6.6 \pm 1.2$ across all models) — by virtue of the fact that above 5, the diagonal is larger than all the other elements in the same row. We see that the range on Figure (b) is much smaller than Figure (a), indicating that positional contribution to the copying is minimal. In Figure (c) we see that direct path values matter a bit more than PVOU, being only $\approx 20\times$ smaller than the typical EVOU difference. They don't matter that much, though, being so small. Additionally, the vertical banding indicates that the primary effect of this is a largely-query-independent bias towards larger numbers, reflecting the fact that the input distribution is biased towards larger numbers being the maximum. The weak diagonal pattern indicates a slight bias towards upweighting the query token itself as a (possible) maximum token.

## B.3    Distribution of model mechanisms

We provide some analysis of the distribution of the mechanisms of the models trained on the same configuration. At a glance, there is not that much variation across models.

The statistics of interest are: (1) $\sigma_1/\sigma_2$, the ratio of the first two singular values of EQKE, a measure of the extent to which the attention score computation is low-rank; (2) $\bar{s}$, the average score (accuracy) of the model across the entire input distribution; (3) $b_{\mathrm{cubic}}/\bar{s}$, the percent-score-recovered accuracy bound achieved by the cubic proof from Section 4.2; (4) $b_{\mathrm{subcubic}}/\bar{s}$, the percent-score-recovered accuracy bound achieved by the (per-model best)[16] subcubic proof from Section 4.3.

For each statistic of interest, Table 6 presents an eleven-number summary of the statistic. Plots, seeds, and statistic values are shown for models whose values are closest to each of the corresponding summary statistics.[17] Additionally, each group contains a boxplot of the summary:

- the minimum, maximum; the first and third quartiles; the median and mean; percentiles $2.15\%$, $97.85\%$, $8.87\%$, and $91.13\%$; these are displayed as:
- top and bottom of the vertical whisker lines; top and bottom of the box; horizontal line inside the box, and the square; horizontal whisker lines and whisker crosshatches.

---

[16]"Per-model best" here means that for each model seed, we select the variant of the subcubic proof with the highest bound.

[17]If a single model is the closest to two statistics, for example when the mean and median are very similar, the model is shown only once.

**Table 6:** Plots of various models. The statistics of interest are: (1) $\sigma_1/\sigma_2$, the ratio of the first two singular values of EQKE, a measure of the extent to which the attention score computation is low-rank; (2) $\bar{s}$, the average score (accuracy) of the model across the entire input distribution; (3) $b_{\text{cubic}}/\bar{s}$, the percent-score-recovered accuracy bound achieved by the cubic proof from Section 4.2; and (4) $b_{\text{subcubic}}/\bar{s}$, the percent-score-recovered accuracy bound achieved by the (per-model best) subcubic proof from Section 4.3. The $y$ axes are: for EQKE and EQKP, the query token; for EVOU, the input token; for $\mathbf{E}_q U$, the input query token; for PVOU, the input position. The $x$ axes are: for EQKE, the key token; for EQKP, the key position; for EVOU, PVOU, and $\mathbf{E}_q U$, the output logit token. All token axes range from 0 at the top (or left) to $d_{\text{vocab}} - 1 = 63$ at the bottom (or right). All position axes range from 0 at the top (or left) to $n_{\text{ctx}} - 1 = 3$ at the bottom (or right).

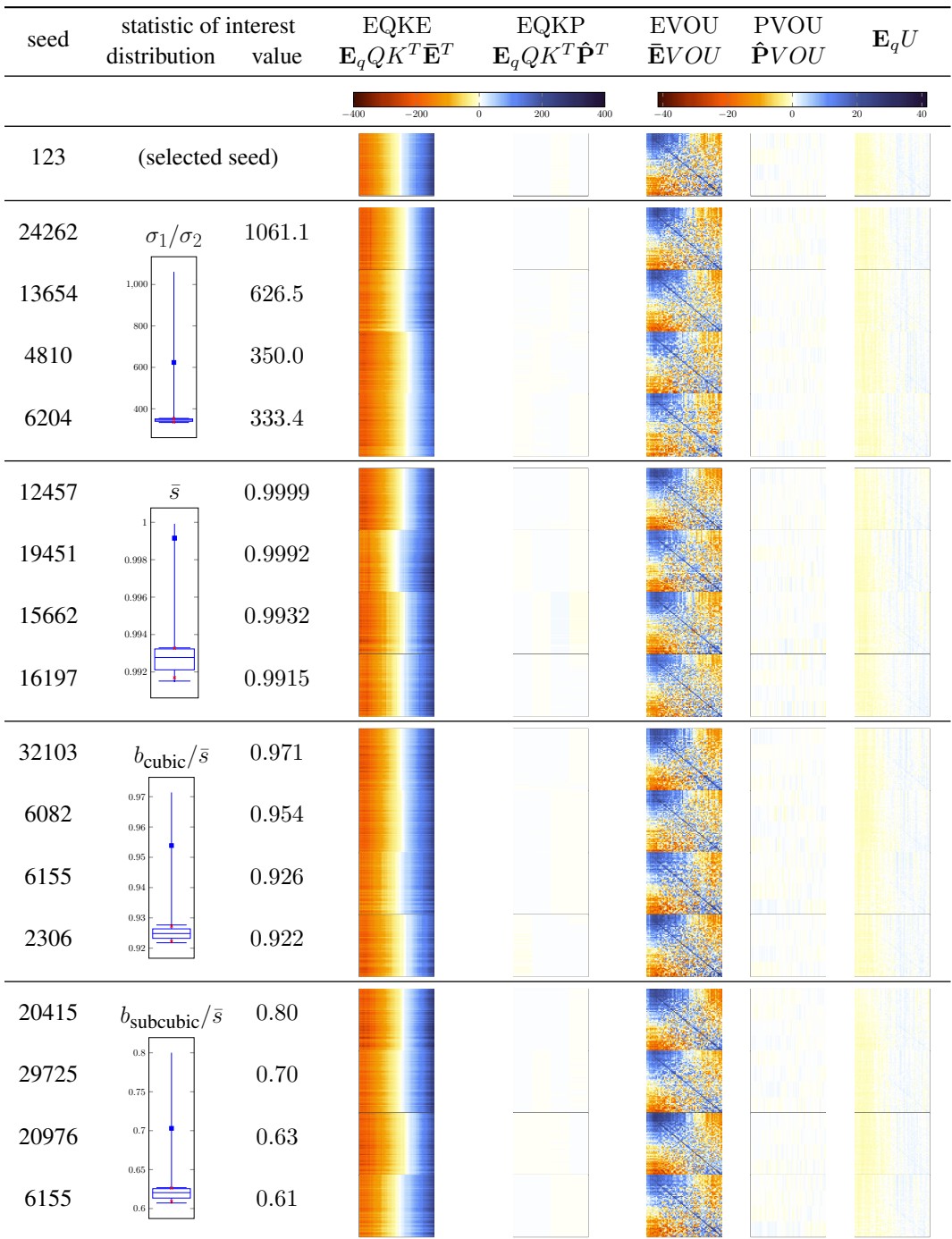

# C   Mathematical definitions

We provide a detailed breakdown of the mathematical notation used in the appendix.

Let

| | |
|---|---|
| $\underline{n}$ | be the finite set on $n$ elements; we write |
| $\mathbb{N}_{<n}$ | $:= \{0, 1, \ldots, n-1\}$ when we care about the elements of $\underline{n}$ |
| $\sigma(\mathbf{v})$ | be the softmax function $e^{\mathbf{v}}/\sum_i e^{v_i}$ |
| $\sigma^*(\mathbf{v})$ | be the casually-masked softmax function, $\sigma^*(\mathbf{v})_i := e^{v_i}/\sum_{j \le i} e^{v_j}$ |
| $d_{\text{vocab}}$ | be the size of the vocabulary |
| $d$ | be the dimension of the attention head, in our case, equal to the hidden dimension of the model $d_{\text{model}}$ (assumption: $d < d_{\text{vocab}}$) |
| $n_{\text{ctx}}$ | be the context length, the number of tokens in the input sequence, equal to $K$ in Max-of-$K$ |
| $P$ | be the $n_{\text{ctx}} \times d_{\text{model}}$ positional embedding |
| $E$ | be the $d_{\text{vocab}} \times d_{\text{model}}$ token embed |
| $Q, K,$ $V, O$ | be the $d_{\text{model}} \times d_{\text{model}}$ query, key, value, and output matrices of the attention head |
| $U$ | be the $d_{\text{model}} \times d_{\text{vocab}}$ unembed matrix |
| $\mathbf{t}$ | be the input token sequence $[t_0, t_1, \ldots, t_{n_{\text{ctx}}-1}]$ |
| $\mathbf{x}$ | be the $n_{\text{ctx}} \times d_{\text{vocab}}$ one-hot-encoded input token sequence $[x_0, x_1, \ldots, x_{n_{\text{ctx}}-1}]$ |
| $x_{\text{query}}$ | $:= x_{-1} := x_{n_{\text{ctx}}-1}$ be the query token |
| $t_{\text{query}}$ | $:= t_{-1} := t_{n_{\text{ctx}}-1}$ be the one-hot encoded query token |
| $t_{\max}$ | be the true maximum token in the input sequence, $\max_i t_i$ |

| | |
|---|---|
| $\mathcal{M}$ | of type $X \to Y$ be the model[18]; sometimes we write |
| $\ell$ | for the logits of the model $\mathcal{M}(\mathbf{t})$ |
| $\mathcal{D}$ | be a probability distribution over input-label pairs $(\mathbf{t}, l) \in X \times L$ |
| $\mathcal{D}\vert_X$ | be $\mathcal{D}$ marginalized to a distribution over $\mathbf{t} \in X$ |
| $f$ | of type $L \times Y \to \mathbb{R}$ be a scoring function for evaluating the performance of the model |
| $P_{\text{avg}}$ | be the average position embeds across positions (of size $d_{\text{model}}$), $\frac{1}{n_{\text{ctx}}} \sum_i P_i$ |
| $\bar{\mathbf{P}}$ | be either $\mathbf{1}_{n_{\text{ctx}}} \otimes P_{\text{avg}}$ or $\mathbf{1}_{d_{\text{vocab}}} \otimes P_{\text{avg}}$ depending on context – that is the result of broadcasting $P_{\text{avg}}$ back into the shape of $P$ or $E$ (that is, $n_{\text{ctx}} \times d_{\text{model}}$ or $d_{\text{vocab}} \times d_{\text{model}}$) |
| $\mathbf{P}_q$ | be $\mathbf{1}_{d_{\text{vocab}}} \otimes P_{\text{query}}$, the broadcasting of $P_{\text{query}}$ |
| $\hat{\mathbf{P}}, \bar{\mathbf{E}}, \mathbf{E}_q$ | be $P - \bar{\mathbf{P}}$, $E + \bar{\mathbf{P}}$, and $E + \mathbf{P}_q$ respectively |

For any vector-valued function $\mathbf{v}$ of length $d_{\text{vocab}}$, parameterized over the input sequence $\mathbf{t}$, let

$\Delta v_i := v_i - v_{t_{\max}}$ be the difference between the $i^{\text{th}}$ element of $\mathbf{v}$ and the element of $\mathbf{v}$ corresponding to the true maximum of the input sequence $t_{\max}$.

For our particular model, we will have

$$d_{\text{vocab}} := 64 \qquad d = d_{\text{model}} := 32 \qquad n_{\text{ctx}} := 4$$
$$X := (\mathbb{N}_{<d_{\text{vocab}}})^{n_{\text{ctx}}} \cong \underline{64}^4 \qquad L := \mathbb{N}_{<d_{\text{vocab}}} \cong \underline{64}$$
$$\mathcal{D}\vert_X := U(0, 1, \ldots, d_{\text{vocab}} - 1)^{n_{\text{ctx}}}, \text{ the uniform distribution}$$

**Figure 11:** Preliminary model definitions

$$\underbrace{\mathbb{E}_{(l,\mathbf{t}) \sim \mathcal{D}}\left[f(l, \mathcal{M}(\mathbf{t}))\right]}_{\bar{s} \text{ (average model score)}} \ge \underbrace{b}_{\text{lower bound}} \qquad \text{(theorem statement)} \tag{3}$$

We define the two typical performance functions corresponding to accuracy and log-loss. Note the shared subterm $\Delta \ell_i$.

$$f^{\text{accuracy}}(t_{\max}, \ell) := \mathbb{1}[\operatorname*{argmax}_i \ell_i = t_{\max}] = \mathbb{1}[0 > \max_{i \neq t_{\max}} \underbrace{\ell_i - \ell_{t_{\max}}}_{\Delta \ell_i}] \tag{4}$$

$$f^{\text{log-loss}}(t_{\max}, \ell) := (\sigma(\ell))_{t_{\max}} = \log(\sum_i \exp(\underbrace{\ell_i - \ell_{t_{\max}}}_{\Delta \ell_i})) \tag{5}$$

We present the model definition in four different regroupings to define via underbrace labels various useful quantities:

$$\mathcal{M}(\mathbf{t}) = \ell(\mathbf{t}) = \sigma^*\Big(\underbrace{(x_{\text{query}}E + P_{\text{query}})\,QK^T\,(\mathbf{x}E+P)^T/\sqrt{d}}_{\text{QK circuit}}\Big) \cdot \underbrace{(\mathbf{x}E+P)\,VOU}_{\text{OV circuit}} + \underbrace{(x_{\text{query}}E+P_{\text{query}})\,U}_{\text{direct path}} \tag{6}$$

$$= \sigma^*\Big(x_{\text{query}}\big(\underbrace{\mathbf{E}_q QK^T \bar{\mathbf{E}}^T}_{\text{EQKE}}\,\mathbf{x}^T + \underbrace{\mathbf{E}_q QK^T \hat{\mathbf{P}}^T}_{\text{EQKP}}\big)/\sqrt{d}\Big) \cdot \underbrace{\Big(\mathbf{x}\underbrace{\bar{\mathbf{E}}VOU}_{\text{EVOU}} + \underbrace{\hat{\mathbf{P}}VOU}_{\text{PVOU}}\Big)}_{\text{EPVOU}(\mathbf{t})} + x_{\text{query}}\underbrace{\mathbf{E}_q U}_{\text{EU}} \tag{7}$$

$$\underbrace{\phantom{= \sigma^*\Big(x_{\text{query}}\big(\mathbf{E}_q QK^T \bar{\mathbf{E}}^T\big)\Big)}}_{\alpha^*(\mathbf{t})}$$

$$= \underbrace{\alpha^*(\mathbf{t}) \cdot \mathbf{x}\bar{\mathbf{E}}VOU}_{\ell^{\text{EVOU}}(\mathbf{t})} + \underbrace{\alpha^*(\mathbf{t}) \cdot \hat{\mathbf{P}}VOU}_{\ell^{\text{PVOU}}(\mathbf{t})} + \underbrace{x_{\text{query}}\mathbf{E}_q U}_{\ell^{\text{EU}}(x_{\text{query}})} \tag{8}$$

$$= \underbrace{\Big(\sum_{i=0}^{n_{\text{ctx}}-1} \underbrace{(\alpha^*(\mathbf{t}))_i x_i \bar{\mathbf{E}}VOU}_{\ell^{\text{EVOU},i}(\mathbf{t})} + \underbrace{(\alpha^*(\mathbf{t}))_i \hat{\mathbf{P}}_i VOU}_{\ell^{\text{PVOU},i}(\mathbf{t})}\Big)}_{\ell^i(\mathbf{t})} + \underbrace{x_{\text{query}}\mathbf{E}_q U}_{\ell^{\text{EU}}(x_{\text{query}})} \tag{9}$$

**Figure 12:** Definitions of the model behavior

# D  Brute-force proof

**Theorem 1.** *For* BRUTE-FORCE$(d_{\text{vocab}}, n_{\text{ctx}}, \mathcal{M})$ *as defined in Algorithm 1,*

$$\mathbb{E}_{\mathbf{t} \sim U(0,1,\ldots,d_{\text{vocab}}-1)^{n_{\text{ctx}}}} \left[ \operatorname*{argmax}_i (\mathcal{M}(\mathbf{t}))_i = \max_i t_i \right] \geq \text{BRUTE-FORCE}(d_{\text{vocab}}, n_{\text{ctx}}, \mathcal{M})$$

*Proof.* In fact the two sides of the inequality are equal by definition. Hence the inequality follows by reflexivity of $\geq$. □

---

**Algorithm 1** Counting Correct Sequences By Brute Force

---

1: **function** CORRECTNESS($\mathcal{M}$, input-sequence)
2:     **return** MODEL-BEHAVIOR($\mathcal{M}$, input-sequence) == MAX(input-sequence)
3: **end function**
4: **function** BRUTE-FORCE($d_{\text{vocab}}$, $n_{\text{ctx}}$, $\mathcal{M}$)
5:     **return** $\frac{1}{d_{\text{vocab}}^{n_{\text{ctx}}}}$ SUM(CORRECTNESS($\mathcal{M}$, tokens) **for** tokens $\in$ (RANGE$(d_{\text{vocab}})$)$^{n_{\text{ctx}}}$)
6: **end function**

---

# E  Details of cubic proof

In this section, we prove formally the result used in Section 4.2, A cubic proof.

At its heart, the convexity of softmax[19] is an extension to a simple idea: a weighted average of scalar values is extremized by putting 100% of the weight on an extremal value.

Using this simple version of the theorem, however, gives a useless bound of 0% accuracy: if we pay no attention to the maximum of the sequence, of course we're going to get the wrong answer. Since in fact the space of possible weightings we may see in practice is much smaller (finite, in fact, with at most $d_{\text{vocab}}^{n_{\text{ctx}}}$ values), we may look for a more general version of this idea that gives us tighter bounds that still cover the space of possible weightings.

The weights are *not* linearly independently choosable (softmax is non-linear), so extremal values do not necessarily result from putting maximal attention on the worst token. It may be, when trying to find the worst case, that some positions are so dis-preferred that it makes more sense to choose a token that is "less bad" for those positions, if it draws enough attention away from the correct token. See Lemma 3 for details.

We thus spend this section characterizing a relaxation of the constraints on weights:

1. that contains all actually possible weightings,

2. that is extremized at weights that still correspond to some notion of "put the most weight on the extremal tokens", and

3. for which computing the extremal weightings is computationally efficient.

Before diving in, let's recall the proof that a weighted average of scalar values is extremized by putting 100% of the weight on extremal values:

**Theorem 2** (Warmup: Extremizing weighted averages). *Fix a set of values $v_i \in \mathbb{R}$. The weighted average is bounded by the extremal values: for any $w_i$ such that $\sum_i w_i = 1$ and $0 \leq w_i \leq 1$,*

$$\min_i v_i \leq \sum_i w_i v_i \leq \max_i v_i$$

---

[18] Note that while in the main body, $\mathcal{M}(\mathbf{t})$ referred to the pre-softmax output logits, in the appendix we abuse notation and occasionally use it to refer to maximum token indicated by the logits where appropriate.

[19] See Appendix A.4 for the reason that we call this "convexity". Note that our use of "convexity" is purely descriptive in this section; all theorems are written out explicitly.

*Proof.* The proof is simple. We have

$$\sum_i w_i v_i - \min_i v_i = \sum_i w_i(v_i - \min_j v_j) \geq 0$$

and

$$\max_i v_i - \sum_i w_i v_i = \sum_i w_i(\max_j v_j - v_i) \geq 0$$

so the result follows. $\square$

## E.1 Proof strategy

The model computes the true maximum $t_{\max}$ when its outputs logits $\ell$ are such that $\Delta\ell_{t^*} := \ell_{t^*} - \ell_{t_{\max}} < 0$ for all $t^* \neq t_{\max}$.[20] As a result, it suffices to lower-bound the proportion of sequences where (an upper bound on) $\Delta\ell_{t^*}$ is negative for all $t^* \neq t_{\max}$. In particular, we will upper-bound the contribution from incorrect tokens $t$ in positions $i$ to the difference $\Delta\ell^i$ between incorrect ($t^*$) and correct ($t_{\max}$) output tokens $\Delta\ell^i_{t^*} = \ell^i_{t^*} - \ell^i_{t_{\max}}$.

We do this by arguing that the logit difference $\Delta\ell_{t^*}$ satisfies a certain notion of convexity over the space of a relaxation of sequences (Theorem 6), and constructing a set of $\Theta(d_{\text{vocab}}{}^3 n_{\text{ctx}})$ "extremal" relaxed sequences where the position and token embedding components of attention are pessimized independently.

We start by first rewriting the contribution of each token through the attention head to the logit difference into the contributions involving PVOU and EVOU:

$$\Delta\ell^t_{t^*}(\mathbf{t}) = \Delta\ell^{\text{PVOU},i}{}_{t^*}(\mathbf{t}) + \Delta\ell^{\text{EVOU},i}{}_{t^*}(\mathbf{t})$$

We then upper bound $\Delta\ell^{\text{PVOU},i}{}_{t^*}(\mathbf{t})$ by noting that because the softmax attention is a weighted average of PVOU,

$$\begin{aligned}
\Delta\ell^{\text{PVOU},i}{}_{t^*}(\mathbf{t}) &= \ell^{\text{PVOU},i}(\mathbf{t})_{t^*} - \ell^{\text{PVOU},i}(\mathbf{t})_{\max_j t_j} \\
&= \alpha_i^*(\mathbf{t})\text{PVOU}_{i,t^*} - \alpha_i^*(\mathbf{t})\text{PVOU}_{i,\max_j t_j} \\
&= \alpha_i^*(\mathbf{t})\left(\text{PVOU}_{i,t^*} - \text{PVOU}_{i,\max_j t_j}\right) \\
&\leq \alpha_i^*(\mathbf{t})\max_i\left(\text{PVOU}_{i,t^*} - \text{PVOU}_{i,\max_j t_j}\right)
\end{aligned}$$

Since $\sum_i \alpha_i^*(\mathbf{t}) = 1$, we have

$$\sum_{i=0}^{n_{\text{ctx}}-1} \Delta\ell^{\text{PVOU},i}{}_{t^*}(\mathbf{t}) \leq \max_i\left(\text{PVOU}_{i,t^*} - \text{PVOU}_{i,\max_j t_j}\right)$$

We then construct a set $\Xi^{\text{pure}}$ of "pure sequences" consisting of only three types of tokens in one of two orders, and show that for each input sequence $\mathbf{t}$ and readoff logit $t^*$, we bound the logit difference from the token embeddings $\Delta\ell^{\text{EVOU},i}{}_{t^*}(\mathbf{t})$ using a small subset $\mathcal{X}$ of $\Xi^{\text{pure}}$:

$$\sum_{i=0}^{n_{\text{ctx}}-1} \Delta\ell^{\text{EVOU},i}{}_{t^*}(\mathbf{t}) \leq \max_{\xi\in\mathcal{X}} \sum_{i=0}^{n_{\text{ctx}}-1} \Delta\ell^{\text{EVOU},i}{}_{t^*}(\xi)$$

We construct a set $X^{\text{relaxed}}$ of relaxed sequences, where each relaxed sequence $\mathbf{t}^{\text{relaxed}}$ consists of a sequence and a position $(\mathbf{t}, i)$, where $\Delta\ell_{t^*}(\mathbf{t}, i)$ is evaluated by separately considering the positional

---

[20]We use the logit difference $\Delta\ell_{t^*}$ because: (a) it is shared in the computation of $f^{0\text{-}1}$ and $f^{\text{log-loss}}$; (b) it is a linear function of the various paths through the model, which can therefore be analyzed separately; (c) it leaves open both the options of pessimizing over output logit before or after combining contributions of various paths through the model.

contribution through attention (that is, the attention weighted PVOU) and the token contribution (that is, the attention-weighted EVOU) and direct contribution (the logit difference through the skip connection EU). Note that $i$ indicates the position that we pay 100% of the attention to for the PVOU contribution.

We argue that $\Delta \ell_{t^*}(\mathbf{t}, i)$ satisfies a certain notion of convexity over mixtures of sequences, such that we can evaluate it only on a set of $\Theta(d_{\text{vocab}}{}^3 n_{\text{ctx}})$ "extremal" sequences in a way that takes $O(d_{\text{vocab}}{}^3 n_{\text{ctx}})$ total time to bound $\Delta \ell_{t^*}(\mathbf{t}, i)$ for *every* possible input sequence. We then use the extremal sequences that the model gets correct to lower bound the proportion of *all* sequences that the model will get correct. Specifically, we argue that Algorithm 3 provides a valid lower bound on the proportion of sequences the model gets correct.

## E.2 Proof outline

We now proceed to the main results of this section.

**Math fact:** For each token $t^*$, the logit difference $\Delta \ell_{t^*}$ for any sequence $\mathbf{t}$ can be decomposed into the direct contribution from the embeds $\ell^{\text{EU}}$, the attention-weighted position contribution (PVOU), and the attention-weighted token contribution (EVOU). Therefore, it suffices to upper bound each of the three components independently, since summing these upper bounds gives a valid upper bound on the logit difference.

We can compute the direct contribution $\ell^{\text{EU}}$ exactly by first computing $\text{EU} = \mathbf{E}_q U$ and then, for each max, subtracting the logit of the max token from each row of the matrix. No theorems needed. For each max token, we can bound the position contribution by its maximum over positions (Theorem 6).

In order to upper bound the token contribution, we argue that any mixed sequence will be upper bounded by the maximum of the corresponding pure sequences (Theorem 7). We then argue that for pure sequences, it suffices to consider orderings where same tokens appear contiguously (Theorem 4).

## E.3 Formal proof

For this subsection, all theorems are parameterized over the following quantities.

**Definition 1** (Common theorem parameters). *Fix a token value function (à la a row difference in* EVOU*)* $v : \mathbb{N}_{<d_{\text{vocab}}} \to \mathbb{R}$ *and a token attention function (à la* EQKE *for a fixed query token)* $a : \mathbb{N}_{<d_{\text{vocab}}} \to \mathbb{R}$. *Fix a position value function (à la a row difference in* PVOU*)* $w : \mathbb{N}_{<n_{\text{ctx}}} \to \mathbb{R}$ *and a position attention function (à la* EQKP *for a fixed query token)* $b : \mathbb{N}_{<n_{\text{ctx}}} \to \mathbb{R}$.

In practice, we'll take, for fixed query token $t_{\text{query}}$, fixed output token of interest $t^*$, and fixed maximum token $t_{\text{max}}$,

$$v_t = \text{EVOU}_{t,t^*} - \text{EVOU}_{t,t_{\text{max}}} \qquad\qquad a_t = \text{EQKE}_{t_{\text{query}},t}/\sqrt{d}$$

$$w_i = \text{PVOU}_{i,t^*} - \text{PVOU}_{i,t_{\text{max}}} \qquad\qquad b_i = \text{EQKP}_{t_{\text{query}},i}/\sqrt{d}$$

**Definition 2** (of a sequence via sorted tokens and a position permutation). *We can* define *a sequence of tokens via sorted tokens and a position permutation by specifying a non-decreasing sequence of tokens* $t_0 \leq \cdots \leq t_{n_{\text{ctx}}-1} \in \mathbb{N}_{<d_{\text{vocab}}}$ *paired with a permutation* $\sigma : \mathbb{N}_{<n_{\text{ctx}}} \to \mathbb{N}_{<n_{\text{ctx}}}$.

**Definition 3** (sequence score). *Given a non-decreasing sequence of tokens* $t_0 \leq \cdots \leq t_{n_{\text{ctx}}-1} \in \mathbb{N}_{<d_{\text{vocab}}}$ *and a permutation* $\sigma : \mathbb{N}_{<n_{\text{ctx}}} \to \mathbb{N}_{<n_{\text{ctx}}}$ *define the* sequence score $s_{t_0,\dots,t_{n_{\text{ctx}}-1},\sigma}$ *as:*

$$s_{t_0,\dots,t_{n_{\text{ctx}}-1},\sigma} := \sum_{0 \leq i < n_{\text{ctx}}} v_{t_i} e^{a_{t_i}+b_{\sigma(i)}} \Bigg/ \sum_{0 \leq i < n_{\text{ctx}}} e^{a_{t_i}+b_{\sigma(i)}}$$

*We will drop the token subscript, writing only* $s_\sigma$, *when the token values are unambiguous by context.*

The sequence score here will be computing $\Delta \ell^{\text{EVOU}}{}_{t^*}$ for some fixed $t^*$ and $t_{\text{max}}$. The way we've set up our definitions, high scores predict $t^*$ (and are thus bad), negative scores predict $t_{\text{max}}$ (and are thus good), and more negative the scores, the stronger the prediction of $t_{\text{max}}$.

**Definition 4** (swap permutation). *Given a permutation* $\sigma : \mathbb{N}_{<n_{\text{ctx}}} \to \mathbb{N}_{<n_{\text{ctx}}}$ *of the* $n_{\text{ctx}}$ *positions and two indices* $0 \leq i, j < n_{\text{ctx}}$, *define the* swap permutation $\sigma_{i \leftrightarrow j}$ *to be the permutation that is* $\sigma$

*except swapping $i$ and $j$:*

$$\sigma_{i\leftrightarrow j}(k) = \begin{cases} \sigma(i) & \textit{if } k = j \\ \sigma(j) & \textit{if } k = i \\ \sigma(k) & \textit{otherwise} \end{cases}$$

*Define $\Delta_{\sigma,i\leftrightarrow j}$ to be the* difference in sequence scores when you swap $i$ and $j$:

$$\Delta_{\sigma,i\leftrightarrow j} := s_{\sigma_{i\leftrightarrow j}} - s_\sigma$$

**Lemma 3** (Characterization of swapping tokens). *Fix a non-decreasing sequence of tokens $t_0 \leq \cdots \leq t_{n_{\mathrm{ctx}}-1} \in \mathbb{N}$. Fix $\sigma : \mathbb{N} \to \mathbb{N}$ be a permutation of the $n_{\mathrm{ctx}}$ positions. Fix indices $0 \leq i, j < n_{\mathrm{ctx}}$.*

*Then there are two cases for $\mathrm{sign}\,(\Delta_{\sigma,i\leftrightarrow j})$:*

1. *If $a_{t_i} = a_{t_j}$ then $\mathrm{sign}\,(\Delta_{\sigma,i\leftrightarrow j}) = -\,\mathrm{sign}\,\big(b_{\sigma(i)} - b_{\sigma(j)}\big)\,\mathrm{sign}\,\big(v_{t_i} - v_{t_j}\big)$.*

2. *Otherwise, $\mathrm{sign}\,(\Delta_{\sigma,i\leftrightarrow j}) = \mathrm{sign}\,\big(a_{t_i} - a_{t_j}\big)\,\mathrm{sign}\,\big(b_{\sigma(i)} - b_{\sigma(j)}\big)\,\mathrm{sign}\,\left( s_\sigma - \dfrac{v_{t_i} e^{a_{t_i}} - v_{t_j} e^{a_{t_j}}}{e^{a_{t_i}} - e^{a_{t_j}}} \right)$.*

Intuitively, Lemma 3 says that, if the token contribution to attention is equal between tokens $t_i$ and $t_j$, then the impact of swapping their positions $\sigma(i)$ and $\sigma(j)$ is entirely determined by how much attention is paid to the positions of $i$ and $j$ and the relative difference in their value. (Notably, by swapping these tokens, we don't affect the attention paid on other tokens, and so the effect of the change does not depend on the values of the other tokens.) Alternatively, if the attentions are not equal, then swapping the positions changes the allocation of attention to other tokens in the sequence, and so it may the case that this change in allocation in attention dominates the attention-weighted values of these two tokens.

*Proof.* First note that the theorem is trivial for $i = j$.

For the rest of the proof, we take $i \neq j$.

The proof proceeds just by algebraic manipulation with no deep insight. We first list the facts we use, the proceed to computing $\mathrm{sign}\,(\Delta_{\sigma,i\leftrightarrow j})$. We abbreviate $\sigma_{i\leftrightarrow j}$ as $\sigma'$ for brevity.

$$\mathrm{sign}\,\big(e^{b_{\sigma(i)}} - e^{b_{\sigma(j)}}\big) = \mathrm{sign}\,\big(b_{\sigma(i)} - b_{\sigma(j)}\big)$$

$$\mathrm{sign}\,(\Delta_{\sigma,i\leftrightarrow j}) = \mathrm{sign}\,(s_{\sigma'} - s_\sigma)$$

$$= \mathrm{sign}\left( \frac{\sum_{0 \leq p < n_{\mathrm{ctx}}} v_{t_p} e^{a_{t_p} + b_{\sigma'(p)}}}{\sum_{0 \leq p < n_{\mathrm{ctx}}} e^{a_{t_p} + b_{\sigma'(p)}}} - s_\sigma \right)$$

Now multiply through by the denominator, which is positive

$$= \mathrm{sign}\left( \sum_{0 \leq p < n_{\mathrm{ctx}}} v_{t_p} e^{a_{t_p} + b_{\sigma'(p)}} - s_\sigma \sum_{0 \leq p < n_{\mathrm{ctx}}} e^{a_{t_p} + b_{\sigma'(p)}} \right)$$

$$= \mathrm{sign}\left( \sum_{0 \leq p < n_{\mathrm{ctx}}} v_{t_p} e^{a_{t_p} + b_{\sigma(p)}} - v_{t_i} e^{a_{t_i}} \big( e^{b_{\sigma(i)}} - e^{b_{\sigma'(i)}} \big) - v_{t_j} e^{a_{t_j}} \big( e^{b_{\sigma(j)}} - e^{b_{\sigma'(j)}} \big) \right.$$

$$\left. - s_\sigma \sum_{0 \leq p < n_{\mathrm{ctx}}} e^{a_{t_p} + b_{\sigma(p)}} + s_\sigma e^{a_{t_i}} \big( e^{b_{\sigma(i)}} - e^{b_{\sigma'(i)}} \big) + s_\sigma e^{a_{t_j}} \big( e^{b_{\sigma(j)}} - e^{b_{\sigma'(j)}} \big) \right)$$

$$= \mathrm{sign}\left( \cancel{\sum_{0 \leq p < n_{\mathrm{ctx}}} v_{t_p} e^{a_{t_p} + b_{\sigma(p)}}} - v_{t_i} e^{a_{t_i}} \big( e^{b_{\sigma(i)}} - e^{b_{\sigma(j)}} \big) - v_{t_j} e^{a_{t_j}} \big( e^{b_{\sigma(j)}} - e^{b_{\sigma(i)}} \big) \right.$$

$$\left. - \cancel{\sum_{0 \leq p < n_{\mathrm{ctx}}} v_{t_p} e^{a_{t_p} + b_{\sigma(p)}}} + s_\sigma e^{a_{t_i}} \big( e^{b_{\sigma(i)}} - e^{b_{\sigma(j)}} \big) + s_\sigma e^{a_{t_j}} \big( e^{b_{\sigma(j)}} - e^{b_{\sigma(i)}} \big) \right)$$

$$= \text{sign}\left(\left(v_{t_j}e^{a_{t_j}} - v_{t_i}e^{a_{t_i}}\right)\left(e^{b_{\sigma(i)}} - e^{b_{\sigma(j)}}\right) + s_\sigma\left(e^{a_{t_i}} - e^{a_{t_j}}\right)\left(e^{b_{\sigma(i)}} - e^{b_{\sigma(j)}}\right)\right)$$

$$= \text{sign}\left(e^{b_{\sigma(i)}} - e^{b_{\sigma(j)}}\right)\text{sign}\left(\left(v_{t_j}e^{a_{t_j}} - v_{t_i}e^{a_{t_i}}\right) + s_\sigma\left(e^{a_{t_i}} - e^{a_{t_j}}\right)\right)$$

$$= \text{sign}\left(b_{\sigma(i)} - b_{\sigma(j)}\right)\text{sign}\left(s_\sigma\left(e^{a_{t_i}} - e^{a_{t_j}}\right) - \left(v_{t_i}e^{a_{t_i}} - v_{t_j}e^{a_{t_j}}\right)\right)$$

Divide through by non-zero values when possible

$$= \text{sign}\left(b_{\sigma(i)} - b_{\sigma(j)}\right)$$

$$\cdot \begin{cases} \text{sign}\left(v_{t_i} - v_{t_j}\right) & \text{if } a_{t_i} = a_{t_j} \\ \text{sign}\left(e^{a_{t_i}} - e^{a_{t_j}}\right)\text{sign}\left(s_\sigma - \frac{v_{t_i}e^{a_{t_i}} - v_{t_j}e^{a_{t_j}}}{e^{a_{t_i}} - e^{a_{t_j}}}\right) & \text{otherwise} \end{cases}$$

$$= \begin{cases} -\text{sign}\left(b_{\sigma(i)} - b_{\sigma(j)}\right)\text{sign}\left(v_{t_i} - v_{t_j}\right) & \text{if } a_{t_i} = a_{t_j} \\ \text{sign}\left(a_{t_i} - a_{t_j}\right)\text{sign}\left(b_{\sigma(i)} - b_{\sigma(j)}\right)\text{sign}\left(s_\sigma - \frac{v_{t_i}e^{a_{t_i}} - v_{t_j}e^{a_{t_j}}}{e^{a_{t_i}} - e^{a_{t_j}}}\right) & \text{otherwise} \end{cases}$$

$\square$

**Definition 5** ($\sigma$ fixes $F$). *Fix a set of fixed indices $F \subseteq \mathbb{N}_{<n_{\text{ctx}}}$ and an assignment of token values to each of the fixed positions $t_F : F \to \mathbb{N}_{<d_{\text{vocab}}}$. (F is the set of positions for which we are not pessimizing over the value of the token in that position.) Fix a non-decreasing sequence of tokens $t_0 \leq \cdots \leq t_{n_{\text{ctx}}-1} \in \mathbb{N}$.*

*Given a permutation $\sigma : \mathbb{N}_{<n_{\text{ctx}}} \to \mathbb{N}^{n_{\text{ctx}}}$, say that $\sigma$ fixes $F$ (relative to $t_0, \ldots, t_{n_{\text{ctx}}-1}$) if $t_i = t_F(\sigma(i))$ whenever $\sigma(i) \in F$.*

Note that in this section, for the cubic proofs, we will in fact generally take $F = \{n_{\text{ctx}} - 1\}$, so that we are fixing the final query token, though in Theorems 7, 8, and 9 $F$ will also contain all positions with the maximum token $t_{\max}$. In Appendix F, we will take $F = \emptyset$ or $F$ to be the set of positions of the maximum token. However, none of these theorems are specific to $F$ being subsingleton, and we prove them in generality.

**Definition 6** (position-sorting permutation). *Fix a set of fixed indices $F \subseteq \mathbb{N}_{<n_{\text{ctx}}}$ and an assignment of token values to each of the fixed positions $t_F : F \to \mathbb{N}_{<d_{\text{vocab}}}$.*

*Define the* position-sorting permutation fixing indices in $F$ $\sigma_s : \mathbb{N}_{<n_{\text{ctx}}} \to \mathbb{N}_{<n_{\text{ctx}}}$ *to be the permutation that sorts the indices not in $F$ according to $b$: for $0 \leq i, j < n_{\text{ctx}}$ with $i, j \notin F$, $b_i \leq b_j$ whenever $\sigma_s(i) < \sigma_s(j)$; and $\sigma_s(i) = i$ for $i \in F$.*

**Definition 7** (contiguous on equal tokens). *Fix a set of fixed indices $F \subseteq \mathbb{N}_{<n_{\text{ctx}}}$ and an assignment of token values to each of the fixed positions $t_F : F \to \mathbb{N}_{<d_{\text{vocab}}}$. Fix a non-decreasing sequence of tokens $t_0 \leq \cdots \leq t_{n_{\text{ctx}}-1} \in \mathbb{N}$.*

*Say that the sequence represented by a permutation $\sigma : \mathbb{N}_{<n_{\text{ctx}}} \to \mathbb{N}_{<n_{\text{ctx}}}$ is* contiguous on equal tokens *if, for all $0 \leq i, j, k < n_{\text{ctx}}$ with $t_i = t_j \neq t_k$ and $i, j, k \notin \sigma^{-1}(F)$, it is never the case that $\sigma_s(\sigma(i)) < \sigma_s(\sigma(k)) < \sigma_s(\sigma(j))$.*

**Theorem 4** (Pessimization over sequence ordering is possible and results in contiguous sequences). *Fix a set of fixed indices $F \subseteq \mathbb{N}_{<n_{\text{ctx}}}$ and an assignment of token values to each of the fixed positions $t_F : F \to \mathbb{N}_{<d_{\text{vocab}}}$. Fix a non-decreasing sequence of tokens $t_0 \leq \cdots \leq t_{n_{\text{ctx}}-1} \in \mathbb{N}$.*

*Let $\sigma_{\min}, \sigma_{\max} : \mathbb{N} \to \mathbb{N}$ be permutations of the $n_{\text{ctx}}$ positions, fixing positions in $F$, satisfying the following property: For all $\sigma : \mathbb{N} \to \mathbb{N}$ a permutation fixing $F$, we have*

$$s_{\sigma_{\min}} \leq s_\sigma \leq s_{\sigma_{\max}} \tag{10}$$

*(Such permutations are guaranteed to exist because the permutation group on $n_{\text{ctx}}$ elements is finite.)*

*Then $\sigma_{\max}$ and $\sigma_{\min}$ may be taken to be contiguous on equal tokens. That is, there exist $\sigma_{\max}$ and $\sigma_{\min}$ satisfying the property of Equation 10 which additionally satisfy the definition of Definition 7.*

The basic idea is that we will assume that one of $\sigma_{\max}$ and $\sigma_{\min}$ cannot be contiguous on equal tokens and derive a contradiction. We will pick the extremal permutation that is closest to being contiguous, take a contiguity violation, and then show that either we can correct the contiguity violation without changing the score—thus violating the presumption that the permutation is *closest*

to being contiguous—or we will find one swap of indices that decreases the score and another swap of indices that increases the score, thus violating the presumption of extremality.

In slightly more detail, but still informally, we will consider the sign of the difference between scores of our purported extremal permutation and a permutation that has swapped some indices. The theorem follows from showing that there exists a triple of indices $i, j, k$ such that the sign of the score difference from swapping $i$ and $j$ is different from the sign of the score difference from swapping $j$ and $k$.

First, a definition and some helpful facts about it.

**Definition 8** (contiguous on equally-attended positions). *Fix a set of fixed indices $F \subseteq \mathbb{N}_{<n_{\text{ctx}}}$ and an assignment of token values to each of the fixed positions $t_F : F \to \mathbb{N}_{<d_{\text{vocab}}}$. Fix a non-decreasing sequence of tokens $t_0 \leq \cdots \leq t_{n_{\text{ctx}}-1} \in \mathbb{N}$.*

*Say that a permutation $\sigma$ is* contiguous on equally-attended positions *if, for all $0 \leq i < n_{\text{ctx}}$ with $i \notin \sigma^{-1}(F)$, the sorting order according $\sigma_s$ on the contiguous block of positions with contribution to the attention score equal to that of $\sigma(i)$, $\{\sigma(j) \mid b_{\sigma(j)} = b_{\sigma(i)} \text{ and } \sigma(j) \notin F\}$, is the same as the sorting order according to the fraction of tokens equal to $t_j$ with $b$-values greater than $b_{\sigma(i)}$, with ties broken by the value of $t_j$. Equationally, this second sorting order is defined by the score*

$$\left( \left| \{k \mid t_k = t_j \text{ and } b_{\sigma(k)} > b_{\sigma(i)} \text{ and } \sigma(k) \notin F\} \right| + \frac{t_j}{d_{\text{vocab}}} \right) \Big/ \left| \{k \mid t_k = t_j \text{ and } \sigma(k) \notin F\} \right|.$$

Most importantly, any permutation that is contiguous on equally-attended positions has the property that for any indices $0 \leq i, j, k < n_{\text{ctx}}$ with $i, j, k \notin \sigma^{-1}(F)$ and $t_i = t_j \neq t_k$ and $\sigma_s(\sigma(i)) < \sigma_s(\sigma(k)) < \sigma_s(\sigma(j))$, we will have the *strict* inequality $b_{\sigma(i)} < b_{\sigma(k)} < b_{\sigma(j)}$. Additionally, we may always sort equally-attended positions to make any permutation contiguous on equally-attended positions.

We will define an additional notion of contiguity-violations which we avoid up-front by arbitrarily swapping involved indices without changing the score $s_\sigma$.

**Definition 9** (needlessly non-contiguous). *Fix a set of fixed indices $F \subseteq \mathbb{N}_{<n_{\text{ctx}}}$ and an assignment of token values to each of the fixed positions $t_F : F \to \mathbb{N}_{<d_{\text{vocab}}}$. Fix a non-decreasing sequence of tokens $t_0 \leq \cdots \leq t_{n_{\text{ctx}}-1} \in \mathbb{N}$.*

*Say that a permutation $\sigma$ is* needlessly non-contiguous at $i, j, k$ *(for $i, j, k \notin \sigma^{-1}(F)$) if $\Delta_{\sigma, i \leftrightarrow k} = 0$ or $\Delta_{\sigma, j \leftrightarrow k} = 0$, for $0 \leq i, j, k < n_{\text{ctx}}$ with $i, j, k \notin \sigma^{-1}(F)$ with $t_i = t_j \neq t_k$ and $\sigma_s(\sigma(i)) < \sigma_s(\sigma(k)) < \sigma_s(\sigma(j))$.*

*Say that a permutation $\sigma$ is* needlessly non-contiguous *if it is needlessly non-contiguous at any $i, j, k \notin \sigma^{-1}(F)$.*

**Lemma 5.** *Fix a set of fixed indices $F \subseteq \mathbb{N}_{<n_{\text{ctx}}}$ and an assignment of token values to each of the fixed positions $t_F : F \to \mathbb{N}_{<d_{\text{vocab}}}$. Fix a non-decreasing sequence of tokens $t_0 \leq \cdots \leq t_{n_{\text{ctx}}-1} \in \mathbb{N}$.*

*Any needlessly non-contiguous sequence $\sigma$ which fixes $F$ can be made into a sequence $\sigma'$ which still fixes $F$ and is both simultaneously contiguous on equally-attended positions and not needlessly non-contiguous, and for which $s_\sigma = s_{\sigma'}$.*

*Proof.* First, sort regions of equally-attended positions to make $\sigma$ contiguous on equally-attended positions. If the resulting permutation is not needlessly non-contiguous, then we are done.

Otherwise, we have $\Delta_{\sigma, i \leftrightarrow k} = 0$ or $\Delta_{\sigma, j \leftrightarrow k} = 0$ for some $i, j, k$, for $0 \leq i, j, k < n_{\text{ctx}}$ with $i, j, n \notin \sigma^{-1}(F)$ and $t_i = t_j \neq t_k$ and $\sigma_s(\sigma(i)) < \sigma_s(\sigma(k)) < \sigma_s(\sigma(j))$. Since the sequence is contiguous on equally-attended positions, we have the strict inequality $b_{\sigma(i)} < b_{\sigma(k)} < b_{\sigma(j)}$.

By Lemma 3, we have two cases. Noting that $t_i = t_j$, we can write them as

1. $v_{t_k} = v_{t_i}$ and $a_{t_i} = a_{t_k}$

2. $a_{t_i} \neq a_{t_k}$ and $s_\sigma = \frac{v_{t_i} e^{a_{t_i}} - v_{t_k} e^{a_{t_k}}}{e^{a_{t_i}} - e^{a_{t_k}}}$

In the first case, we may fully freely interchange tokens equal to $t_i$ with tokens equal to $t_k$ without changing the score; in this case we may use the token value as a sorting tie-breaker and swap tokens until there are no more needlessly non-contiguous triples falling into case (1).

In the second case, since swapping tokens does not change $s_\sigma$, the property will continue to hold for these tokens after the swap. We may then swap tokens, again using token value as a tie-breaker, until there are no more needlessly non-contiguous triples falling into case (2). $\qquad\square$

We can now finally make our argument for Theorem 4 more precise.

*Proof of Theorem 4.* Choose $\sigma_{\max}$ and $\sigma_{\min}$ to be contiguous on equally-attended positions and not needlessly non-contiguous, and suppose that we have $\sigma \in \{\sigma_{\max}, \sigma_{\min}\}$ such that for some $0 \le i, j, k < n_{\text{ctx}}$ with $i, j, k \notin \sigma^{-1}(F)$ and $t_i = t_j \ne t_k$, we have $b_{\sigma(i)} < b_{\sigma(k)} < b_{\sigma(j)}$. We will derive a contradiction with the presumption that $\sigma$ is extremal by showing that we can swap $i$ and $k$ to change the score in one direction and that we can swap $j$ and $k$ to change the score in the other direction.

Take $\sigma_0'$ to be $\sigma$ but swapping $i$ and $k$, and take $\sigma_1'$ to be $\sigma$ but swapping $j$ and $k$.

Now we will consider the cases for the sign of the score difference $\Delta_0 := s_{\sigma_0'} - s_\sigma$ and $\Delta_1 := s_{\sigma_1'} - s_\sigma$. By the presumption of not being needlessly non-contiguous, $\Delta_z \ne 0$ for $z \in \{0, 1\}$. If we can show that the sign of $\Delta_0$ is distinct from the sign of $\Delta_1$, then we will have a contradiction with extremality because we will have either $s_{\sigma_0'} < s_\sigma < s_{\sigma_1'}$ or $s_{\sigma_1'} < s_\sigma < s_{\sigma_0'}$. That is, we would be able to swap $i \leftrightarrow k$ and $j \leftrightarrow k$ to get a lower and higher score, making $\sigma$ not extremal.

Noting that $t_i = t_j$,

$$\text{sign}\left(\Delta_0\right) = \text{sign}\left(b_{\sigma(i)} - b_{\sigma(k)}\right) \begin{cases} \text{sign}\left(v_{t_k} - v_{t_i}\right) & \text{if } a_{t_i} = a_{t_k} \\ \text{sign}\left(a_{t_i} - a_{t_k}\right)\text{sign}\left(s_\sigma - \dfrac{v_{t_i}e^{a_{t_i}} - v_{t_k}e^{a_{t_k}}}{e^{a_{t_i}} - e^{a_{t_k}}}\right) & \text{otherwise} \end{cases}$$

$$\text{sign}\left(\Delta_1\right) = \text{sign}\left(b_{\sigma(j)} - b_{\sigma(k)}\right) \begin{cases} \text{sign}\left(v_{t_k} - v_{t_i}\right) & \text{if } a_{t_i} = a_{t_k} \\ \text{sign}\left(a_{t_i} - a_{t_k}\right)\text{sign}\left(s_\sigma - \dfrac{v_{t_i}e^{a_{t_i}} - v_{t_k}e^{a_{t_k}}}{e^{a_{t_i}} - e^{a_{t_k}}}\right) & \text{otherwise} \end{cases}$$

Noting that the product is non-zero by presumption, that right multiplicand is equal for $\Delta_0$ and $\Delta_1$, and $\text{sign}\left(b_{\sigma(i)} - b_{\sigma(k)}\right) = -1$ and $\text{sign}\left(b_{\sigma(j)} - b_{\sigma(k)}\right) = 1$, we have our desired contradiction. $\qquad\square$

Note that the proof of Theorem 4 does not go through if we include the position value function $w$ in the score, because we may trade off the position value function against the token value function. We now show that we can *independently* pessimize over positional attention.

**Definition 10** (full sequence score). *Given a non-decreasing sequence of tokens $t_0 \le \cdots \le t_{n_{\text{ctx}}-1} \in \mathbb{N}_{<d_{\text{vocab}}}$ and a permutation $\sigma : \mathbb{N}_{<n_{\text{ctx}}} \to \mathbb{N}_{<n_{\text{ctx}}}$ define the* full sequence score $s'_{t_0,\ldots,t_{n_{\text{ctx}}-1},\sigma}$ *as:*

$$s'_{t_0,\ldots,t_{n_{\text{ctx}}-1},\sigma} := \sum_{0 \le i < n_{\text{ctx}}}(v_{t_i} + w_{\sigma(i)})e^{a_{t_i} + b_{\sigma(i)}} \Bigg/ \sum_{0 \le i < n_{\text{ctx}}} e^{a_{t_i} + b_{\sigma(i)}}$$

*We will drop the token subscript, writing only $s'_\sigma$, when the token values are unambiguous by context.*

The sequence score here will be computing $\Delta\ell^{\text{EPVOU}}_{t^*} := \Delta\ell^{\text{EVOU}}{}_{t^*} + \Delta\ell^{\text{PVOU}}{}_{t^*}$ for some fixed $t^*$ and $t_{\max}$. As with Definition 3, with the way we've set up our definitions, high scores predict $t^*$ (and are thus bad), negative scores predict $t_{\max}$ (and are thus good), and more negative the scores, the stronger the prediction of $t_{\max}$.

**Definition 11** (relaxed sequence score). *Given a non-decreasing sequence of tokens $t_0 \le \cdots \le t_{n_{\text{ctx}}-1} \in \mathbb{N}_{<d_{\text{vocab}}}$ and a permutation $\sigma : \mathbb{N}_{<n_{\text{ctx}}} \to \mathbb{N}_{<n_{\text{ctx}}}$ define the* relaxed sequence scores $r_{t_0,\ldots,t_{n_{\text{ctx}}-1},\sigma,\min}$ *and* $r_{t_0,\ldots,t_{n_{\text{ctx}}-1},\sigma,\max}$ *as:*

$$r_{t_0,\ldots,t_{n_{\text{ctx}}-1},\sigma,\min} := s_{t_0,\ldots,t_{n_{\text{ctx}}-1},\sigma} + \min_{0 \le i < n_{\text{ctx}}} w_i$$

$$r_{t_0,\ldots,t_{n_{\text{ctx}}-1},\sigma,\max} := s_{t_0,\ldots,t_{n_{\text{ctx}}-1},\sigma} + \max_{0 \le i < n_{\text{ctx}}} w_i$$

*We will drop the token subscript, writing only $r_{\sigma,\min}$ or $r_{\sigma,\max}$, when the token values are unambiguous by context.*

**Theorem 6** (Independent pessimization over positional contributions is possible). *Fix non-decreasing sequences of tokens $t_0 \leq \cdots \leq t_{n_{\mathrm{ctx}}-1} \in \mathbb{N}$ and $t'_0 \leq \cdots \leq t'_{n_{\mathrm{ctx}}-1} \in \mathbb{N}$ and permutations $\sigma, \sigma' : \mathbb{N}_{<n_{\mathrm{ctx}}} \to \mathbb{N}_{<n_{\mathrm{ctx}}}$. Let $r_{\sigma,\min}$ and $r_{\sigma,\max}$ denote $r_{t_0,\ldots,t_{n_{\mathrm{ctx}}-1},\sigma,\min}$ and $r_{t_0,\ldots,t_{n_{\mathrm{ctx}}-1},\sigma,\max}$; let $s_\sigma$ denote $s_{t_0,\ldots,t_{n_{\mathrm{ctx}}-1},\sigma}$; and let $s_{\sigma'}$ and $s'_{\sigma'}$ denote $s_{t'_0,\ldots,t'_{n_{\mathrm{ctx}}-1},\sigma'}$ and $s'_{t'_0,\ldots,t'_{n_{\mathrm{ctx}}-1},\sigma'}$.*

*Then we have*

$$\min_{0\leq i<n_{\mathrm{ctx}}} w_i = r_{\sigma,\min} - s_\sigma \leq s'_{\sigma'} - s_{\sigma'} \leq r_{\sigma,\max} - s_\sigma = \max_{0\leq i<n_{\mathrm{ctx}}} w_i.$$

*That is, the difference between the relaxed sequence score and the sequence score of any given sequence always bounds the difference between the full sequence score and the sequence score for any (related or unrelated) sequence.*

*Proof.* This proof follows straightforwardly from the softmax weighting being an affine weighting.

We have

$$r_{\sigma,\min} - s_\sigma = \min_{0\leq i<n_{\mathrm{ctx}}} w_{\sigma(i)} = \min_{0\leq i<n_{\mathrm{ctx}}} w_i$$

$$r_{\sigma,\max} - s_\sigma = \max_{0\leq i<n_{\mathrm{ctx}}} w_{\sigma(i)} = \max_{0\leq i<n_{\mathrm{ctx}}} w_i$$

$$s'_{\sigma'} - s_{\sigma'} = \sum_{0\leq i<n_{\mathrm{ctx}}} w_{\sigma'(i)} e^{a_{t'_i}+b_{\sigma'(i)}} \bigg/ \sum_{0\leq i<n_{\mathrm{ctx}}} e^{a_{t'_i}+b_{\sigma'(i)}}$$

$$= \sum_{0\leq i<n_{\mathrm{ctx}}} w_{\sigma'(i)} \frac{e^{a_{t'_i}+b_{\sigma'(i)}}}{\sum_{0\leq j<n_{\mathrm{ctx}}} e^{a_{t'_j}+b_{\sigma'(j)}}}$$

Since $e^x$ is non-negative for all real $x$, we have

$$\min_{0\leq j<n_{\mathrm{ctx}}} w_{\sigma'(j)} \frac{\sum_{0\leq i<n_{\mathrm{ctx}}} e^{a_{t'_i}+b_{\sigma'(i)}}}{\sum_{0\leq j<n_{\mathrm{ctx}}} e^{a_{t'_j}+b_{\sigma'(j)}}} \leq s'_{\sigma'} - s_{\sigma'} \leq \max_{0\leq j<n_{\mathrm{ctx}}} w_{\sigma'(j)} \frac{\sum_{0\leq i<n_{\mathrm{ctx}}} e^{a_{t'_i}+b_{\sigma'(i)}}}{\sum_{0\leq j<n_{\mathrm{ctx}}} e^{a_{t'_j}+b_{\sigma'(j)}}}.$$

Thus we get as desired

$$\min_{0\leq i<n_{\mathrm{ctx}}} w_i \leq s'_{\sigma'} - s_{\sigma'} \leq \max_{0\leq i<n_{\mathrm{ctx}}} w_i.$$

$\square$

Note that we could prove a more fine-grained theorem, that pessimizes over attention paid to positions only for sequences compatible with the chosen fixed tokens $F$ and $t_F$, but since the positional contribution is so small we do not bother.

**Theorem 7** (For a fixed ordering, softmax is convex over token counts and only pure sequences need be considered). *Fix a set of fixed indices $F \subseteq \mathbb{N}_{<n_{\mathrm{ctx}}}$ and an assignment of token values to each of the fixed positions $t_F : F \to \mathbb{N}_{<d_{\mathrm{vocab}}}$. Fix a set $S \subseteq \mathbb{N}_{<d_{\mathrm{vocab}}}$ of valid other tokens in the sequence. (In our uses of this theorem, $S$ will be the largest subset of $\mathbb{N}_{<t_{\max}}$ for which we can guarantee that the model behaves correctly on all sequences compatible with $F$ and $t_{\max}$ and with tokens otherwise drawn from $S$.)*

*Define a comparison on non-negative integers less than $d_{\mathrm{vocab}}$:*

$$c := \sum_{i\in F} v_{t_F(i)} e^{a_{t_F(i)}+b_i} \qquad d := \sum_{i\in F} e^{a_{t_F(i)}+b_i} \qquad f := \sum_{\substack{0\leq i<n_{\mathrm{ctx}} \\ i\notin F}} e^{b_i}$$

$$\mathrm{cmp}(x,y) := \mathrm{sign}\left(d(e^{a_x} v_x - e^{a_y} v_y) - c(e^{a_x} - e^{a_y}) + f e^{a_x+a_y}\left(v_x e^{a_x+a_y} - v_y e^{a_x+a_y}\right)\right)$$

*Let $t_{\mathrm{cmp\,min}}$ and $t_{\mathrm{cmp\,max}}$ be the minimum and maximum elements of $S$ according to $\mathrm{cmp}$.[21]*

*For a given choice of a non-decreasing sequence of tokens $t_0 \leq \cdots \leq t_{n_{\mathrm{ctx}}-1} \in \mathbb{N}$ compatible with $F$ and $S$ and a given choice of permutation $\sigma : \mathbb{N} \to \mathbb{N}$ of the $n_{\mathrm{ctx}}$ positions fixing $F$ ($t_i = t_F(\sigma(i))$*

---

[21] We will prove that $\mathrm{cmp}$ is transitive in the process of proving this theorem.

*for $\sigma(i) \in F$; and $t_i \in S$ for $\sigma(i) \notin F$): let $s_{\sigma,\min}$ (and $s_{\sigma,\max}$) denote $s_{t_0,\dots,t_{n_{\mathrm{ctx}}-1},\sigma}$ when $t_i = t_{\mathrm{cmp\,min}}$ for all $\sigma(i) \notin F$ (or $t_{\mathrm{cmp\,max}}$, respectively).*

*Then for all such choices of sequence-permutation pairs,*

$$s_{\sigma,\min} \le s_{t_0,\dots,t_{n_{\mathrm{ctx}}-1},\sigma} \le s_{\sigma,\max}.$$

This theorem follows by chaining two lemmas: that scores are extremized by considering pure sequences, and that the extremal pure sequences match the comparison function defined in the theorem statement.

**Lemma 8** (Sequences scores are extremized on purer sequences). *Fix all the same quantities as in Theorem 7.*

*For any indices $0 \le i < j < n_{\mathrm{ctx}}$, token values $x, y \in S$, the score for a sequence with $t_i = x \ne y = t_j$ is bounded on both sides by sequences with $t_i = t_j = x$ and $t_i = t_j = y$.*

*Proof.* Let $s_{\alpha,\beta}$ be the sequence score with $t_i = \alpha$ and $t_j = \beta$, and define the score differences $\Delta_x := s_{x,x} - s_{x,y}$ and $\Delta_y := s_{y,y} - s_{x,y}$. It suffices to show that $\mathrm{sign}(\Delta_x \Delta_y) \le 0$. To show this, we must only compute the sign of $\Delta_\alpha$ for $\alpha \in \{x, y\}$ and show that whenever both $\Delta_x$ and $\Delta_y$ are non-zero, they have opposite signs.

We proceed by computation after defining some convenience variables for brevity:

$$C := \sum_{\substack{0 \le k < n_{\mathrm{ctx}} \\ k \ne i,j}} v_{t_k} e^{a_{t_k}+b_{\sigma(k)}} \qquad\qquad D := \sum_{\substack{0 \le k < n_{\mathrm{ctx}} \\ k \ne i,j}} e^{a_{t_k}+b_{\sigma(k)}}$$

$$\tilde\alpha := \begin{cases} x & \text{if } \alpha = y \\ y & \text{if } \alpha = x \end{cases} \qquad i_\alpha := \begin{cases} i & \text{if } \alpha = x \\ j & \text{if } \alpha = y \end{cases} \qquad i_{\tilde\alpha} := \begin{cases} i & \text{if } \tilde\alpha = x \\ j & \text{if } \tilde\alpha = y \end{cases}$$

$$\mathrm{sign}(\Delta_\alpha) = \mathrm{sign}\left( \frac{v_\alpha e^{a_\alpha+b_{\sigma(i)}} + v_\alpha e^{a_\alpha+b_{\sigma(j)}} + C}{e^{a_\alpha+b_{\sigma(i)}} + e^{a_\alpha+b_{\sigma(j)}} + D} - \frac{v_x e^{a_x+b_{\sigma(i)}} + v_y e^{a_y+b_{\sigma(j)}} + C}{e^{a_x+b_{\sigma(i)}} + e^{a_y+b_{\sigma(j)}} + D} \right)$$

$$= \mathrm{sign}\left( \frac{v_\alpha e^{a_\alpha+b_{\sigma(i_\alpha)}} + v_\alpha e^{a_\alpha+b_{\sigma(i_{\tilde\alpha})}} + C}{e^{a_\alpha+b_{\sigma(i_\alpha)}} + e^{a_\alpha+b_{\sigma(i_{\tilde\alpha})}} + D} - \frac{v_\alpha e^{a_\alpha+b_{\sigma(i_\alpha)}} + v_{\tilde\alpha} e^{a_{\tilde\alpha}+b_{\sigma(i_{\tilde\alpha})}} + C}{e^{a_\alpha+b_{\sigma(i_\alpha)}} + e^{a_{\tilde\alpha}+b_{\sigma(i_{\tilde\alpha})}} + D} \right)$$

Multiply through by positive denominators and simplify

$$= \mathrm{sign}\left( C\left( e^{a_{\tilde\alpha}+b_{\sigma(i_{\tilde\alpha})}} - e^{b_{\sigma(i_{\tilde\alpha})}+a_\alpha} \right) + D\left( v_\alpha e^{b_{\sigma(i_{\tilde\alpha})}+a_\alpha} - v_{\tilde\alpha} e^{a_{\tilde\alpha}+b_{\sigma(i_{\tilde\alpha})}} \right) \right.$$

$$\left. + v_\alpha \left( e^{b_{\sigma(i_{\tilde\alpha})}} + e^{b_{\sigma(i_\alpha)}} \right) e^{a_{\tilde\alpha}+b_{\sigma(i_{\tilde\alpha})}+a_\alpha} - v_{\tilde\alpha} \left( e^{b_{\sigma(i_{\tilde\alpha})}} + e^{b_{\sigma(i_\alpha)}} \right) e^{a_{\tilde\alpha}+b_{\sigma(i_{\tilde\alpha})}+a_\alpha} \right)$$

Pulling out $e^{b_{\sigma(i_{\tilde\alpha})}}$

$$= \mathrm{sign}\left( e^{a_{\tilde\alpha}+a_\alpha} \left( e^{b_{\sigma(i_{\tilde\alpha})}} + e^{b_{\sigma(i_\alpha)}} \right) (v_\alpha - v_{\tilde\alpha}) + C\left( e^{a_{\tilde\alpha}} - e^{a_\alpha} \right) + D\left( e^{a_\alpha} v_\alpha - e^{a_{\tilde\alpha}} v_{\tilde\alpha} \right) \right)$$

Note that swapping $\alpha$ and $\tilde\alpha$ negates the sign. Hence, we have $\mathrm{sign}(\Delta_x) = -\mathrm{sign}(\Delta_y)$ and hence $s_{x,x} \le s_{x,y} \le s_{y,y}$ or $s_{y,y} \le s_{x,y} \le s_{x,x}$ as desired. $\qquad\square$

**Lemma 9** (Pure sequences are sorted according to cmp in Theorem 7). *Fix all the same quantities as in Theorem 7.*

*Fix tokens $x, y \in S$. Let $n := n_{\mathrm{ctx}} - |F|$ be the number of non-fixed tokens. Fix sequences with $n$ copies of $x$ and $y$ respectively: fix $t_{x,0} \le \cdots \le t_{x,n_{\mathrm{ctx}}-1} \in \mathbb{N}$ and $t_{y,0} \le \cdots \le t_{y,n_{\mathrm{ctx}}-1} \in \mathbb{N}$ compatible with $F$ and $S$ and given choices of permutations $\sigma_x, \sigma_y : \mathbb{N} \to \mathbb{N}$ of the $n_{\mathrm{ctx}}$ positions fixing $F$: $t_{x,i} = t_F(\sigma_x(i))$ for $\sigma_x(i) \in F$; $t_{y,i} = t_F(\sigma_y(i))$ for $\sigma_y(i) \in F$; $t_{x,i} = x$ for $\sigma_x(i) \notin F$; and $t_{y,i} = y$ for $\sigma_y(i) \notin F$.*

*Then*

$$\mathrm{sign}\left( (s_{\sigma_x, t_{x,0},\dots,t_{x,n_{\mathrm{ctx}}-1}}) - (s_{\sigma_y, t_{y,0},\dots,t_{y,n_{\mathrm{ctx}}-1}}) \right) = \mathrm{cmp}(x, y)$$

$$\mathrm{EQKE}(t_{-1}, t_i) := t_{-1}\mathbf{E}_q QK^T\bar{\mathbf{E}}^T t_i{}^T/\sqrt{d}$$
$$\mathrm{EQKP}(t_{-1}, i) := t_{-1}\mathbf{E}_q QK^T\hat{\mathbf{P}}_i^T/\sqrt{d}$$
$$\mathrm{EVOU}(t_i) := t_i\bar{\mathbf{E}}VOU$$
$$\mathrm{PVOU}(i) := \hat{\mathbf{P}}_i VOU$$
$$\ell^{\mathrm{EU}}(t_{-1}) := t_{-1}\mathbf{E}_q U$$
$$\Delta\ell^{\mathrm{EU}}{}_{t^*}(t_{-1}, \max_i t_i) := \ell^{\mathrm{EU}}(t_{-1})_{t^*} - \ell^{\mathrm{EU}}(t_{-1})_{\max_i t_i}$$

**Figure 13:** Recapitulation of some relevant definitions from Figure 12, parameterized by the arguments they actually depend on.

*Proof.* The proof goes by straightforward computation.

$$\mathrm{sign}((s_{\sigma_x, t_{x,0}, \ldots, t_{x,n_{\mathrm{ctx}}-1}}) - (s_{\sigma_y, t_{y,0}, \ldots, t_{y,n_{\mathrm{ctx}}-1}}))$$
$$= \mathrm{sign}\left(\frac{v_x e^{a_x} f + c}{e^{a_x} f + d} - \frac{v_y e^{a_y} f + c}{e^{a_y} f + d}\right)$$

Multiply through by non-negative denominators

$$= \mathrm{sign}\left((v_x e^{a_x} f + c)(e^{a_y} f + d) - (v_y e^{a_y} f + c)(e^{a_x} f + d)\right)$$
$$= \mathrm{sign}\left(-cfe^{a_x} + cfe^{a_y} + dfv_x e^{a_x} - dfv_y e^{a_y} + f^2 v_x e^{a_x+a_y} - f^2 v_y e^{a_x+a_y}\right)$$

Use $f > 0$

$$= \mathrm{sign}\left(-ce^{a_x} + ce^{a_y} + dv_x e^{a_x} - dv_y e^{a_y} + fv_x e^{a_x+a_y} - fv_y e^{a_x+a_y}\right)$$
$$= \mathrm{sign}\left(c(e^{a_y} - e^{a_x}) + d(v_x e^{a_x} - v_y e^{a_y}) + f(v_x e^{a_x+a_y} - v_y e^{a_x+a_y})\right)$$
$$= \mathrm{cmp}(x, y)$$

$\square$

**Corollary 10.** *Define the relation $\leq_{\mathrm{cmp}}$ by $x \leq_{\mathrm{cmp}} y$ if and only if $\mathrm{cmp}(x, y) \in \{-1, 0\}$. The relation $\leq_{\mathrm{cmp}}$ is always transitive.*

*Proof.* Note that by Lemma 9, cmp is comparing two sequence scores. Since $\leq$ is transitive over the reals, the relation $\leq_{\mathrm{cmp}}$ is also transitive. $\square$

Finally, we combine the previous lemmas to complete our proof of Theorem 7:

*Proof of Theorem 7.* Extremal sequences with scores $s_{\sigma,\min}$ and $s_{\sigma,\max}$ are guaranteed to exist because there are only finitely many elements of $S$ and therefore only finitely many sequences. By Lemma 8, the extremal sequences must be pure (have $t_i = t_j$ whenever $\sigma(i), \sigma(j) \notin F$). By Lemma 9, the extremal sequences must have tokens that are extremal according to cmp. $\square$

We now have all the tools necessary to prove the following theorem. We refer to Algorithm 3 and Algorithm 4 or the proof of Theorem 11 for a definition of the CUBIC algorithm.

**Theorem 11.**

$$\mathbb{E}_{\mathbf{t} \sim U(0,1,\ldots,d_{\mathrm{vocab}}-1)^{n_{\mathrm{ctx}}}}\left[\underset{i}{\mathrm{argmax}}(\mathcal{M}(\mathbf{t}))_i = \max_i t_i\right] \geq \mathrm{CUBIC}(d_{\mathrm{vocab}}, n_{\mathrm{ctx}}, \mathcal{M})$$

Before we give the proof of this theorem, we introduce some helpful notation.

**Definition 12.** *Fix an element $(r_m, r_q, c) \in \{0, \ldots, d_{\mathrm{vocab}}\}^2 \times \{0, \ldots 3\}$ such that $r_m \geq r_q$. We define $X_{(r_m, r_q, c)}$ to be the set of tokens $\mathbf{t}$ such that*

    *1. The max token $t_{\max}$ is equal to $r_m$,*

---

**Algorithm 2** Counting Correct Sequences in Cubic Time: Preliminaries

---

1: **function** CORRECTNESS($\mathcal{M}$, input-sequence)
2:     **return** MODEL-BEHAVIOR($\mathcal{M}$, input-sequence) $==$ MAX(input-sequence)
3: **end function**
4: **function** MODEL-BEHAVIOR($\mathcal{M}$, input-sequence)
**Require:** input-sequence is a tensor of shape $(n_{\text{ctx}}, )$ with values in $\mathbb{N}_{<d_{\text{vocab}}}$
5:     $t_{\max} \leftarrow$ MAX(input-sequence)                                             $\triangleright t_{\max} \leftarrow$ max-token
6:     $\mathbf{t} \leftarrow$ input-sequence
7:     skip-score$_{t^*} \leftarrow \Delta\ell^{\text{EU}}{}_{t^*}(t_{n_{\text{ctx}}-1}, t_{\max})$
8:     attn-weights-unscaled$_i \leftarrow$ EQKE$(t_{n_{\text{ctx}}-1}, t_i) +$ EQKP$(t_{n_{\text{ctx}}-1}, i)$
9:     attn-weights $\leftarrow$ SOFTMAX(attn-weights-unscaled$/\sqrt{d}$)
10:    $v_t \leftarrow$ EVOU$(t)$
11:    $w_i \leftarrow$ PVOU$(i)$
12:    $\Delta v_{t,t^*} \leftarrow v_{t,t^*} - v_{t,t_{\max}}$
13:    $\Delta w_{i,t^*} \leftarrow w_{i,t^*} - w_{i,t_{\max}}$
14:    **return** $\max_{t^* \neq t_{\max}}(\text{skip-score}_{t^*} + \sum_{i=0}^{n_{\text{ctx}}-1}(\Delta v_{i,t^*} + \Delta w_{i,t^*}) \cdot \text{attn-weights}_i)$
15: **end function**
16: **function** CORRECTNESS-PESSIMIZING-OVER-POSITION-SLOW($\mathcal{M}$, input-sequence)
17:    $\mathbf{t} \leftarrow$ input-sequence
18:    **return** ALL(CORRECTNESS($\mathcal{M}$, perm $+ [t_{-1}]$) **for all** perm $\in$ PERMUTATIONS($t_{0:-1}$))
19: **end function**

---

    2. *The query token $t_{\text{query}}$ is equal to $r_q$,*

    3. *The cardinality of tokens that are not at the query position and not equal to $t_{\max}$ is equal to c.*

For clarity, we list all the possible cases. We always take $t_{\text{query}} \leq t_{\max}$ and let $S_3$ act on sequences by permuting the first three factors (i.e. keeping the query position fixed).

    1. If $c = 0$, then $X_{(t_{\max}, t_{\text{query}}, 0)} = \{[t_{\max}, t_{\max}, t_{\max}, t_{\text{query}}]\}$,

    2. If $c = 1$, then $X_{(t_{\max}, t_{\text{query}}, 1)} = S_3.\{[t_1, t_{\max}, t_{\max}, t_{\text{query}}] \mid t_1 < t_{\max}\}$,

    3. If $c = 2$, then $X_{(t_{\max}, t_{\text{query}}, 2)} = S_3.\{[t_1, t_2, t_{\max}, t_{\text{query}}] \mid t_i < t_{\max}\}$,

    4. If $c = 3$, then $X_{(t_{\max}, t_{\max}, 3)} = S_3.\{[t_1, t_2, t_3, t_{\max}] \mid t_i < t_{\max}\}$.

**Definition 13.** *Let $\mathbf{t} \in X$ be a sequence. We say t is pure, if it has at most three distinct tokens: the max token $t_{\max}$, the query token $t_{\text{query}}$, and optionally a third token $t^* < t_{\max}$.*

*We denote by $X^{pure}$ the subset of pure tokens. For any subset $Y \subset X$, we set $Y^{pure} := Y \cap X^{pure}$.*

We now come to the proof of Theorem 11. We will show how to use the previous theorems to get explicit bounds and explain how CUBIC$(d_{\text{vocab}}, n_{\text{ctx}}, \mathcal{M})$ computes these bounds.

*Proof of Theorem 11.* First of all, we note that the algorithm CUBIC $=$ CUBIC$(d_{\text{vocab}}, n_{\text{ctx}}, \mathcal{M})$ yields a lower bound for the accuracy on the set $X_{(t_{\max}, t_{\text{query}}, c)}$. We can therefore compute the bound on $X = \coprod_{(t_{\max}, t_{\text{query}}, c)} X_{(t_{\max}, t_{\text{query}}, c)}$ by computing it for each such choice $(t_{\max}, t_{\text{query}}, c)$ and summing over them

$$\mathbb{E}_{\mathbf{t} \sim U(0, 1, ..., d_{\text{vocab}}-1)^{n_{\text{ctx}}}} \left[ \arg\max_i (\mathcal{M}(\mathbf{t}))_i = \max_i t_i \right] \geq \sum_{(t_{\max}, t_{\text{query}}, c)} \text{CUBIC}(X_{(t_{\max}, t_{\text{query}}, c)}).$$

So from now on we will fix one such subset $X_{(t_{\max}, t_{\text{query}}, c)}$.

We begin by defining a map

$$f : X_{(t_{\max}, t_{\text{query}}, c)} \to \{0, ..., d_{\text{vocab}}\}^c$$

which sends a sequence to the subsequence of elements which are not at the query position and not equal to $t_{\max}$. Then Theorem 7 can be restated as follows[22]:

---

[22] In fact, the theorem yields a stronger result, but we will only need the following formulation.

**Algorithm 3** Counting Correct Sequences in Cubic Time, Part I. Lines are annotated with comments indicating the parameters for a cache to avoid duplicate computations.

---

1: **function** MODEL-BEHAVIOR-RELAXED($\mathcal{M}$, query-tok, max-tok, non-max-tok, n-copies-nonmax)
2: $\quad$ $t_{\text{query}} \leftarrow$ query-tok, $t_{\max} \leftarrow$ max-tok, $t' \leftarrow$ non-max-tok, $c \leftarrow$ n-copies-nonmax
**Require:** $0 \le t_{\text{query}} \le t_{\max} < d_{\text{vocab}}, 0 \le t' \le t_{\max} < d_{\text{vocab}}, 0 \le c < n_{\text{ctx}}$
**Require:** **if** n-copies-nonmax $= 0$ **then** non-max-tok $=$ max-tok
**Require:** **if** query-tok $\ne$ max-tok **then** n-copies-nonmax $< n_{\text{ctx}} - 1$
**Ensure:** **return** $\ge$ MODEL-BEHAVIOR($\mathcal{M}, \mathbf{t}$) **for all** $\mathbf{t}$ with specified $t_{\text{query}}$, $c$ copies of $t'$ in non-query positions, and the remainder of the tokens equal to $t_{\max}$
3: $\quad$ skip-score$_{t^*} \leftarrow \Delta\ell^{\text{EU}}{}_{t^*}(t_{\text{query}}, t_{\max})$ $\qquad\qquad\qquad$ ▷ Cache by $t_{\max}, t_{\text{query}}, t^*$
4: $\quad$ $w_i \leftarrow \text{PVOU}(i)$ $\quad$ **for** $0 \le i < n_{\text{ctx}}$ $\qquad\qquad\qquad\qquad$ ▷ Cache by $i$
5: $\quad$ $\Delta w_{\max,t^*} \leftarrow \max_{0 \le i < n_{\text{ctx}}}(w_{i,t^*} - w_{i,t_{\max}})$ $\qquad\qquad$ ▷ Cache by $t_{\max}, t^*$
6: $\quad$ $v_t \leftarrow \text{EVOU}(t), \Delta v_{t,t^*} \leftarrow v_{t,t^*} - v_{t,t_{\max}}$ $\quad$ **for** $t \in \{t_{\text{query}}, t_{\max}, t'\}$ ▷ Cache by $t_{\max}, t,$ $t^*$
7: $\quad$ $a_t \leftarrow \text{EQKE}(t_{\text{query}}, t)/\sqrt{d}$ $\quad$ **for** $t \in \{t_{\text{query}}, t_{\max}, t'\}$ $\qquad$ ▷ Cache by $t_{\text{query}}, t$
8: $\quad$ $b_{n_{\text{ctx}}-1} \leftarrow \text{EQKP}(t_{\text{query}}, n_{\text{ctx}} - 1)/\sqrt{d}$ $\qquad\qquad$ ▷ Cache by $t_{\text{query}}$
9: $\quad$ $b_{0,:-1} \leftarrow \text{SORT}(\text{EQKP}(t_{\text{query}}, :-1))/\sqrt{d}$ $\qquad\qquad$ ▷ Cache by $t_{\text{query}}, i$
10: $\quad$ $b_{1,:-1} \leftarrow \text{REVERSE}(b_{0,:-1})$
11: $\quad$ attn-weights-unscaled$_{:,n_{\text{ctx}}-1} \leftarrow a_{t_{\text{query}}} + b_{n_{\text{ctx}}-1}$ $\qquad$ ▷ Cache by $t_{\text{query}}$
12: $\quad$ attn-weights-unscaled$_{0,i} \leftarrow a_{t_{\max}} + b_{0,i}$ $\quad$ **for** $0 \le i < n_{\text{ctx}} - c - 1$ $\;$ ▷ Cache by $t_{\max}, c, i,$ $t_{\text{query}}$
13: $\quad$ attn-weights-unscaled$_{1,i} \leftarrow a_{t_{\max}} + b_{1,i}$ $\quad$ **for** $0 \le i < n_{\text{ctx}} - c - 1$ $\;$ ▷ Cache by $t_{\max}, c, i,$ $t_{\text{query}}$
14: $\quad$ attn-weights-unscaled$_{0,i} \leftarrow a_{t'} + b_{0,i}$ $\quad$ **for** $n_{\text{ctx}} - c - 1 \le i < n_{\text{ctx}} - 1$ ▷ Cache by $t', c, i,$ $t_{\text{query}}$
15: $\quad$ attn-weights-unscaled$_{1,i} \leftarrow a_{t'} + b_{1,i}$ $\quad$ **for** $n_{\text{ctx}} - c - 1 \le i < n_{\text{ctx}} - 1$ ▷ Cache by $t', c, i,$ $t_{\text{query}}$
16: $\quad$ attn-weights$_0 \leftarrow \text{SOFTMAX}(\text{attn-weights-unscaled}_0)$ $\qquad$ ▷ Cache by $t_{\max}, t', c, i, t_{\text{query}}$
17: $\quad$ attn-weights$_1 \leftarrow \text{SOFTMAX}(\text{attn-weights-unscaled}_1)$ $\qquad$ ▷ Cache by $t_{\max}, t', c, i, t_{\text{query}}$
18: $\quad$ **if** $c = 0$ **then** ▷ In this case, attn-weights$_{0,i} =$ attn-weights$_{1,i}$, so we drop the first subscript
19: $\qquad$ **return** $\max_{t^* \ne t_{\max}}($skip-score$_{t^*} + \Delta w_{\max,t^*} + \Delta v_{t_{-1},t^*}$attn-weights$_{-1} + \Delta v_{t_{\max},t^*} \sum_{i=0}^{n_{\text{ctx}}-2}$ attn-weights$_i)$
20: $\quad$ **else**
21: $\qquad$ $\Delta v_{i,t^*} \leftarrow \Delta v_{t_{\max},t^*}$ $\quad$ **for** $0 \le i < n_{\text{ctx}} - c - 1$
22: $\qquad$ $\Delta v_{i,t^*} \leftarrow \Delta v_{t',t^*}$ $\quad$ **for** $n_{\text{ctx}} - c - 1 \le i < n_{\text{ctx}} - 1$
23: $\qquad$ $\Delta v_{n_{\text{ctx}}-1,t^*} \leftarrow \Delta v_{t_{\text{query}},n_{\text{ctx}}-1}$
24: $\qquad$ **return** $\max_{t^* \ne t_{\max}}$ skip-score$_{t^*}$+max $\begin{cases} \sum_{i=0}^{n_{\text{ctx}}-1} \max_{t^* \ne t_{\max}}(\Delta w_{\max,t^*} + \Delta v_{i,t^*}) \cdot \text{attn-weights}_{0,i} \\ \sum_{i=0}^{n_{\text{ctx}}-1} \max_{t^* \ne t_{\max}}(\Delta w_{\max,t^*} + \Delta v_{i,t^*}) \cdot \text{attn-weights}_{1,i} \end{cases}$
25: $\quad$ **end if**
26: **end function**
27: **function** RELAXED-CORRECTNESS-PESSIMIZING-OVER-POSITION($\mathcal{M}, t_{\text{query}}, t_{\max}, t', c$)
28: $\qquad$ ▷ runs the model on a relaxed variant of input sequences compatible with the arguments
**Ensure:** **return** is False **if** CORRECTNESS-PESSIMIZING-OVER-POSITION-SLOW($\mathcal{M}, \mathbf{t}$) is False **for** *any* $\mathbf{t}$ with specified $t_{\text{query}}$, $c$ copies of $t'$ in non-query positions, and the remainder of the tokens equal to $t_{\max}$
29: $\quad$ **return** MODEL-BEHAVIOR-RELAXED($\mathcal{M}, t_{\text{query}}, t_{\max}, t', c$) $< 0$
30: **end function**

---

---

**Algorithm 4** Counting Correct Sequences in Cubic Time, Part II

---

1: **function** CUBIC($d_{\text{vocab}}, n_{\text{ctx}}, \mathcal{M}$)
2:     count $\leftarrow 0$                                         $\triangleright$ # of correct sequences
3:     **for** $t_{\max} \in$ RANGE($d_{\text{vocab}}$) **do**                           $\triangleright$ $t_{\max} \leftarrow$ max-token
4:         **for** $0 \leq t_{\text{query}} \leq t_{\max}$ **do**                         $\triangleright$ $t_{\text{query}} \leftarrow$ query-token
5:             $c_{\max} \leftarrow n_{\text{ctx}} - 1$ **if** $t_{\text{query}} = t_{\max}$ **else** $n_{\text{ctx}} - 2$    $\triangleright$ maximum copies of nonmax
6:             **for** $0 \leq c \leq c_{\max}$ **do**          $\triangleright$ number of valid choices for the non-max token
7:                 RCPOP($\vec{\chi}$) $\leftarrow$ RELAXED-CORRECTNESS-PESSIMIZING-OVER-POSITION($\mathcal{M}, \vec{\chi}$)
8:                 **if** $c = 0$ **then**
9:                     t-count $\leftarrow 1$ **if** RCPOP($t_{\text{query}}, t_{\max}, t_{\max}, 0$) **else** $0$
10:                 **else**
11:                     t-count $\leftarrow \sum_{t'=0}^{t_{\max}-1} 1$ **if** RCPOP($t_{\text{query}}, t_{\max}, t', c$) **else** $0$
12:                 **end if**
13:                 count $\leftarrow$ count $+ \binom{n_{\text{ctx}}-1}{c} \cdot$ (t-count)$^c$       $\triangleright$ taking $0^0 = 0$ conventionally
14:             **end for**
15:         **end for**
16:     **end for**
17:     **return** count $\cdot \frac{1}{d_{\text{vocab}}{}^{n_{\text{ctx}}}}$
18: **end function**

---

Let $S \subset \{0, ..., d_{\text{vocab}}\}$. Then full accuracy $f^{-1}(S^c)^{pure} := X^{pure}_{(t_{\max}, t_{\text{query}}, c)} \cap f^{-1}(S^c)$, implies full accuracy on $f^{-1}(S^c)$.

Now instead of computing the output of the model for every element $f^{-1}(S^c)^{pure}$, we use Theorem 4 (combined with Theorem 6) to run a relaxed version of this. In particular, we may assume that the pure sequence is contiguous on equal tokens. Here contiguous on equal tokens means that for the positional part of the attention (i.e. the EQKP part), we have either $b_{t_{\max}} < \{b_i, b_j\}$ or $b_{t_{\max}} > \{b_i, b_j\}$, where $i, j \in \{0, ..., n_{\text{ctx}} - 1\}$ are indices of tokens not equal to $t_{\max}$.

For the algorithm CUBIC($d_{\text{vocab}}, n_{\text{ctx}}, \mathcal{M}$) we fix a $t^* \in \{0, ..., t_{\max}-1\}$ (unless $c = 0$, in which case there is no such choice). We then run the relaxed accuracy computation RCPOP($t_{\text{query}}, t_{\max}, t', c$) as described in Theorem 6. If RCPOP($t_{\text{query}}, t_{\max}, t', c$) $< 0$, we add $t'$ to $S$. If we do, we add $t^*$ to $S$. Therefore by construction of $S$ we know that we get full accuracy on $f^{-1}(S^c)^{pure}$ and therefore we get full accuracy on $f^{-1}(S^c)$.

Now we count the cardinality of $f^{-1}(S^c)$ and add it to the count of correct sequences.

$\square$

**Theorem 12.** *The running time of Algorithm 3, after using caching to avoid duplicate computations, is $\mathcal{O}(d_{\text{vocab}}{}^3 n_{\text{ctx}}{}^2)$.*

*Proof.* The nested loops in CUBIC execute the innermost body $\mathcal{O}(d_{\text{vocab}}{}^2 n_{\text{ctx}})$ times, and the summation on Line 13 costs $\mathcal{O}(n_{\text{ctx}})$ per iteration. What remains is to show that the call to RELAXED-CORRECTNESS-PESSIMIZING-OVER-POSITION($\mathcal{M}, t_{\text{query}}, t_{\max}, t', c$) costs $\mathcal{O}(n_{\text{ctx}})$ when $c \neq 0$ and at most $\mathcal{O}(d_{\text{vocab}} n_{\text{ctx}})$ when $c = 0$ and $t' = t_{\max}$.

The matrix multiplications in EQKE, EQKP, EVOU, PVOU, and $\ell^{\text{EU}}$ can be cached upfront, costing $\mathcal{O}(\max(d_{\text{vocab}}, d_{\text{model}}, n_{\text{ctx}})^2 d_{\text{model}}) \leq \mathcal{O}(d_{\text{vocab}}{}^3)$ since we assume $d_{\text{vocab}} > d_{\text{model}}$ and $d_{\text{vocab}} > n_{\text{ctx}}$.

The sorting on Line 9 can also be cached upfront (per $t_{\text{query}}$), costing $\mathcal{O}(d_{\text{vocab}} n_{\text{ctx}} \log n_{\text{ctx}})$.

Note that each variable assignment in RELAXED-CORRECTNESS-PESSIMIZING-OVER-POSITION can be cached into a table parameterized over at most three variables which range over $d_{\text{vocab}}$ and over at most two variables that range over $n_{\text{ctx}}$.

What remains is the **return** statements.

When $c = 0$, we have on Line 19: **return** $\max_{t^* \neq t_{\max}}(\text{skip-score}_{t^*} + \Delta w_{\max, t^*} + \Delta v_{t_{-1}, t^*} \text{attn-weights}_{-1} + \Delta v_{t_{\max}, t^*} \sum_{i=0}^{n_{\text{ctx}}-2} \text{attn-weights}_i)$. This is $\mathcal{O}(d_{\text{vocab}} n_{\text{ctx}})$ as desired.

When $c \neq 0$, we have on Line 24:

$$\textbf{return} \max_{t^* \neq t_{\max}} \text{skip-score}_{t^*} + \max \begin{cases} \sum_{i=0}^{n_{\text{ctx}}-1} \max_{t^* \neq t_{\max}} (\Delta w_{\max,t^*} + \Delta v_{i,t^*}) \cdot \text{attn-weights}_{0,i} \\ \sum_{i=0}^{n_{\text{ctx}}-1} \max_{t^* \neq t_{\max}} (\Delta w_{\max,t^*} + \Delta v_{i,t^*}) \cdot \text{attn-weights}_{1,i} \end{cases}$$

We can cache $\max_{t^* \neq t_{\max}} \text{skip-score}_{t^*}$ per $t_{\max}$ and $t_{\text{query}}$, costing $\mathcal{O}(d_{\text{vocab}}{}^3 n_{\text{ctx}})$. We can cache $\max_{t^* \neq t_{\max}} (\Delta w_{\max,t^*} + \Delta v_{i,t^*})$ per $t_{\max}$ and $t'$ costing $\mathcal{O}(d_{\text{vocab}}{}^3)$, since each $\Delta v_{i,t^*}$ will be $\Delta v_{t,t^*}$ for some $t \in \{t_{\text{query}}, t_{\max}, t'\}$. Finally, we can compute the summation in cost $\mathcal{O}(n_{\text{ctx}})$ per loop iteration, as required. $\qquad\square$

# F   Quadratic counting for a sub-cubic proof

In this section we fill in the details lacking from Section 4.3.

In Appendix E we proved an intricate version of convexity of softmax where, modulo pessimizing in unrealistic ways over the attention paid to positions for the computation done on positional encodings, all extremal relaxed sequences correspond to actual sequences.

When we only get a budget of $\mathcal{O}(d_{\text{vocab}}{}^2 n_{\text{ctx}})$ extremal relaxed cases to consider, though, we must pessimize more, which gives us a simpler version of the convexity theorem and proof. Notably, when we restrict our sequences to have only two tokens (the max token $t_{\max}$ and the non-max token $t'$), most of the theorems from Appendix E.3 get significantly simpler.

Additionally, we must pessimize separately over the token value ($v$) and token attention ($b$) computations in order to allow efficient computation (Theorem 15).

## F.1   Proof of baseline sub-cubic result

For this subsection, all theorems are parameterized over the following quantities.

**Definition 14** (Common theorem parameters). *Fix a total number of tokens $n_{\text{ctx}}$. Fix a token value function (à la a row-difference in* EVOU*) $v : \mathbb{N}_{<d_{\text{vocab}}} \to \mathbb{R}$ and a token attention function (à la* EQKE *for a fixed query token) $a : \mathbb{N}_{<d_{\text{vocab}}} \to \mathbb{R}$. Fix a position value function (à la a row-difference in* PVOU*) $w : \mathbb{N}_{<n_{\text{ctx}}} \to \mathbb{R}$ and a position attention function (à la* EQKP *for a fixed query token) $b : \mathbb{N}_{<n_{\text{ctx}}} \to \mathbb{R}$.*

In practice, as in Appendix E.3, we'll take, for fixed query token $t_{\text{query}}$,

$$v_t = \text{EVOU}_{t,t^*} - \text{EVOU}_{t,t_{\max}} \qquad\qquad a_t = \text{EQKE}_{t_{\text{query}},t}/\sqrt{d}$$

$$w_i = \text{PVOU}_{i,t^*} - \text{PVOU}_{i,t_{\max}} \qquad\qquad b_i = \text{EQKP}_{t_{\text{query}},i}/\sqrt{d}$$

Note that unlike in Appendix E.3, we pessimize independently over the query token and the non-max token, so the "fixed" query token may not in fact appear in any key-side position in the relaxed sequence we consider.

**Definition 15** (of a sequence via mapping from positions). *We can define a sequence of tokens via mapping from positions by specifying a subset of valid tokens $S \subseteq \mathbb{N}_{<d_{\text{vocab}}}$ paired with a function $T : \mathbb{N}_{<n_{\text{ctx}}} \to S$ specifying which token is in each position.*

**Definition 16** (sequence score). *Given a subset of valid tokens $S \subseteq \mathbb{N}_{<d_{\text{vocab}}}$ and a function $T : \mathbb{N}_{<n_{\text{ctx}}} \to S$ specifying which token is in each position, define the* sequence score

$$s_T := \sum_{0 \leq i < n_{\text{ctx}}} v_{T(i)} e^{a_{T(i)} + b_i} \Big/ \sum_{0 \leq i < n_{\text{ctx}}} e^{a_{T(i)} + b_i}$$

**Definition 17** (swapped mapping). *Given a subset of valid tokens $S \subseteq \mathbb{N}_{<d_{\text{vocab}}}$ and a function $T : \mathbb{N}_{<n_{\text{ctx}}} \to S$ specifying which token is in each position and two indices $0 \leq i, j < n_{\text{ctx}}$, define the* swapped mapping $T_{i \leftrightarrow j}$ *be the function that is $T$ except swapping $i$ and $j$:*

$$T_{i \leftrightarrow j}(k) = \begin{cases} T(i) & \text{if } k = j \\ T(j) & \text{if } k = i \\ T(k) & \text{otherwise} \end{cases}$$

**Lemma 13** (Characterization of swapping tokens in a two-token sequence). *Fix two tokens $t_0 < t_1 \in \mathbb{N}$ and a function $T : \mathbb{N}_{<n_{\mathrm{ctx}}} \to \{t_0, t_1\}$ specifying which token is in each position.*

*Define $\Delta_{T, i \leftrightarrow j}$ to be the difference in sequence scores when you swap $i$ and $j$:*

$$\Delta_{T, i \leftrightarrow j} := s_{T_{i \leftrightarrow j}} - s_T$$

*Then*

$$\mathrm{sign}\left(\Delta_{T, i \leftrightarrow j}\right) = -\,\mathrm{sign}\left(b_i - b_j\right) \mathrm{sign}\left(v_{T(i)} - v_{T(j)}\right)$$

*Proof.* [Lemma 3] gives us the result directly when $a_{T(i)} = a_{T(j)}$. Otherwise, we get

$$\mathrm{sign}\left(\Delta_{T, i \leftrightarrow j}\right) = \mathrm{sign}\left(a_{T(i)} - a_{T(j)}\right) \mathrm{sign}\left(b_i - b_j\right) \mathrm{sign}\left(s_T - \frac{v_{T(i)} e^{a_{T(i)}} - v_{T(j)} e^{a_{T(j)}}}{e^{a_{T(i)}} - e^{a_{T(j)}}}\right)$$

Hence all that remains is to show that

$$\mathrm{sign}\left(s_T \left(e^{a_{T(i)}} - e^{a_{T(j)}}\right) - v_{T(i)} e^{a_{T(i)}} + v_{T(j)} e^{a_{T(j)}}\right) = -\,\mathrm{sign}\left(v_{T(i)} - v_{T(j)}\right)$$

Define $\bar{v} := \frac{1}{2}(v_{T(i)} + v_{T(j)})$ and define $\Delta v := \frac{1}{2}(v_{T(i)} - v_{T(j)})$ so that $v_{T(i)} = \bar{v} + \Delta v$ and $v_{T(j)} = \bar{v} - \Delta v$. Assume WLOG that $T(i) = 0$ and $T(j) = 1$ so that $v_{T(p)} = \bar{v} + (-1)^{T(p)} \Delta v$ for all $p$.

Then we have

$$\mathrm{sign}\left(s_T \left(e^{a_{T(i)}} - e^{a_{T(j)}}\right) - v_{T(i)} e^{a_{T(i)}} + v_{T(j)} e^{a_{T(j)}}\right)$$
$$= \mathrm{sign}\left(s_T \left(e^{a_{T(i)}} - e^{a_{T(j)}}\right) - \bar{v}\left(e^{a_{T(i)}} - e^{a_{T(j)}}\right) - \Delta v\, e^{a_{T(i)}} - \Delta v\, e^{a_{T(j)}}\right)$$
$$= \mathrm{sign}\left(\frac{\displaystyle\sum_{0 \le p < n_{\mathrm{ctx}}} v_{T(p)} e^{a_{T(p)} + b_p}}{\displaystyle\sum_{0 \le p < n_{\mathrm{ctx}}} e^{a_{T(p)} + b_p}} \left(e^{a_{T(i)}} - e^{a_{T(j)}}\right) - \bar{v}\left(e^{a_{T(i)}} - e^{a_{T(j)}}\right) - \Delta v\left(e^{a_{T(i)}} + e^{a_{T(j)}}\right)\right)$$
$$= \mathrm{sign}\left(\frac{\displaystyle\sum_{0 \le p < n_{\mathrm{ctx}}} \left(\bar{v} + (-1)^{T(p)} \Delta v\right) e^{a_{T(p)} + b_p}}{\displaystyle\sum_{0 \le p < n_{\mathrm{ctx}}} e^{a_{T(p)} + b_p}} \left(e^{a_{T(i)}} - e^{a_{T(j)}}\right) - \bar{v}\left(e^{a_{T(i)}} - e^{a_{T(j)}}\right) - \Delta v\left(e^{a_{T(i)}} + e^{a_{T(j)}}\right)\right)$$
$$= \mathrm{sign}(\Delta v)\,\mathrm{sign}\left(\frac{e^{a_{T(i)}} \displaystyle\sum_{\substack{0 \le p < n_{\mathrm{ctx}} \\ T(p) = T(i)}} e^{b_p} - e^{a_{T(j)}} \displaystyle\sum_{\substack{0 \le p < n_{\mathrm{ctx}} \\ T(p) = T(j)}} e^{b_p}}{\displaystyle\sum_{0 \le p < n_{\mathrm{ctx}}} e^{a_{T(p)} + b_p}} \left(e^{a_{T(i)}} - e^{a_{T(j)}}\right) - e^{a_{T(i)}} - e^{a_{T(j)}}\right)$$
$$= \mathrm{sign}(v_{T(i)} - v_{T(j)})\,\mathrm{sign}\left(\frac{e^{a_{T(i)}} \displaystyle\sum_{\substack{0 \le p < n_{\mathrm{ctx}} \\ T(p) = T(i)}} e^{b_p} - e^{a_{T(j)}} \displaystyle\sum_{\substack{0 \le p < n_{\mathrm{ctx}} \\ T(p) = T(j)}} e^{b_p}}{\displaystyle\sum_{0 \le p < n_{\mathrm{ctx}}} e^{a_{T(p)} + b_p}} \left(e^{a_{T(i)}} - e^{a_{T(j)}}\right) - e^{a_{T(i)}} - e^{a_{T(j)}}\right)$$

Define

$$P_i := \sum_{\substack{0 \le p < n_{\mathrm{ctx}} \\ T(p) = T(i)}} e^{b_p} \qquad\qquad P_j := \sum_{\substack{0 \le p < n_{\mathrm{ctx}} \\ T(p) = T(j)}} e^{b_p}$$

so that we get

$$\mathrm{sign}\left(s_T \left(e^{a_{T(i)}} - e^{a_{T(j)}}\right) - v_{T(i)} e^{a_{T(i)}} + v_{T(j)} e^{a_{T(j)}}\right)$$

$$= \text{sign}(v_{T(i)} - v_{T(j)}) \, \text{sign}\left(\frac{e^{a_{T(i)}}P_i - e^{a_{T(j)}}P_j}{e^{a_{T(i)}}P_i + e^{a_{T(j)}}P_j}\left(e^{a_{T(i)}} - e^{a_{T(j)}}\right) - e^{a_{T(i)}} - e^{a_{T(j)}}\right)$$

Multiply through by the positive denominator and expand out so that we get

$$= \text{sign}(v_{T(i)} - v_{T(j)}) \, \text{sign}\left(-2e^{a_{T(i)}+a_{T(j)}}P_i - 2e^{a_{T(i)}+a_{T(j)}}P_j\right)$$

$$= -\text{sign}(v_{T(i)} - v_{T(j)}) \, \text{sign}\left(e^{a_{T(i)}+a_{T(j)}}P_i + e^{a_{T(i)}+a_{T(j)}}P_j\right)$$

$$= -\text{sign}(v_{T(i)} - v_{T(j)})$$

$\square$

**Theorem 14** (Pessimization over sequence ordering for two-token sequences is simple). *Let $\sigma_s : \mathbb{N} \to \mathbb{N}$ denote a permutation of the $n_{\text{ctx}}$ positions that sorts them according to $b$: for $0 \le i, j < n_{\text{ctx}}$, $b_i \le b_j$ whenever $\sigma_s(i) < \sigma_s(j)$. Fix two tokens $t_0 < t_1 \in \mathbb{N}$.*

*Let $n_{t_0}$ be the number of $p \in [0, n_{\text{ctx}})$ with $T(p) = t_0$ and let $n_1$ be the number of $p \in [0, n_{\text{ctx}})$ with $T(p) = t_{t_1}$. Note that $n_{t_0} + n_{t_1} = n_{\text{ctx}}$.*

*Define $t_{\min} := \text{argmin}_{t \in \{t_0, t_1\}} v_t$ and define $t_{\max} := \text{argmax}_{t \in \{t_0, t_1\}} v_t$.*

*Define $T_{\min}, T_{\max} : \mathbb{N}_{<n_{\text{ctx}}} \to \{t_0, t_1\}$ to be the assignment of tokens to positions that pays the least (respectively, most) attention to $t_{\max}$:*

$$T_{\min}(i) := \begin{cases} t_{\max} & \text{if } 0 \le \sigma_s(i) < n_{t_{\max}} \\ t_{\min} & \text{if } n_{t_{\max}} \le \sigma_s(i) < n_{\text{ctx}} \end{cases}$$

$$T_{\max}(i) := \begin{cases} t_{\min} & \text{if } 0 \le \sigma_s(i) < n_{t_{\min}} \\ t_{\max} & \text{if } n_{t_{\min}} \le \sigma_s(i) < n_{\text{ctx}} \end{cases}$$

*Then we have that*

$$s_{T_{\min}} \le s_T \le s_{T_{\max}}$$

*Proof.* The extremality of $s_{T_{\min}}$ and $s_{T_{\max}}$ follows straightforwardly from Theorem 4.

All that remains is $s_{T_{\min}} \le s_{T_{\max}}$.

This follows from noting by Lemma 13 that swapping two tokens in $T_{\min}$ *increases* the sequence score, while the reverse is true of $s_{T_{\max}}$, thus showing that it must be $s_{T_{\min}}$ that is the minimum and $s_{T_{\max}}$ that is the maximum and not vice versa. $\square$

**Definition 18** (full sequence score). *Given a subset of valid tokens $S \subseteq \mathbb{N}_{<d_{\text{vocab}}}$ and a function $T : \mathbb{N}_{<n_{\text{ctx}}} \to S$ specifying which token is in each position define the* full sequence score *$s'_T$:*

$$s'_T := \sum_{0 \le i < n_{\text{ctx}}} (v_{T(i)} + w_i)e^{a_{T(i)}+b_i} \Bigg/ \sum_{0 \le i < n_{\text{ctx}}} e^{a_{T(i)}+b_i}$$

**Theorem 15** (Independent pessimization over positional contributions and token attention and token value is possible). *Fix two tokens $t_0 < t_1 \in \mathbb{N}$. Let $T_{\min}, T_{\max} : \mathbb{N}_{<n_{\text{ctx}}} \to \{t_0, t_1\}$ and $t_{\max}, t_{\min}$ be as in Theorem 14. Fix a set $S$ of valid tokens with $t_0, t_1 \in S$.*

*Define relaxed versions $T'_{\max}, T'_{\min} : \mathbb{N}_{<n_{\text{ctx}}} \to S$ of $T_{\max}$ and $T_{\min}$:*

$$T'_{\max}(i) := \begin{cases} T_{\max}(i) & \text{if } T_{\max}(i) = t_{\max} \\ \text{argmin}_{\substack{j \in S \\ j \ne t_{\max}}} a_j & \text{otherwise} \end{cases}$$

$$T'_{\min}(i) := \begin{cases} T_{\min}(i) & \text{if } T_{\max}(i) = t_{\max} \\ \text{argmax}_{\substack{j \in S \\ j \ne t_{\max}}} a_j & \text{otherwise} \end{cases}$$

*That is, $T'_{\max}$ replaces $t_{\min}$ with whatever token in $S$ draws the least attention away from $t_{\max}$, while $T'_{\min}$ replaces $t_{\min}$ with whichever token in $S$ draws the most attention away from $t_{\max}$.*

*Define* relaxed extremal sequence scores $r_{T_{\max}}$, $r_{T_{\min}}$:

$$r_{T_{\min}} := \min_{0 \le i < n_{\text{ctx}}} w_i + \left( \sum_{0 \le i < n_{\text{ctx}}} v_{T_{\min}(i)} e^{a_{T'_{\min}(i)} + b_i} \middle/ \sum_{0 \le i < n_{\text{ctx}}} e^{a_{T'_{\min}(i)} + b_i} \right)$$

$$r_{T_{\max}} := \max_{0 \le i < n_{\text{ctx}}} w_i + \left( \sum_{0 \le i < n_{\text{ctx}}} v_{T_{\min}(i)} e^{a_{T'_{\max}(i)} + b_i} \middle/ \sum_{0 \le i < n_{\text{ctx}}} e^{a_{T'_{\max}(i)} + b_i} \right)$$

*Then* $r_{T_{\min}} \le s'_{T_{\min}}$ *and* $s'_{T_{\max}} \le r_{T_{\max}}$.

*Proof.* (sketch) Essentially the same as the proof of Theorem 6. $\qquad\qquad\square$

Note that in practice, we take $S$ to be the set of all tokens less than $t_{\max} - g$ for some minimum gap $g$. This allows us to share computation across the various maximum tokens to reduce overall computational complexity.

---

**Algorithm 5** Counting Correct Sequences in Subcubic Time, Preliminaries

---

1: **function** INPUT-SEQUENCE-COMPATIBLE-WITH(input-sequence, $d_{\text{vocab}}$, $n_{\text{ctx}}$, $t_{\max}$, $t_{\text{query}}$, $c$, $g$)
2:      $\mathbf{t} \leftarrow$ input-sequence
3:      **return** False **if** $\mathbf{t} \notin (\mathbb{N}_{<d_{\text{vocab}}})^{n_{\text{ctx}}}$             ▷ the sequence is not made of valid tokens
4:      **return** False **if** $t_{-1} \neq t_{\text{query}}$                            ▷ wrong query token
5:      **return** False **if** $\max_i t_i \neq t_{\max}$                        ▷ wrong max token
6:      **return** False **if** $|\{i \in \mathbb{N}_{<n_{\text{ctx}}} \mid t_i \neq t_{\max}\}| \neq c$     ▷ wrong count of non-max toks
7:      **return** ALL($t_i = t_{\max}$ **or** $t_{\max} - t_i \geq g$ **for** $0 \le i < n_{\text{ctx}}$)     ▷ check gap on non-max toks
8: **end function**
9: **function** CORRECTNESS-PESSIMIZING-OVER-GAP-SLOW($\mathcal{M}$, $d_{\text{vocab}}$, $n_{\text{ctx}}$, $t_{\max}$, $t_{\text{query}}$, $c$, $g$)
10:      **return** ALL(CORRECTNESS($\mathcal{M}$, $\mathbf{t}$) **for all** $\mathbf{t}$ s.t. INPUT-SEQUENCE-COMPATIBLE-WITH($\mathbf{t}$, $d_{\text{vocab}}$, $n_{\text{ctx}}$, $t_{\max}$, $t_{\text{query}}$, $c$, $g$))
11: **end function**
12: **function** SUBCUBIC($d_{\text{vocab}}$, $n_{\text{ctx}}$, $\mathcal{M}$, $G$)
13:      count $\leftarrow 0$                                       ▷ # of correct sequences
14:      $G_{t_{\max}, t_{\text{query}}, c} \leftarrow \text{MIN}(t_{\max}, \text{MAX}(1, G_{t_{\max}, t_{\text{query}}, c}))$      ▷ Clip $G$ to valid range
15:      $G^*_{t_{\max}, c} \leftarrow \min_{t \le t_{\max}} G_{t_{\max}, t, c}$                    ▷ Cache running minima
16:      **for** $t_{\max} \in \text{RANGE}(d_{\text{vocab}})$ **do**                   ▷ $t_{\max} \leftarrow$ max-token
17:          **for** $0 \le t_{\text{query}} \le t_{\max}$ **do**                ▷ $t_{\text{query}} \leftarrow$ query-token
18:              $c_{\min} \leftarrow 0$ **if** $t_{\text{query}} = t_{\max}$ **else** $1$      ▷ minimum copies of nonmax
19:             $c_{\max} \leftarrow \begin{cases} 0 & \text{if } t_{\max} = 0 \\ n_{\text{ctx}} - 1 & \textbf{otherwise} \end{cases}$      ▷ maximum copies of nonmax
20:             **for** $c_{\min} \le c \le c_{\max}$ **do**         ▷ valid choices for the number of non-max tokens
21:                 $g \leftarrow G_{t_{\max}, t_{\text{query}}, c}$
22:                 $g^* \leftarrow G^*_{t_{\max}, c}$
23:                 q-gap $\leftarrow t_{\max} - t_{\text{query}}$
24:                 $\text{RCPOG}(\vec{\chi}) \leftarrow$ RELAXED-CORRECTNESS-PESSIMIZING-OVER-GAP($\mathcal{M}$, $\vec{\chi}$)
25:                 **if** (q-gap $= 0$ **or** q-gap $\ge g$) **and** $\text{RCPOG}(d_{\text{vocab}}, n_{\text{ctx}}, t_{\max}, t_{\text{query}}, c, g, g^*)$ **then**
26:                     $c' \leftarrow c$ **if** $t_{\text{query}} = t_{\max}$ **else** $c - 1$      ▷ # of non-max non-query tokens
27:                     count $\leftarrow$ count $+ \binom{n_{\text{ctx}}-1}{c'}(t_{\max} - g)^{c'}$     ▷ taking $0^0 = 1$ conventionally
28:                 **end if**
29:             **end for**
30:          **end for**
31:      **end for**
32:      **return** count $\cdot \frac{1}{d_{\text{vocab}}^{n_{\text{ctx}}}}$
33: **end function**

---

---

**Algorithm 6** Counting Correct Sequences in Subcubic Time

---

1: **function** MODEL-BEHAVIOR-RELAXED-OVER-GAP($\mathcal{M}, t_{\max}, t_{\text{query}}, c, g, g^*$)
**Ensure:** CORRECTNESS-PESSIMIZING-OVER-GAP-SLOW is False $\implies$ result is False
**Require:** $0 \leq g^* \leq g \leq t_{\max}$
**Require: if** $c = 0$ **then** $t_{\text{query}} = t_{\max}$
2:      skip-score $\leftarrow \max_{t^*} \ell^{\text{EU}}(t_{\text{query}})_{t^*} - \min_{t^*} \ell^{\text{EU}}(t_{\text{query}})_{t^*}$         $\triangleright$ Cache by $t_{\text{query}}$
3:      $v_t \leftarrow \text{EVOU}(t)$
4:      $w_i \leftarrow \text{PVOU}(i)$
5:      $\Delta w_{\max,t^*} \leftarrow \max_i w_{i,t^*} - w_{i,t_{\max}}$         $\triangleright$ Cache by $t_{\max}, t^*$
6:      $\Delta w_{\max,\max} \leftarrow \max_{t^*} \Delta w_{\max,t^*}$         $\triangleright$ Cache by $t_{\max}$
7:      $\Delta v_t \leftarrow \max_{t^*} v_{t,t^*} - \min_{t^*} v_{t,t^*}$         $\triangleright$ Cache by $t$
8:      $\Delta v_{\max} \leftarrow \max_{0 \leq t \leq t_{\max} - g^*} \Delta v_t$         $\triangleright$ Cache by $t_{\max} - g^*$
9:      $\Delta v_{t^*}^{t_{\max}} \leftarrow v_{t_{\max},t^*} - v_{t_{\max},t_{\max}}$         $\triangleright$ Cache by $t_{\max}$
10:     $\Delta v_{\max}^{t_{\max}} \leftarrow \max_{t^* \neq t_{\max}} \Delta v_{t^*}^{t_{\max}}$         $\triangleright$ Cache by $t_{\max}$
11:     **if** $c = 0$ **then**
12:         $\ell_{t^*} \leftarrow \ell^{\text{EU}}(t_{\max})_{t^*} + v_{t_{\max},t^*} + \Delta w_{\max,t^*}$
13:         **return** $\max_{t^* \neq t_{\max}} (\ell_{t^*} - \ell_{t_{\max}})$
14:     **end if**
15:     $b_{:,n_{\text{ctx}}-1} \leftarrow \text{EQKP}(t_{\text{query}}, n_{\text{ctx}} - 1)/\sqrt{d}$         $\triangleright$ Cache by $t_{\text{query}}$
16:     $b_{0,:-1} \leftarrow \text{SORT}(\text{EQKP}(t_{\text{query}}, :-1))/\sqrt{d}$         $\triangleright$ Cache by $t_{\text{query}}, i$
17:     $b_{1,:-1} \leftarrow \text{REVERSE}(b_{0,:-1})$
18:     $a_t \leftarrow \text{EQKE}(t_{\text{query}}, t)/\sqrt{d}$         $\triangleright$ Cache by $t_{\text{query}}, t$
19:     $a_{\min,t} \leftarrow \min_{0 \leq t'' \leq t} a_{t''}$         $\triangleright$ Cache by $t_{\text{query}}, t$, compute in amortized $\mathcal{O}(d_{\text{vocab}}^2)$
20:     $a_{\max,t} \leftarrow \max_{0 \leq t'' \leq t} a_{t''}$         $\triangleright$ Cache by $t_{\text{query}}, t$, compute in amortized $\mathcal{O}(d_{\text{vocab}}^2)$
21:     $\Delta a_{\max} \leftarrow a_{t_{\max}} - a_{\min,t_{\max} - g}$         $\triangleright$ Cache by $t_{\text{query}}, t_{\max}, c$
22:     $\Delta a_{\min} \leftarrow a_{t_{\max}} - a_{\max,t_{\max} - g}$         $\triangleright$ Cache by $t_{\text{query}}, t_{\max}, c$
23:     idx-set $\leftarrow \{0, \ldots, n_{\text{ctx}} - c - 1\}$ **if** $t_{\max} \neq t_{\text{query}}$ **else** $\{0, \ldots, n_{\text{ctx}} - c - 2, n_{\text{ctx}} - 1\}$
24:     attn-weights-unscaled$_{0,i} \leftarrow b_{0,i} + (\Delta a_{\min}$ **if** $i \in$ idx-set **else** $0)$
25:     attn-weights-unscaled$_{1,i} \leftarrow b_{1,i} + (\Delta a_{\max}$ **if** $i \in$ idx-set **else** $0)$   $\triangleright$ Cache by $t_{\text{query}}, t_{\max}, i, c$
26:     attn-weights$_0 \leftarrow \text{SOFTMAX}(\text{attn-weights-unscaled}_0)$         $\triangleright$ Cache by $t_{\text{query}}, t_{\max}, i, c$
27:     attn-weights$_1 \leftarrow \text{SOFTMAX}(\text{attn-weights-unscaled}_1)$         $\triangleright$ Cache by $t_{\text{query}}, t_{\max}, i, c$
28:     attn-max$_0 \leftarrow \sum_{i \in \text{idx-set}} \text{attn-weights}_{0,i}$
29:     attn-max$_1 \leftarrow \sum_{i \in \text{idx-set}} \text{attn-weights}_{1,i}$
30:     attn-max $\leftarrow$ attn-max$_0$ **if** $\Delta v_{\max}^{t_{\max}} < \Delta v_{\max}$ **else** attn-max$_1$
31:             $\triangleright$ Recall that $\Delta v_{\max}^{t_{\max}}$ is negative when the model outputs the correct answer
32:     **return** skip-score $+ \Delta w_{\max,\max} +$ attn-max $\cdot \Delta v_{\max}^{t_{\max}} + (1 - \text{attn-max})\Delta v_{\max}$
33: **end function**
34: **function** RELAXED-CORRECTNESS-PESSIMIZING-OVER-GAP($\mathcal{M}, d_{\text{vocab}}, n_{\text{ctx}}, t_{\max}, t_{\text{query}}, c,$ $g, g^*$)
35:         $\triangleright$ runs the model on a relaxed variant of input sequences compatible with the arguments
**Ensure:** CORRECTNESS-PESSIMIZING-OVER-GAP-SLOW is False $\implies$ result is False
**Ensure: return** is False **if** CORRECTNESS-PESSIMIZING-OVER-GAP-SLOW($\mathcal{M}, \mathbf{t}$) is False **for** *any*
    $\mathbf{t}$ with specified $t_{\max}, t_{\text{query}}$, and $c$ tokens not equal to $t_{\max}$
36:     **return** MODEL-BEHAVIOR-RELAXED-OVER-GAP($\mathcal{M}, t_{\max}, t_{\text{query}}, c, g, g^*$) $< 0$
37: **end function**

---

**Theorem 16.** *For all G,*

$$\mathbb{E}_{\mathbf{t} \sim U(0,1,\ldots,d_{\text{vocab}}-1)^{n_{\text{ctx}}}} \left[ \underset{i}{\text{argmax}}(\mathcal{M}(\mathbf{t}))_i = \max_i t_i \right] \geq \text{SUBCUBIC}(d_{\text{vocab}}, n_{\text{ctx}}, \mathcal{M}, G)$$

*Proof.* (sketch) Apply preceding lemmas and theorems to Algorithm 6 □

**Theorem 17.** *The running time of Algorithm 6, after using caching to avoid duplicate computations, is* $\mathcal{O}(d_{\text{vocab}}{}^2 d_{\text{model}} + d_{\text{vocab}}{}^2 n_{\text{ctx}}{}^2)$.

*Proof.* (sketch) Sum the complexities indicated along the right side of Algorithm 3. The $d_{\text{vocab}}{}^2 d_{\text{model}}$ term comes from the precomputing EVOU, EU, and EQKP. The $d_{\text{vocab}}{}^2 n_{\text{ctx}}{}^2$ term comes from the softmax over $n_{\text{ctx}}$ tokens for $O(d_{\text{vocab}}{}^2 n_{\text{ctx}})$ pessimized pure sequences. Confirming that none of the complexities on the right side exceeds $\mathcal{O}(d_{\text{vocab}}{}^2 d_{\text{model}} + d_{\text{vocab}}{}^2 n_{\text{ctx}}{}^2)$ completes the proof.

□

# G  Subcubic proof strategies

In this section, we present a number of proof strategies that we use to reduce the computational cost of the proof, ultimately driving down the cost of EU and EQKE verification to $\mathcal{O}(d_{\text{vocab}} d_{\text{model}})$, while unfortunately leaving the cost of EVOU verification at $\mathcal{O}(d_{\text{vocab}}{}^2 d_{\text{model}})$.

The three main tricks we cover are the mean+diff trick (Appendix G.1), the max row-diff trick (Appendix G.2.2), and the rank one / rank two SVD decomposition of EQKE (Appendix G.2.3). While the mean+diff trick is useful for getting slightly better bounds, the SVD decomposition of EQKE is the place where we get to insert the most understanding (without which we'd have no hope of non-vacuous bounds below $\mathcal{O}(d_{\text{vocab}}{}^2 d_{\text{model}})$), and the max row-diff trick is the workhorse that allows us to drive down the error term computations from cubic to quadratic without getting completely vacuous bounds.

## G.1  The mean+diff trick

Suppose we have quantities $f_{x,y}$ and $g_{y,z}$ and we want to pessimize (WLOG, suppose minimize) the quantity $f_{x,y} + g_{y,z}$ over $x$, $y$, and $z$ in time less than $\mathcal{O}(n_x n_y n_z)$, say we allow $\mathcal{O}(n_x n_y + n_y n_z + n_x n_z)$. Also suppose the variation of $f$ over the $y$ axis is much larger than the variation of f over the x-axis.

We can of course say

$$\min_{x,y} f_{x,y} + \min_{y,z} g_{y,z} \leq f_{x,y} + g_{y,z}$$

But we can do better!

Note that

$$f_{x,y} = \mathbb{E}_x f_{x,y} + (f_{x,y} - \mathbb{E}_x f_{x,y})$$

Suppose that $f_{x,y}$ varies much less over $x$ than it does over $y$, and much less than $g_{y,z}$ varies over either of $y$ and $z$. This will make the following bound a good approximation, though the bound is sound even without this assumption. We can write

$$\begin{aligned} f_{x,y} + g_{y,z} &\geq \min_{x,y,z}[f_{x,y} + g_{y,z}] \\ &= \min_{x,y,z}[\mathbb{E}_x f_{x,y} + g_{y,z} + f_{x,y} - \mathbb{E}_x f_{x,y}] \\ &\geq \min_{x,y,z}[\mathbb{E}_x f_{x,y} + g_{y,z}] + \min_{x,y,z}[f_{x,y} - \mathbb{E}_x f_{x,y}] \\ &= \min_{y,z}[\mathbb{E}_x f_{x,y} + g_{y,z}] + \min_{x,y}[f_{x,y} - \mathbb{E}_x f_{x,y}] \end{aligned}$$

By averaging the variation over certain axes, we have

**Theorem 18** (Mean+Diff).

$$\min_{x,y,z} f_{x,y} + g_{y,z} \geq \min_{y,z}[\mathbb{E}_x f_{x,y} + g_{y,z}] + \min_{x,y}[f_{x,y} - \mathbb{E}_x f_{x,y}]$$

$$\max_{x,y,z} f_{x,y} + g_{y,z} \leq \max_{y,z}[\mathbb{E}_x f_{x,y} + g_{y,z}] + \max_{x,y}[f_{x,y} - \mathbb{E}_x f_{x,y}]$$

*and the RHSs can be computed in time* $\mathcal{O}(n_x n_y + n_y n_z + n_x n_z)$ *for* $n_x$, $n_y$, *and* $n_z$ *the number of possible values of* $x$, $y$, *and* $z$, *respectively.*

Example for how this helps with small variation:

Take any function $k(y)$ and then take

$$f_{x,y} := k(y) + \varepsilon_1(x,y)$$
$$g_{y,z} := -k(y) + \varepsilon_2(y,z)$$

Then we have

$$\min_{x,y,z}[f_{x,y} + g_{y,z}] = \min_{x,y,z}[\varepsilon_1(x,y) + \varepsilon_2(y,z)]$$

$$\min_{x,y} f_{x,y} + \min_{y,z} g_{y,z} = \min_y k(y) + \min_y -k(y) + \min_{x,y} \varepsilon_1(x,y) + \min_{y,z} \varepsilon_2(y,z)$$

$$= \min_y k(y) - \max_y k(y) + \min_{x,y} \varepsilon_1(x,y) + \min_{y,z} \varepsilon_2(y,z)$$

$$\min_{x,y}[f_{x,y} - \mathbb{E}_x f_{x,y}] + \min_{y,z}[g_{y,z} + \mathbb{E}_x f_{x,y}] = \min_{x,y} \varepsilon_1(x,y) + \min_{y,z}[\varepsilon_2(y,z) + \mathbb{E}_x \varepsilon_1(x,y)]$$

If $\varepsilon_1$ and $\varepsilon_2$ are small compared to $\min_y k(y) - \max_y k(y)$, then using $\mathbb{E}_x f_{x,y}$ gives a much better bound.

Note, though, that this could be a worse bound if the assumption of small variation does not hold.

Note also that this trick is not restricted to adding and subtracting $\mathbb{E}_x f_{x,y}$. If $f$ is a matrix indexed by $x$ and $y$, we might also try taking SVD and using the first principal component instead. A basic application of the triangle inequality gives the following, more general, result:

**Theorem 19** (Summarize+Diff). *For any* $h_y$ *which can be computed in time* $\mathcal{O}(n_h)$,

$$\min_{x,y,z} f_{x,y} + g_{y,z} \geq \min_{y,z}[h_y + g_{y,z}] + \min_{x,y}[f_{x,y} - h_y]$$

$$\max_{x,y,z} f_{x,y} + g_{y,z} \leq \max_{y,z}[h_y + g_{y,z}] + \max_{x,y}[f_{x,y} - h_y]$$

*and the RHSs can be computed in time* $\mathcal{O}(n_x n_y + n_y n_z + n_h)$ *for* $n_x$, $n_y$, *and* $n_z$ *the number of possible values of* $x$, $y$, *and* $z$, *respectively.*

We see that if the variation of $f$ in the $x$-axis is indeed much smaller than the variation in the $y$-axis, then letting

$$f_{x,y} = h_y + \varepsilon_{x,y}$$

we get

$$\left| \min_{x,y,z} f_{x,y} + g_{y,z} - \min_{y,z}[h_y + g_{y,z}] - \min_{x,y}[f_{x,y} - h_y] \right|$$

$$\leq \left| \min_{x,y,z}[f_{x,y} + g_{y,z}] - \min_{y,z}[h_y + g_{y,z}] \right| + \left| \min_{x,y}[\varepsilon_{x,y}] \right|$$

$$\leq 2 \max_{x,y} |\varepsilon_{x,y}|$$

so indeed this bound isn't too much worse and we are able to compute it in quadratic rather than cubic time.

### G.2  Details of SVD of QK proof

As discussed in , to further reduce the computation cost of proof, we need to avoid computing the residual stream, EVOU, and EPQKE matrices fully. Using mechanistic insight or otherwise, we observe that these matrices (apart from EVOU) can be well-approximated by rank one matrices. This will remove the dominant computation cost of $\mathcal{O}(d_{\text{vocab}}^2 \cdot d_{\text{model}})$.

### G.2.1 Comments on relationship between mechanistic insight and proof size

Up to this point, we haven't really said much in our proofs about what the model is doing. All the mechanistic insight has been of the form "the model varies more along this axis than this other axis" or "the input data is distributed such that handling these inputs is more important than handling these other inputs" or, at best, "the model computes the answer by attending to the maximum token of the sequence; everything else is noise".

Here, finally, our proof-size constraints are tight enough that we will see something that we could plausibly call "how the model pays attention to the maximum token more than anything else", i.e., (if we squint a bit) "the model pays more attention to larger tokens in general.

### G.2.2 The max row-diff trick

As stated above, we are breaking matrices into their rank one approximation and some error term. To bound the error, i.e. to bound expressions of the form $\prod_i (A_i + E_i) - \prod_i A_i$, where $E_i$ denote the matrix errors, we can use the following trick:

**Lemma 20** (Max Row-Diff (vector-matrix version)). *For a row vector $\mathbf{a}$ and a matrix $B$,*

$$\max_{i,j} \left( (\mathbf{a}B)_i - (\mathbf{a}B)_j \right) \leq \sum_k |a_k| \max_{i,j} \left( B_{k,i} - B_{k,j} \right)$$

*Moreover, for a collection of $n$ row vectors $A_r$, if the shape of $B$ is $m \times p$, the right hand side can be computed for all $r$ in time $\mathcal{O}(nm + mp)$.*

*Proof.*

$$\max_{i,j} (\mathbf{a}B)_i - (\mathbf{a}B)_j$$

$$= \max_{i,j} \sum_k a_k \left( B_{k,i} - B_{k,j} \right)$$

$$\leq \sum_k \max_{i,j} a_k \left( B_{k,i} - B_{k,j} \right)$$

$$= \sum_k a_k \begin{cases} \max_{i,j} \left( B_{k,i} - B_{k,j} \right) & \text{if } a_k \geq 0 \\ \min_{i,j} \left( B_{k,i} - B_{k,j} \right) & \text{if } a_k < 0 \end{cases}$$

$$= \sum_k a_k \begin{cases} \max_{i,j} \left( B_{k,i} - B_{k,j} \right) & \text{if } a_k \geq 0 \\ -\max_{i,j} \left( B_{k,i} - B_{k,j} \right) & \text{if } a_k < 0 \end{cases}$$

$$= \sum_k |a_k| \max_{i,j} \left( B_{k,i} - B_{k,j} \right)$$

The asymptotic complexity of computing the result follows from caching the computation of $\max_{i,j} \left( B_{k,i} - B_{k,j} \right)$ for each $k$ independently of $r$, as the computation does not depend on $A_r$. □

**Theorem 21** (Max Row-Diff). *For matrices $A$ and $B$,*

$$\max_{r,i,j} \left( (AB)_{r,i} - (AB)_{r,j} \right) \leq \max_r \sum_k |A_{r,k}| \max_{i,j} \left( B_{k,i} - B_{k,j} \right)$$

*Proof.* By taking the max of Lemma 20 over rows $r$ of $A$. □

Lemma 20 can also be applied recursively for a product of more than two matrices.

**Lemma 22** (Max Row-Diff (vector-matrix recursive version)). *For a row vector $\mathbf{a}$ and a sequence of $n$ matrices $B_p$ of shapes $r_p \times c_p$,*

$$\max_{i,j} \left( \left( \mathbf{a} \prod_p B_p \right)_i - \left( \mathbf{a} \prod_p B_p \right)_j \right) \leq \sum_{k_0} |a_{k_0}| \cdots \sum_{k_n} \left| (B_{n-1})_{k_{n-1}, k_n} \right| \max_{i,j} \left( (B_n)_{k_n, i} - (B_n)_{k_n, j} \right)$$

*Moreover, for a collection of $q$ row vectors $A_\alpha$, the right hand side can be computed for all $\alpha$ in time $\mathcal{O}(qr_0 + \sum_p r_p c_p)$.*

*Proof.* We proceed by induction on $n$.

For $n = 1$, the statement is identical to Lemma 20.

Suppose the theorem holds for all positive $n = s$; we show the theorem holds for $n = s + 1$. We reassociate the matrix multiplication as

$$\max_{i,j} \left( \left( \mathbf{a} \prod_{p=1}^{s+1} B_p \right)_i - \left( \mathbf{a} \prod_{p=1}^{s+1} B_p \right)_j \right)$$

$$= \max_{i,j} \left( (\mathbf{a}B_1) \left( \left( \prod_{p=2}^{s+1} B_p \right)_i - \left( \prod_{p=2}^{s+1} B_p \right)_j \right) \right)$$

Using the induction hypothesis gives

$$\leq \sum_{k_1} \left| \sum_{k_0} a_{k_0} (B_1)_{k_0,k_1} \right| \sum_{k_2} |(B_2)_{k_1,k_2}| \cdots \sum_{k_{s+1}} |(B_s)_{k_s,k_{s+1}}| \max_{i,j} \left( (B_{s+1})_{k_{s+1},i} - (B_{s+1})_{k_{s+1},j} \right)$$

The triangle inequality gives

$$\leq \sum_{k_1} \sum_{k_0} |a_{k_0} (B_1)_{k_0,k_1}| \sum_{k_2} |(B_2)_{k_1,k_2}| \cdots \sum_{k_{s+1}} |(B_s)_{k_s,k_{s+1}}| \max_{i,j} \left( (B_{s+1})_{k_{s+1},i} - (B_{s+1})_{k_{s+1},j} \right)$$

and algebra gives

$$= \sum_{k_0} |a_{k_0}| \sum_{k_1} |(B_1)_{k_0,k_1}| \sum_{k_2} |(B_2)_{k_1,k_2}| \cdots \sum_{k_{s+1}} |(B_s)_{k_s,k_{s+1}}| \max_{i,j} \left( (B_{s+1})_{k_{s+1},i} - (B_{s+1})_{k_{s+1},j} \right)$$

The asymptotic complexity of computing the right hand side also follows straightforwardly by induction. $\qquad\square$

**Theorem 23** (Max Row-Diff (recursive)). *For a sequence of $n + 1$ matrices $A_0, \ldots, A_n$,*

$$\max_{r,i,j} \left( \left( \prod_p A_p \right)_{r,i} - \left( \prod_p A_p \right)_{r,j} \right) \leq \max_r \sum_{k_0} |(A_0)_{r,k_0}| \cdots \sum_{k_n} |(A_{n-1})_{k_{n-1},k_n}| \max_{i,j} \left( (A_n)_{k_n,i} - (A_n)_{k_n,j} \right)$$

*Proof.* By taking the max of Lemma 22 over rows $r$ of $A_0$. $\qquad\square$

Note that Theorem 21 is compatible with the mean+diff trick of Appendix G.1.

**Theorem 24** (Combined Mean+Diff and Max Row-Diff). *For matrices A and B, and any column-wise summary vector $H_k$ of A (for example we may take $H_k := \mathbb{E}_r A_{r,k}$)*

$$\max_{r,i,j} \left( (AB)_{r,i} - (AB)_{r,j} \right) \leq \left( \max_{i,j} \sum_k H_k (B_{k,i} - B_{k,j}) \right) + \max_r \sum_k |A_{r,k} - H_k| \max_{i,j} (B_{k,i} - B_{k,j})$$

*Proof.*

$$\max_{r,i,j} \left( (AB)_{r,i} - (AB)_{r,j} \right)$$

$$= \max_{r,i,j} \sum_k A_{r,k} (B_{k,i} - B_{k,j})$$

$$= \max_{r,i,j} \sum_k (H_k + (A_{r,k} - H_k)) (B_{k,i} - B_{k,j})$$

$$= \max_{i,j} \left( \sum_k H_k (B_{k,i} - B_{k,j}) + \max_r \sum_k (A_{r,k} - H_k) (B_{k,i} - B_{k,j}) \right)$$

$$\leq \left( \max_{i,j} \sum_k H_k \left( B_{k,i} - B_{k,j} \right) \right) + \max_r \sum_k \max_{i,j} \left( A_{r,k} - H_k \right) \left( B_{k,i} - B_{k,j} \right)$$

$$\leq \left( \max_{i,j} \sum_k H_k \left( B_{k,i} - B_{k,j} \right) \right) + \max_r \sum_k |A_{r,k} - H_k| \max_{i,j} \left( B_{k,i} - B_{k,j} \right)$$

$\square$

**Theorem 25** (Combined Mean+Diff and Vector-Matrix Recursive Max Row-Diff). *For a row vector* **a**, *a vector of summaries* **h** *corresponding to* **a** *(for example, if* **a** *is a row of a matrix,* **h** *might be the average of the rows), a sequence of $n$ matrices $B_p$ of shapes $r_p \times c_p$, and a corresponding sequence of column-wise summary vectors* $\mathbf{h_p}$ *of $B_p$ (for example we may take $(h_p)_k := \mathbb{E}_r(B_p)_{r,k}$),*

$$\max_{i,j} \left( \left( \mathbf{a} \prod_p B_p \right)_i - \left( \mathbf{a} \prod_p B_p \right)_j \right)$$

$$\leq \max_{i,j} \left( \sum_{k_0} h_{k_0} \cdots \sum_{k_n} (B_{n-1})_{k_{n-1},k_n} \left( (B_n)_{k_n,i} - (B_n)_{k_n,j} \right) \right)$$

$$+ \sum_{k_0} |a_{k_0} - h_{k_0}| \cdots \left( \left( \max_{i,j} \sum_{k_n} (h_{n-1})_{k_n} \left( (B_n)_{k_n,i} - (B_n)_{k_n,j} \right) \right) \right.$$

$$\left. + \sum_{k_n} \left| (B_{n-1})_{k_{n-1},k_n} - (h_{n-1})_{k_n} \right| \max_{i,j} \left( (B_n)_{k_n,i} - (B_n)_{k_n,j} \right) \right)$$

*Moreover, for a collection of $q$ row vectors $A_\alpha$, the right hand side can be computed for all $\alpha$ in time $\mathcal{O}(q r_0 + \sum_p r_p c_p)$.*

*Proof sketch.* Apply the triangle inequality recursively, fusing the proofs of Lemmas 22, and 24. $\square$

**Theorem 26** (Combined Mean+Diff and Recursive Max Row-Diff). *For a sequence of $n+1$ matrices $A_0, \ldots, A_n$, and corresponding column-wise summary vectors $\mathbf{h_0}, \ldots, \mathbf{h_{n-1}}$ of $A_0, \ldots, A_{n-1}$,*

$$\max_{r,i,j} \left( \left( \prod_p A_p \right)_{r,i} - \left( \prod_p A_p \right)_{r,j} \right)$$

$$\leq \max_{r,i,j} \left( \sum_{k_0} (h_0)_{k_0} \cdots \sum_{k_n} (A_{n-1})_{k_{n-1},k_n} \left( (A_n)_{k_n,i} - (A_n)_{k_n,j} \right) \right)$$

$$+ \sum_{k_0} |(A_0)_{k_0} - (h_0)_{k_0}| \cdots \left( \left( \max_{i,j} \sum_{k_n} (h_{n-1})_{k_n} \left( (A_n)_{k_n,i} - (A_n)_{k_n,j} \right) \right) \right.$$

$$\left. + \sum_{k_n} \left| (A_{n-1})_{k_{n-1},k_n} - (h_{n-1})_{k_n} \right| \max_{i,j} \left( (A_n)_{k_n,i} - (A_n)_{k_n,j} \right) \right)$$

*Moreover, if the matrices $A_p$ have shapes $r_p \times c_p$, the right hand side can be computed in time $\mathcal{O}(\sum_p r_p c_p)$.*

*Proof.* By taking the max of Theorem 25 over rows $r$ of $A_0$. $\square$

### G.2.3 Exploring rank one approximation via SVD

Let us first look at

$$\text{EQKE} := \mathbf{E}_q Q K^T \bar{\mathbf{E}}^T.$$

From Figure 9a, we see that there is not much variation along long query token direction. We can confirm this by performing a singular value decomposition (SVD) on EQKE, as seen in Figure 14.

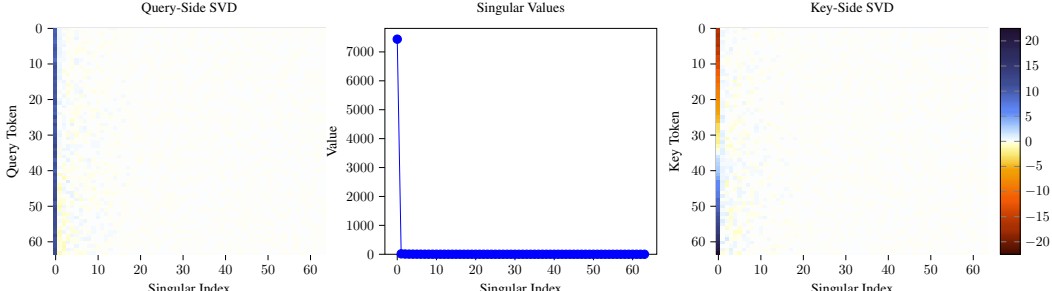

**Figure 14:** SVD of EQKE for seed 123, with principal component vectors scaled by the square root of the corresponding singular value. This scaling allows us to see visually that there is not much going on beyond the first singular component. Numerically: the first singular value is just over 7440, while the second singular value is just under 15.

The first singular value is just over 7440 ($7800 \pm 380$ across all seeds), while the second singular value is just under 15 ($13.1 \pm 2.8$ across all seeds). The ratio across all seeds is $620 \pm 130$. There's really not much going on here beyond the first singular component.[23]

Call the first singular component of EQKE the "query direction" $d_q$ and the "size direction" $d_k$ on the query-side and key-side, respectively.

There are two ways that we can decompose EQKE into a low-rank component that we can compute exactly, and a full-rank error term that we approximate bounds for.

### G.2.4  The simple SVD decomposition of QK

In time $\mathcal{O}(d_{\text{vocab}}d_{\text{model}}{}^2)$ we can perform SVD on each of the four component matrices $\mathbf{E}_q$, $Q$, $K$, $\bar{\mathbf{E}}$ and perform low-rank SVD on the matrix product $\mathbf{E}_q Q K^T \bar{\mathbf{E}}^T$.

We can then bound the difference between two elements in the same row of EQKE by computing exactly the difference between the two elements in the same row of the rank one approximation of EQKE, and adding to that a bound on the difference between the two elements in the same row of the error term.

That is, we can decompose $E$ into a part parallel to $d_q$ and a part orthogonal to $d_q$, say $\mathbf{E}_q = E_q + E_q^{\perp}$, and similarly $\bar{\mathbf{E}} = E_k + E_k^{\perp}$. Note that $E_q$ and $E_k$ are both rank one, and hence can be multiplied with other matrices of shape $d_{\text{model}} \times a$ in time $\mathcal{O}(d_{\text{model}}a)$ rather than time $\mathcal{O}(d_{\text{vocab}}d_{\text{model}}a)$.

Hence we can define EQKE_err$_1$ (subscript one for "rank one") and decompose EQKE as

$$\text{EQKE} = E_q Q K^T (E_k)^T + \text{EQKE\_err}_1.$$

Define for any vector $v$

$$\Delta_{i,j} v := v_i - v_j$$

so that we get

$$\Delta_{i,j}(E_q Q K^T (E_k)^T)_{t_{\text{query}}} + \min_{i \neq j} \Delta_{i,j}(\text{EQKE\_err}_1)_{t_{\text{query}}}$$
$$\leq \Delta_{i,j}\text{EQKE}_{t_{\text{query}}} \leq$$
$$\Delta_{i,j}(E_q Q K^T (E_k)^T)_{t_{\text{query}}} + \max_{i \neq j} \Delta_{i,j}(\text{EQKE\_err}_1)_{t_{\text{query}}}$$

Then we may use any method we please to pessimize $\Delta_{i,j}(\text{EQKE\_err}_1)_{t_{\text{query}}}$ quickly. For example, since for any matrix $M$ we have $\sigma_1(M) = \sup_x \|Mx\| / \|x\|$, considering vectors with one 1, one

---

[23]We might be tempted to keep analyzing the SVD, and notice that the query direction is mostly uniform, while the key direction is monotonic (nearly linear, even). But the proof complexity doesn't demand this level of analysis, yet, and so we can't expect that any automated compact proof discovery system will give it to us.

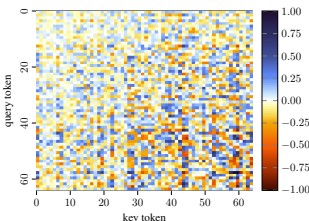

**Figure 15:** The error term EQKE_err for seed 123.

$-1$, and zero elsewhere, the maximum difference between elements in a row upper bounded by $\sqrt{2}\sigma_1(M)$:

$$\left|\Delta_{i,j}\text{EQKE\_err}_{1\,t_{\text{query}}}\right| \leq \sqrt{2}\sigma_1(\text{EQKE\_err}_1) \tag{11}$$

### G.2.5 The complicated SVD decomposition of QK

While the "most mechanistic" interpretation would proceed with the analysis in terms of $E_q$ and $E_k$, perhaps decomposing them further, we can get more bang for our buck by extracting out all the low-rank structure available $E$, $Q$, and $K$, so as to make our error bounds as tight as possible.

To this end, we perform SVD on $E_q^\perp$, $E_k^\perp$, $Q$, and $K$ and peel off the first singular components so as to get the decomposition

$$\mathbf{E}_q = E_q + E_{q,2} + E_{q,2}^\perp$$
$$\bar{\mathbf{E}} = E_k + E_{k,2} + E_{k,2}^\perp$$
$$Q = Q_0 + Q^\perp$$
$$K = K_0 + K^\perp$$

Then EQKE, a product of these four matrices, can be expressed as a sum of $2^2 3^2 - 1 = 35$ rank one products and one high-rank error term. We can compute the sum of the rank one products in time $\mathcal{O}(d_{\text{vocab}}^2)$ and express EQKE as, say, $\text{EQKE}_2 + E_{q,2}^\perp Q^\perp (E_{k,2}^\perp K^\perp)^T$. Call the second term EQKE_err (Figure 15). We must now bound for each $q$ and $m$ the quantity $\max_{i \leq m-G} \text{EQKE\_err}[q,i] - \text{EQKE\_err}[q,m]$.

How big is this?

Even if we relax to $\max_{i,j} \text{EQKE\_err}[q,i] - \text{EQKE\_err}[q,j]$, the maximum such value across all rows is under 1.85 ($1.99 \pm 0.68$ across all seeds). And the rows don't have any particular structure to them; the maximum absolute element of the entire matrix is just barely over 1 ($1.12 \pm 0.40$ across all seeds), so doubling that doesn't give too bad an estimate.

But we somehow need to compute this value without multiplying out the four matrices.

One option is to try to use singular value decomposition again. Since $\sigma_1(M) = \sup_x \|Mx\| / \|x\|$, considering vectors with one 1, one $-1$, and zero elsewhere, the maximum difference between elements in a row upper bounded by $\sqrt{2}\sigma_1(M)$. The largest singular value of EQKE_err (Figure 16) is just under 7.6 ($8.4 \pm 2.0$ across all seeds), giving a row-diff bound of about 10.7 ($11.8 \pm 2.8$ across all seeds), which is large but not unusably so.

If we perform SVD before multiplying out the matrices (Figure 17), however, their first singular values are about 4, 1.4, 1.4, and 4, giving a product of about 30, which when multiplied by $\sqrt{2}$ is about 43. (Across all seeds, these numbers are $3.79 \pm 0.12$, $1.525 \pm 0.067$, $1.513 \pm 0.073$, and $3.78 \pm 0.12$, giving a product of about $33.1 \pm 2.9$, which when multiplied by $\sqrt{2}$ is about $46.8 \pm 4.2$.) This works because $\sigma_1(AB) \leq \sigma_1(A)\sigma_1(B)$, but note that we can do factored SVD without needing to use this technique. This bound is still usable, but pretty big.

**Can we use Frobenius?** Note that using anything close to this method to drop below $d_{\text{vocab}}d_{\text{model}}^2$ might seem infeasible (it'll eventually turn out not to be). For example, the best bound we know on the largest singular value that can be verified even in the worst-case in strictly less time than

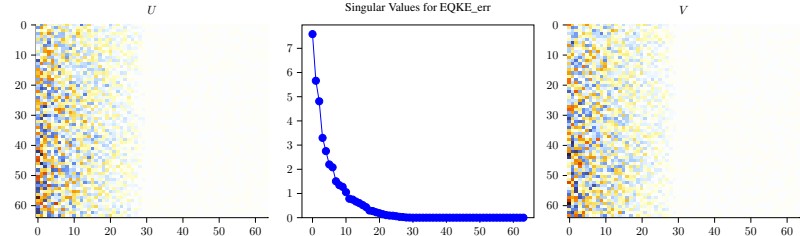

**Figure 16:** SVD of EQKE_err for seed 123.

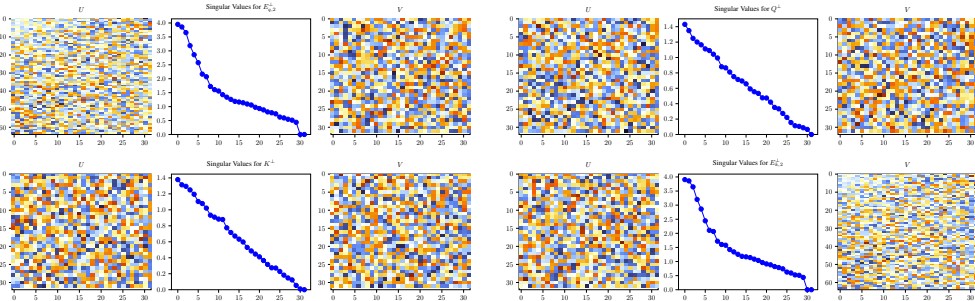

**Figure 17:** SVD of the four component matrices of EQKE_err for seed 123. Matrices look like noise.

it takes to compute the full SVD is the Frobenius norm, which is defined as $\text{tr}(MM^T)$, can be computed in $d_{\text{model}}d_{\text{vocab}}$ time, and is equal to the square root of the sum of the squares of the singular values. While the Frobenius norm of EQKE_err is only about 12 (giving a bound of about 17 on the row-diff), the Frobenius norms of the four multiplicand matrices are a bit over 10, 4, 4, and 10, giving a product of 1932 and a bound of 2732(!). (Across all seeds, the Frobenius norm of EQKE_err is about $13.1 \pm 1.9$ (giving a bound of about $18.6 \pm 2.7$ on the row-diff), the Frobenius norms of the four multiplicand matrices are a bit over $9.92 \pm 0.19$, $4.43 \pm 0.01$, $4.361 \pm 0.095$, and $9.85 \pm 0.19$, giving a product of $1888 \pm 99$ and a bound of $2670 \pm 140$.) This is unusably large.

However, we can get a much better bound on the max row-diff of EQKE_err without having to multiply out all four matrices. We can use an approach vaguely similar to the mean+diff trick, as follows.

If we want to compute the max row-diff of a product of matrices $AB$, we can compute by Theorem 21

$$\max_{r,i,j}\left((AB)_{r,i} - (AB)_{r,j}\right) \leq \max_r \sum_k |A_{r,k}| \max_{i,j}\left(B_{k,i} - B_{k,j}\right) \tag{12}$$

or by combining this approximation with Theorem 18 via Theorem 24 we may compute

$$\max_{r,i,j}\left((AB)_{r,i} - (AB)_{r,j}\right)$$
$$\leq \left(\max_{i,j} \sum_k \mathbb{E}_r A_{r,k}\left(B_{k,i} - B_{k,j}\right)\right) + \max_r \sum_k |A_{r,k} - \mathbb{E}_r A_{r,k}| \max_{i,j}\left(B_{k,i} - B_{k,j}\right)$$

taking whichever bound is better.

The first gives us a bound of 7.94 on the maximum row-diff, which is better than we can get by doing SVD on the product of the matrices! We can get an even better bound by peeling off the first two singular values of all four matrices before multiplying them; this gives us a bound of 5.67. Combining it with the avg+diff trick wouldn't give us much (8.05 and 5.66 respectively), as we've effectively already done this by peeling off the leading singular contributions; the mean of EQKE_err over dimension zero has norm 0.025 ($0.030 \pm 0.012$ across all seeds).

Although this error bound is no longer the leading asymptotic bottleneck, we can peek ahead to what we get if we want to be linear in parameter count. In this case, we can apply the recursive version of Equation 12 via Theorem 23, giving a bound of 97.06 on the maximum row-diff.

The mechanistic understanding we get here is roughly "for any given basis vector of the residual stream, the difference between the overlap of any two input tokens with this direction is small once we factor out the first two singular components", and this is sufficient to drive a low error term overall if we factor out the leading singular components in other places. We don't mechanistically understand how to combine the $\mathbf{E}_q Q K^T$ (without multiplying them out) in a way that allows getting a good bound, though, which corresponds to our inability to drop below $d_{\mathrm{vocab}} d_{\mathrm{model}}^2$ here.

If we use this trick on QK only, and use the mean+diff trick on final attention handling (without which we lose about $19\,\%$), we can achieve a bound of $0.7840$ ($0.661 \pm 0.035$ across all seeds).

If we use this trick on the skip connection (EU) only, we can achieve a bound of $0.6768$ ($0.632 \pm 0.061$ across all seed).

Using this trick on both EU and QK drops us down only to $0.6354$ ($0.601 \pm 0.060$ across all seeds).

If we use this trick on EU and use the recursive version of this trick on QK, we get a bound of $0.2927$ ($0.281 \pm 0.036$ across all seeds).

Unfortunately, it's not clear how this trick would apply to EVOU. A fancier convex hull checking algorithm seems required, and an analysis thereof is in progress.

## G.3    The algorithm

We now put all of these tricks together into the subcubic algorithm Algorithm 7, which is the full version of Algorithm 6. The format we give here is parameterized over the summarization strategy (from Theorem 19 in Appendix G.1), the decomposition of EQKE, and the handling of EQKE_err and EU.

**Algorithm 7** Counting Correct Sequences in Subcubic Time

---

1: **function** MODEL-BEHAVIOR-RELAXED-OVER-GAP($\mathcal{M}, t_{\max}, t_{\text{query}}, c, g, g^*$)

**Ensure:** CORRECTNESS-PESSIMIZING-OVER-GAP-SLOW is False $\implies$ result is False

**Require:** $0 \leq g^* \leq g \leq t_{\max}$

**Require:** **if** $c = 0$ **then** $t_{\text{query}} = t_{\max}$

2: $\quad \overline{\text{skip-score}}_{t^*} \leftarrow \text{SUMMARIZE}_{\text{EU},t_{\text{query}}}(\ell^{\text{EU}}(t_{\text{query}})_{t^*})$ $\hfill \triangleright$ Cache by $t^*$

3: $\quad \text{skip-score} \leftarrow \max_{t^*} \ell^{\text{EU}}(t_{\text{query}})_{t^*} - \min_{t^*} \ell^{\text{EU}}(t_{\text{query}})_{t^*}$ $\hfill \triangleright$ Cache by $t_{\text{query}}$

4: $\quad v_t \leftarrow \text{EVOU}(t)$

5: $\quad w_i \leftarrow \text{PVOU}(i)$

6: $\quad \Delta w_{\max,t^*} \leftarrow \max_i w_{i,t^*} - w_{i,t_{\max}}$ $\hfill \triangleright$ Cache by $t_{\max}, t^*$

7: $\quad \Delta w_{\max,\max} \leftarrow \max_{t^*} \Delta w_{\max,t^*}$ $\hfill \triangleright$ Cache by $t_{\max}$

8: $\quad \Delta v_t \leftarrow \max_{t^*} v_{t,t^*} - \min_{t^*} v_{t,t^*}$ $\hfill \triangleright$ Cache by $t$

9: $\quad \Delta v_{\max} \leftarrow \max_{0 \leq t \leq t_{\max} - g^*} \Delta v_t$ $\hfill \triangleright$ Cache by $t_{\max} - g^*$

10: $\quad \Delta v_{t^*}^{t_{\max}} \leftarrow v_{t_{\max},t^*} - v_{t_{\max},t_{\max}}$ $\hfill \triangleright$ Cache by $t_{\max}$

11: $\quad \Delta v_{\max}^{t_{\max}} \leftarrow \max_{t^* \neq t_{\max}} \Delta v_{t^*}^{t_{\max}}$ $\hfill \triangleright$ Cache by $t_{\max}$

12: $\quad$ **if** $c = 0$ **then**

13: $\quad\quad \ell_{t^*} \leftarrow \ell^{\text{EU}}(t_{\max})_{t^*} + v_{t_{\max},t^*} + \Delta w_{\max,t^*}$

14: $\quad\quad$ **return** $\max_{t^* \neq t_{\max}} (\ell_{t^*} - \ell_{t_{\max}})$

15: $\quad$ **end if**

16: $\quad b_{:,n_{\text{ctx}}-1} \leftarrow \text{EQKP}(t_{\text{query}}, n_{\text{ctx}} - 1)$ $\hfill \triangleright$ Cache by $t_{\text{query}}$

17: $\quad b_{0,:-1} \leftarrow \text{SORT}(\text{EQKP}(t_{\text{query}}, :-1))$ $\hfill \triangleright$ Cache by $t_{\text{query}}, i$

18: $\quad b_{1,:-1} \leftarrow \text{REVERSE}(b_{0,:-1})$

19: $\quad \text{EQKE}^{(1)}, \text{EQKE\_err} \leftarrow \text{DECOMPOSE}(\text{EQKE})$

**Require:** $\text{EQKE}^{(1)}(t_{\text{query}}, t) - \text{EQKE}^{(1)}(t_{\text{query}}, t_{\max}) - \text{EQKE\_err}_{t_{\text{query}}} \leq \text{EQKE}(t_{\text{query}}, t) - \text{EQKE}(t_{\text{query}}, t_{\max}) \leq \text{EQKE}^{(1)}(t_{\text{query}}, t) - \text{EQKE}^{(1)}(t_{\text{query}}, t_{\max}) + \text{EQKE\_err}_{t_{\text{query}}}$

20: $\quad a_t \leftarrow \text{EQKE}^{(1)}(t_{\text{query}}, t)$ $\hfill \triangleright$ Cache by $t_{\text{query}}, t$

21: $\quad a_{\min,t} \leftarrow \min_{0 \leq t'' \leq t} a_{t''}$ $\hfill \triangleright$ Cache by $t_{\text{query}}, t$, compute in amortized $\mathcal{O}(d_{\text{vocab}}^2)$

22: $\quad a_{\max,t} \leftarrow \max_{0 \leq t'' \leq t} a_{t''}$ $\hfill \triangleright$ Cache by $t_{\text{query}}, t$, compute in amortized $\mathcal{O}(d_{\text{vocab}}^2)$

23: $\quad \Delta a_{\max} \leftarrow a_{t_{\max}} - a_{\min,t_{\max}-g} + \text{EQKE\_err}_{t_{\text{query}}}$ $\hfill \triangleright$ Cache by $t_{\text{query}}, t_{\max}, c$

24: $\quad \Delta a_{\min} \leftarrow a_{t_{\max}} - a_{\max,t_{\max}-g} - \text{EQKE\_err}_{t_{\text{query}}}$ $\hfill \triangleright$ Cache by $t_{\text{query}}, t_{\max}, c$

25: $\quad \text{idx-set} \leftarrow \{0, \ldots, n_{\text{ctx}} - c - 1\}$ **if** $t_{\max} \neq t_{\text{query}}$ **else** $\{0, \ldots, n_{\text{ctx}} - c - 2, n_{\text{ctx}} - 1\}$

26: $\quad \text{attn-weights-unscaled}_{0,i} \leftarrow b_{0,i} + (\Delta a_{\min}$ **if** $i \in \text{idx-set}$ **else** $0)$

27: $\quad \text{attn-weights-unscaled}_{1,i} \leftarrow b_{1,i} + (\Delta a_{\max}$ **if** $i \in \text{idx-set}$ **else** $0)$ $\hfill \triangleright$ Cache by $t_{\text{query}}, t_{\max}, i, c$

28: $\quad \text{attn-weights}_0 \leftarrow \text{SOFTMAX}(\text{attn-weights-unscaled}_0/\sqrt{d})$ $\hfill \triangleright$ Cache by $t_{\text{query}}, t_{\max}, i, c$

29: $\quad \text{attn-weights}_1 \leftarrow \text{SOFTMAX}(\text{attn-weights-unscaled}_1/\sqrt{d})$ $\hfill \triangleright$ Cache by $t_{\text{query}}, t_{\max}, i, c$

30: $\quad \text{attn-max}_0 \leftarrow \sum_{i \in \text{idx-set}} \text{attn-weights}_{0,i}$

31: $\quad \text{attn-max}_1 \leftarrow \sum_{i \in \text{idx-set}} \text{attn-weights}_{1,i}$

32: $\quad \text{attn-max} \leftarrow \text{attn-max}_0$ **if** $\Delta v_{\max}^{t_{\max}} \geq \Delta v_{\max}$ **else** $\text{attn-max}_1$

33: $\quad \overline{\text{attn-max}} \leftarrow \text{SUMMARIZE}_{\text{attn},t_{\text{query}}}(\text{attn-max})$ $\hfill \triangleright$ Cache by $t_{\max}, c$

34: $\quad \text{attn-max}' \leftarrow \text{attn-max} - \overline{\text{attn-max}}$

35: $\quad \text{summary}_{t^*} \leftarrow \Delta w_{\max,t^*} + \overline{\text{skip-score}}_{t^*} + \overline{\text{attn-max}} \Delta v_{t^*}^{t_{\max}} + (1 - \overline{\text{attn-max}}) \Delta v_{\max}$ $\hfill \triangleright$ Cache by $t_{\max}, t^*$

36: $\quad$ **return** $\text{skip-score} + \text{attn-max}' \cdot \Delta v_{\max}^{t_{\max}} + (-\text{attn-max}')\Delta v_{\max} + \max_{t^* \neq t_{\max}} \text{summary}_{t^*}$

37: **end function**

# H Comparison of proof strategies

In this section, we compare the various proof strategies that we have developed in Appendix G. We do some traditional mechanistic interpretability analysis to justify that the choices that we made could be expected to lead to reasonably good bounds in Appendix H.1. We then compare the complexities and performance of various proof strategies in Appendix H.2 to line up with the legends of Figures 3, and 4. We close with a figure relating the various categories of proof strategies.

## H.1 Justification of pessimization choices

In Sections 4.3, F, and G we make a number of choices about which axes of variation are more or less important to track at various points in the bound computation.

Here we do some more traditional mechanistic interpretability analysis to justify that the choices that we made could be expected to lead to reasonably good bounds.

### H.1.1 Justifying the gap

We take advantage of the fact that attention is mostly monotonically increasing in input integers and that for most sequences, the attentional contribution of the particular query token matters much more than the particular non-max token in the sequence.

We justify this as follows.

We can look at the typical diff, when attending to the max token, between the largest non-max logit and the max logit. As shown in Figure 18a, the largest difference between an off-diagonal entry of EVOU and the diagonal of that row is typically at most $-7$.[24] The typical worst contribution to the wrong logit from a non-max token (this is typical over non-max tokens, worst over choice of output token-logit index) is around $43$, as shown in Figure 18b.

The difference in attention between tokens is approximately linear in the gap between the tokens, as seen in Figure 19. The slope of the line, that is, the difference in pre-softmax attention scores divided by the gap between the key token and the max token, is approximately $1.2$.

Exponentiating, the post-softmax attention paid to the max is typically about $3\times$ larger than to the token one below the max; here the logit difference between the max and non-max token is significant, typically being around $13$ $(43/3)$ for the worst output logit. But by the time the gap is 3, this difference has dropped to about $1.1$, and by the time the gap is 4 it is around $0.3$.

---

[24]"Typically" here means about $96\,\%$ of the time.

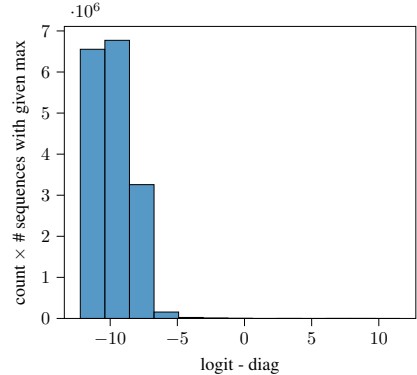
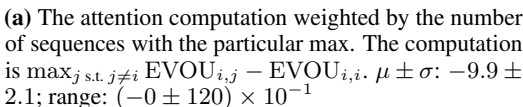
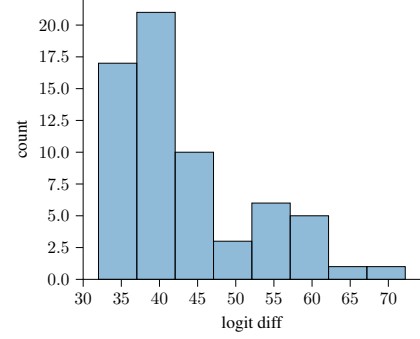

**(a)** The attention computation weighted by the number of sequences with the particular max. The computation is $\max_{j \text{ s.t. } j \neq i} \text{EVOU}_{i,j} - \text{EVOU}_{i,i}$. $\mu \pm \sigma$: $-9.9 \pm 2.1$; range: $(-0 \pm 120) \times 10^{-1}$

**(b)** Histogram of the maximum difference between two logits contributed by a single row of EVOU. The computation is, for each $i$, $\max_h \text{EVOU}_{i,j} - \min_j \text{EVOU}_{i,j}$. $\mu \pm \sigma$: $43.4 \pm 9.0$; range: $52 \pm 20$

**Figure 18:** Plots of the difference in logit for the attention computation, $\text{EVOU} := \bar{\text{E}}VOU$ for seed 123.

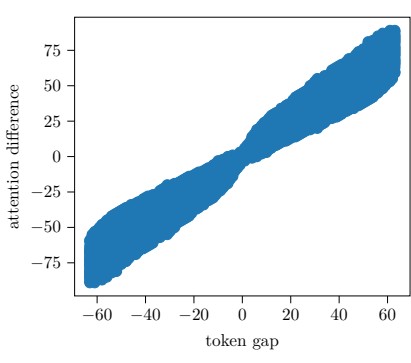
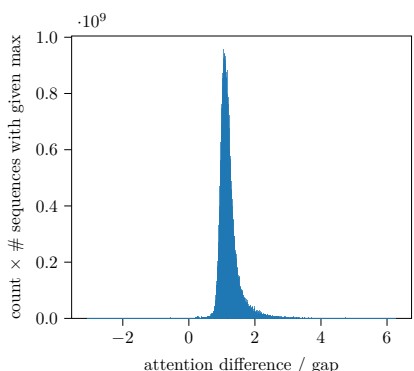

**(a)** $(\mathrm{EQKE}_i - \mathrm{EQKE}_j)/\sqrt{d}$ vs. $i - j$

**(b)** $(\mathrm{EQKE}_i - \mathrm{EQKE}_j)/(\sqrt{d}(i - j))$, weighted by sequence count. $\mu \pm \sigma = 1.22 \pm 0.13$

**Figure 19:** Plots of attention difference vs. token gap, for $\mathrm{EQKE} := \mathbf{E}_q Q K^T \bar{\mathbf{E}}^T$ for seed 123. The difference in attention between tokens is approximately linear in the gap between the tokens.

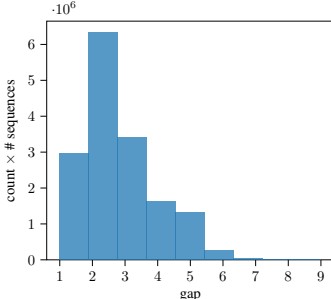

**Figure 20:** Histogram of the minimum gap between the max token and the largest non-max token, for the seed 123.

So for sequences where the largest non-max and the max are close together, the particular structure of the non-max EVOU matters a lot; but when the max is separated from the largest non-max by a modest gap, the structure of the non-max EVOU does not matter so much.

The upshot is that to handle most sequences, we need only ask an oracle for the minimum gap $g > 0$ between the max token $t_{\max}$ and largest non-max tokens $t' \neq t_{\max}$, such that the model outputs the correct answer for all sequences where the non-max, non-query tokens have value at most $t_{\max} - g$.

While computing this gap may be expensive (and indeed the naïve computation of the oracle takes longer than the brute-force proof—though it should be very easy to optimize), we don't have to pay the cost of computing the gap in the size of the proof, only the cost of storing the gap table ($\mathcal{O}(d_{\mathrm{vocab}}{}^2 n_{\mathrm{ctx}})$) and of verifying the gap. Empirically, gaps are typically 1–5, as seen in Figure 20.

If we rely on the gaps, this results in leaving behind about $6.9\,\%$ of sequences.

**Picking up more sequences** In this paragraph / bulleted list, we sketch out how we might go about picking up more sequences to get a tighter bound. This is not coded up, and is left as future work. We propose computing the following quantities:

- First, we could build in time ($\mathcal{O}(d_{\mathrm{vocab}}{}^2)$) a table indexed on pairs $(t, t_{\max})$ of the maximum token and a non-maximum token: the table would store pessimal logit contributions from $t$ to maximum output tokens $\leq$ the $t_{\max}$ parameter. The table could be further split to pessimize separately for tokens within and outside of the gap window.

- Compute a table of pre-softmax attention differences between tokens $t$ and $t + 1$ in time ($\mathcal{O}(d_{\mathrm{vocab}}{}^2)$).

- Next sort the queries by overlap with the query direction.

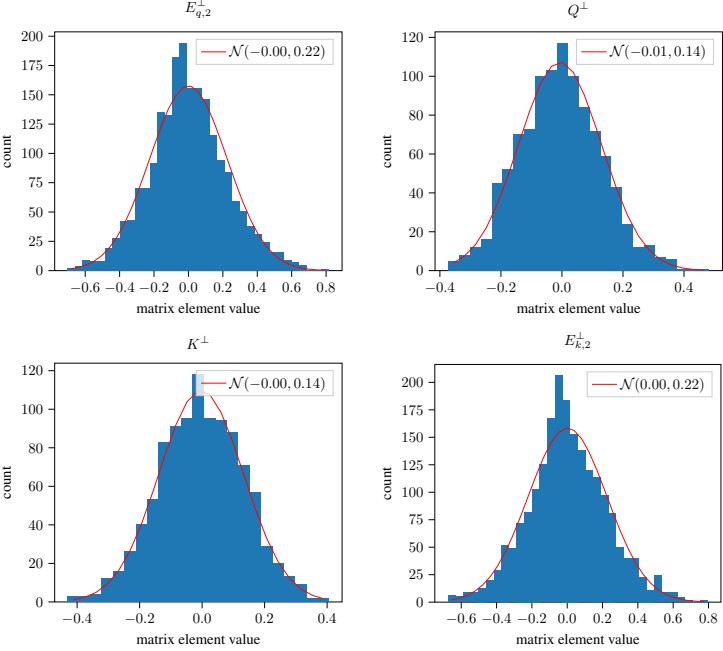

**Figure 21:** The distribution of entries of the four residual matrices (after removing two principal components from $\mathbf{E}_q$ and $\bar{\mathbf{E}}$ and one principal component from $Q$ and $K$). Distributions look pretty close to normal. Plots are for the seed 123.

- Compute for each number of queries handled (where we assume we handle all queries with greater overlap than the current one) and for each maximum input token $t_{\max}$, how many of the query tokens $t_{\text{query}}$ fall strictly below the max $t_{\max}$ (and whether or not the model succeeds when $t_{\max} = t_{\text{query}}$). This will tell us how many query tokens we can count for a given maximum token.
- Compute a table indexed on pairs of # of queries handled and input tokens $t$ which stores the smallest difference in more attention paid to $t + 1$ than to $t$ ($\mathcal{O}(d_{\text{vocab}}{}^2)$).
- Compute a table indexed on pairs $t_{\max}, t$ storing an upper bound on amount more attention paid to non-maximum tokens than to $t_{\max}$ by Oracle-permitted query tokens (the Oracle is indexed only on $t_{\max}$) ($\mathcal{O}(d_{\text{vocab}}{}^2)$).
- For each # queries permitted: compute for each $t_{\max}$, $t$, $c$, if the non-maximum token $t$ contributes little enough to incorrect logits that even with the worst skip connection the model still gets the correct answer.

### H.1.2 Stopping after 1–2 principal components of QK

Did we miss out on any structure in the error term of EQKE? The distribution of entries of the four matrices looks pretty close to normal as seen in Figure 21.

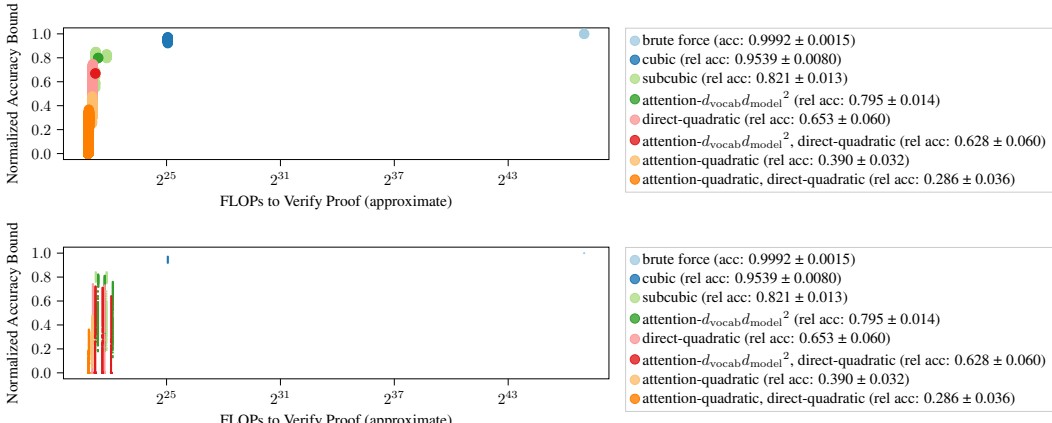

**Figure 22:** Recreations of Figure 3 for ease of viewing of the legend. Top is a strict recreation; bottom includes points not on the Pareto frontier.

If we replace the entries of $E_{q,2}^{\perp}$, $E_{k,2}^{\perp}$, $Q^{\perp}$, and $K^{\perp}$ with randomly sampled values, we get (sample size 100) that the maximum row-diff of the product of the matrices is approximately $1.31 \pm 0.13$ (sampling without replacement from the empirical distribution) or $1.31 \pm 0.14$ (sampling from the normal distribution). So in fact our max row-diff is unusually high (by about $4\sigma$).[25]

### H.2  How various combinations of tricks perform

Recall Figures 3 and 4 on page 8 and on page 9 from Section 5, recapitulated here without captions for convenience as Figures 22, and 23.

We describe what each subcubic proof strategy in the legend means. Note that all subcubic proof strategies (that is, all proof strategies except for "brute force" and "cubic") use the quadratic counting algorithm of Appendix F.

#### H.2.1  Proof strategies grouped by complexity

In Figures 3, and 22, proof strategies are grouped by computational complexity.

The 102 proof strategies break down into $1 + 1 + 2 \times 5 \times 10 \times 2$ strategies.

The **brute force** and **cubic** proofs $(1 + 1)$ were fully covered in Appendices D, and E.

There are 5 options for handling EU:

**direct-quadratic** refers to handling EU in time $\mathcal{O}(d_{\mathrm{vocab}}d_{\mathrm{model}})$ with either the max row-diff trick (Appendix G.2.2)[26] or the max row-diff trick fused with mean+diff or some other summary statistic (Theorem 24)[27].

When **direct** is not mentioned, this indicates that we handle EU in time $\mathcal{O}(d_{\mathrm{vocab}}^2 d_{\mathrm{model}})$ by first multiplying out $\mathbf{E}_q U$ and then either taking the maximum row-diff in each row[28] or by taking the maximum row-diff across all rows[29]. The latter is included purely for comparison's sake, and never gives a tighter bound than the former.

There are 10 options for handling the high-rank attention error term EQKE_err:

---

[25]This shows up in the bias towards having larger values (both positive and negative) in the lower-right corner of the plot, indicating that errors are larger for larger query and key values. We hypothesize that this is due to the distribution of data: larger values are more likely to have more space between the maximum and next-most-maximum token, so a bit of noise matters less for larger maxes than for smaller ones.

[26]This strategy is labeled "`max_diff`" in the Python source code.

[27]These strategies are labeled "`mean_query+max_diff`" and "`svd_query+max_diff`" in the Python source code

[28]"`max_diff_exact`"

[29]"`global_max_diff_exact`"

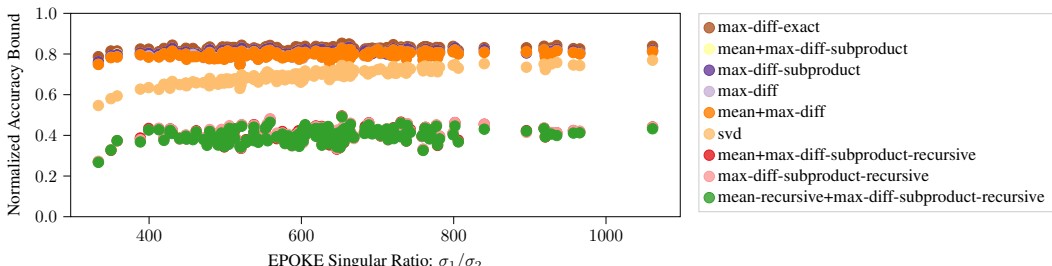

**Figure 23:** Recreation of Figure 4 for ease of viewing of the legend.

**attention-quadratic** refers to handling the high-rank attention error term EQKE_err from Appendix G.2.5 in time $\mathcal{O}(d_{\text{vocab}} d_{\text{model}})$ either with the recursive max row-diff trick (Theorem 23)[30] or with the recursive max row-diff trick fused with the mean+diff trick either just on the query side[31] or throughout[32] (Theorem 26).

**attention-$d_{\text{vocab}} d_{\text{model}}^2$** indicates that we use one of the various $\mathcal{O}(d_{\text{vocab}} d_{\text{model}}^2)$ strategies for handling EQKE_err$_1$ from Appendix G.2.4 or EQKE_err from Appendix G.2.5. These include using $\sqrt{2}\sigma_1$—computed via low-rank SVD—as the bound (Equation 11)[33], considering all ways of multiplying out a subset of the matrices and taking the maximum row-diff of the resulting pair of matrices[34] (Theorem 21), or fusing the max row-diff trick with the mean+diff trick[35] (Theorem 24).

When **attention is not mentioned**, this indicates that we handle the attention error term in time $\mathcal{O}(d_{\text{vocab}}^2 d_{\text{model}})$, either by taking the per-row maximum row-diff[36] or by using the full rank EQKE matrix and taking the per-row maximum row diff[37].

Finally, note that in combining the rank one attention computation with EVOU, PVOU, and EU, we may either use the mean+diff trick[38] (Appendix G.1) or not[39]; this makes up the final factor of 2.

### H.2.2    Proof strategies grouped by attention handling

This section slightly reorganizes the information just covered in Appendix H.2.1, for convenience of legend correspondence. Here we group by the strategy used to handle the attention error term. Strategies that involve using the full rank EQKE matrix are elided. The dashed descriptors here correspond to underscore-joined descriptors in footnotes of Appendix H.2.1.

**max-diff-exact** ($\mathcal{O}(d_{\text{vocab}}^2 d_{\text{model}})$) corresponds to taking the full rank EQKE_err$_1$ term and taking the maximum row-diff in each row.

**mean+max-diff-subproduct** ($\mathcal{O}(d_{\text{vocab}} d_{\text{model}})$) corresponds to fusing the max row-diff trick with the mean+diff trick (Theorem 24) and considering all ways of associating the multiplication of EQKE_err.

**max-diff-subproduct** ($\mathcal{O}(d_{\text{vocab}} d_{\text{model}})$) corresponds to using the max row-diff trick (Theorem 21) and considering all ways of associating the multiplication of EQKE_err.

**max-diff** ($\mathcal{O}(d_{\text{vocab}} d_{\text{model}}^2)$) corresponds to using the max row-diff trick (Theorem 21) on the factored SVD of EQKE_err$_1$.

**mean+max-diff** ($\mathcal{O}(d_{\text{vocab}} d_{\text{model}}^2)$) corresponds to fusing the max row-diff trick with the mean+diff trick (Theorem 24) and applying it on the factored SVD of EQKE_err$_1$.

---

[30]"`max_diff_subproduct_recursive`"

[31]"`mean+max_diff_subproduct_recursive`"

[32]"`mean_recursive+max_diff_subproduct_recursive`"

[33]"`svd`"

[34]"`max_diff`" for EQKE_err$_1$, "`max_diff_subproduct`" for EQKE_err

[35]"`mean+max_diff`" for EQKE_err$_1$, "`mean+max_diff_subproduct`" for EQKE_err

[36]"`max_diff_exact`"

[37]"`exact_EQKE+max_diff_exact`"

[38]"`mean_query+diff`"

[39]"`drop_average_query_per_output_logit_reasoning`"

**svd** ($\mathcal{O}(d_{\text{vocab}}d_{\text{model}}^2)$) corresponds to using $\sqrt{2}\sigma_1$—computed via low-rank SVD—as the bound (Equation 11).

**mean+max-diff-subproduct-recursive** ($\mathcal{O}(d_{\text{vocab}}d_{\text{model}})$) corresponds to handling the high-rank attention error term EQKE_err from Appendix G.2.5 with the recursive max row-diff trick fused with the mean+diff trick on the query-side only (Theorem 26, taking all but the first summary vector to be zero).

**max-diff-subproduct-recursive** ($\mathcal{O}(d_{\text{vocab}}d_{\text{model}})$) corresponds to handling the high-rank attention error term EQKE_err from Appendix G.2.5 with the recursive max row-diff trick (Theorem 23).

**mean-recursive+max-diff-subproduct-recursive** ($\mathcal{O}(d_{\text{vocab}}d_{\text{model}})$) corresponds to handling the high-rank attention error term EQKE_err from Appendix G.2.5 with the recursive max row-diff trick recursively fused with the mean+diff trick (Theorem 26).

### H.2.3   What understanding do we get from each proof strategy?

Throughout most of this paper, we talk about doing mechanistic interpretability and using understanding to allow more compact proofs to have tighter bounds. We can also look at the reverse problem: we can take a collection of proof strategies, check by brute force which strategies give the tightest bounds for each model, and ask what this implies about how that model works. We do this here.

In general, which proof methods perform best is an indication of where structure exists in the model. For example, in quadratic EU proofs, when `max_diff` performs worse than `mean_query+max_diff` and `svd_query+max_diff`, this indicates that $E$ has a relatively strong behavioral component shared across query tokens that $U$ is not that good at filtering out. Similarly, when, e.g., `mean_recursive+max_diff_subproduct_recursive` performs better than `max_diff_subproduct_recursive`, this indicates that even after removing the first one or two principle components from $\mathbf{E}_q$, $Q$, $K$, and $\bar{\mathbf{E}}$, there is still enough common structure that it is worth factoring out the mean behavior.

