# OpenReview forum: "Compact Proofs of Model Performance via Mechanistic Interpretability"
_NeurIPS.cc/2024/Conference — NeurIPS 2024 poster_

### Official Review · Reviewer_ZhNY · 2024-07-07

**Soundness:** 3
**Presentation:** 2
**Contribution:** 2
**Rating:** 6
**Confidence:** 2

**Summary:**

This paper makes progress towards the formal verification of model performance by using insights from mechanistic interpretability to reduce proof length. It presents a case study in a simplified setup: a single-layer, single-head attention-only transformer applied to a synthetic task of finding the maximum of four integers. The authors train a large number of models on this task and derive formal guarantees for their performance using different proof strategies that make use of different insights derived by mechanistically interpreting the model. The first strategy with cubic complexity, uses the model's decomposition into direct and indirect paths, and the observation that the model makes minimal use of positional embeddings. They also consider different sub-cubic proof strategies that rely on different simplifications, such as the observation the EQKE matrix is approximately rank 1, corresponding to the size of the key token. These strategies are compared to a brute-force solution in terms of proof length (estimated using the number of FLOPs) and the ratio of the certified bound (lower bound b divided by the expected value s). The authors observe a trade-off between proof length and the certified bound, noting that establishing tighter bounds is more computationally expensive. They also find that compact proofs are less faithful to model internals. Finally, the authors identify compounding noise as a challenge for proofs of global behaviour.

**Strengths:**

- The paper demonstrates great ambition by working towards provable guarantees about global model behaviour. Developing techniques to provide such guarantees is of significant interest, especially for deploying these models in safety-critical environments.
- The authors employ a variety of methods to analyse the model and cleverly use mechanistic insights to reduce proof length. To the best of my knowledge, this is the first paper that attempts to verify global model behaviour using insights from reverse-engineering the neural network.
- The proposed measure to assess mechanistic understanding (degrees of freedom that have to be evaluated using brute-force) is interesting, as current evaluations of interpretability results often rely on proxy metrics.

**Weaknesses:**

The results appear to be merely an interesting case study in a highly simplified setting. The proposed proof strategies rely on various unrealistic simplifications, including the use of attention-only transformers, and the learned algorithm for the task allows for further simplifications, such as ignoring positional embeddings. These simplifications raise doubts about the transferability of the proposed proof strategies to other settings. For example, the observation that compact proofs, which rely on various simplifying assumptions, are already less faithful in this synthetic setting suggests that obtaining compact proofs in more realistic settings would be incredibly challenging. The paper could have been significantly strengthened by either studying a more realistic setting or characterising fundamental methods to leverage mechanistic understanding for any task to improve upon brute-force search.

Minor Issues:
- The title on the paper (“Compact Proofs of Model Performance via Mechanistic Interpretability”) does not match the title in OpenReview (“Provable Guarantees for Model Performance via Mechanistic Interpretability”).
- Typo in line 95: “we denote the true maximum of the by t_{max}.”
- Line 162: one “the” too many: “… sequence x, [the] define the ….”
- Line 189: “consider” was probably meant to be “considered.”

**Questions:**

1. You only compare sequence lengths of up to four because the brute-force solution is infeasible beyond that. Could your proposed proof strategies be used to make guarantees beyond this length? I believe that demonstrating that your approach allows us to provide guarantees for settings where brute-force solutions are (nearly) impossible to derive would greatly strengthen the results.

**Limitations:**

The authors have properly discussed the limitations of their approach.

---

> ### Author Rebuttal · Authors · 2024-08-07
>
> Thank you for your feedback!
> We have also rectified the typographical issues.
>
>
> > You only compare sequence lengths of up to four because the brute-force solution is infeasible beyond that.
> Could your proposed proof strategies be used to make guarantees beyond this length? I believe that demonstrating that your approach allows us to provide guarantees for settings where brute-force solutions are (nearly) impossible to derive would greatly strengthen the results.
>
> Thanks for the suggestion, we agree that this is a serious concern.
> We have generated new results on Max-of-5, Max-of-10, and Max-of-20 (holding parameters other than n\_ctx fixed).
> We ran the cubic proof on each of these models, giving bounds of 94% (Max-of-5), 93% (Max-of-10), and 83% (Max-of-20) in under two minutes.
> For comparison, the brute-force approach on Max-of-20 would take roughly $64^{20} \cdot 20 \cdot 64 \cdot 32 \approx 10^{40}$ FLOPs and it took about $10^{23}$ FLOPs to train GPT-3.
> We will add a section on the scalability of proof strategies to larger models, which includes these experiments as validation that our strategies result in lower complexity proofs.
>
>
> > The results appear to be merely an interesting case study in a highly simplified setting.
> The proposed proof strategies rely on various unrealistic simplifications, including the use of attention-only transformers, and the learned algorithm for the task allows for further simplifications, such as ignoring positional embeddings.
> These simplifications raise doubts about the transferability of the proposed proof strategies to other settings.
> For example, the observation that compact proofs, which rely on various simplifying assumptions, are already less faithful in this synthetic setting suggests that obtaining compact proofs in more realistic settings would be incredibly challenging.
>
> Our goal in the project was to prototype verification of transformer models, lay out desiderata for verification (compactness of proof length for feasible cost, and tightness of bound for guarantees to be materially useful), and study what the key challenges to achieving the desiderata (faithfulness of interpretation, compounding error terms).
> Obstacles that arise in prototyping are likely to be even more challenging in complex, real-world settings.
> This approach is standard in the field of mechanistic interpretability: starting in simplified settings and pushing for extensive exploration [1]. Simplifications of the model architecture in our work are comparable to state of the art analysis of toy models [2]. And the challenges of decomposing even small models adequately are discussed in various papers in the field [3] [4].
>
> However, we agree that we could have taken the alternative approach of targeting the verification of a large model, which would have been an impressive result.
>
> > The authors employ a variety of methods to analyse the model and cleverly use mechanistic insights to reduce proof length. To the best of my knowledge, this is the first paper that attempts to verify global model behaviour using insights from reverse-engineering the neural network.
>
> We appreciate the highlights about our contributions! We want to add that to the best of our knowledge, we are the first to (1) verify the performance of a transformer-architecture model, (2) guarantee global performance bounds on specific models, and (3) incorporate mechanistic information to obtain shorter proofs. While we work in a simplified setting, we are excited about follow-up work building upon these milestones.
>
> > The proposed measure to assess mechanistic understanding (degrees of freedom that have to be evaluated using brute-force) is interesting, as current evaluations of interpretability results often rely on proxy metrics.
>
> We appreciate this highlight as well. Whereas current work relies upon human judgment, with ambitious projects pushing towards procedural arguments, by pushing mechanistic interpretability to formal proofs, we were able to find that degrees of freedom remain remain even when we have decent informal interpretations.
>
> Please don't hesitate to ask any further questions. We would love the chance to clarify our approach and provide details that might encourage you to increase your score!
>
> [1]:  Elhage, et al., "Toy Models of Superposition", Transformer Circuits Thread, 2022.
>
> [2]:  Elhage, et al., "A Mathematical Framework for Transformer Circuits", Transformer Circuits Thread, 2021.
>
> [3]:  Stander, D., Yu, Q., Fan, H. and Biderman, S., 2023. Grokking Group Multiplication with Cosets. *arXiv preprint arXiv:2312.06581*.
>
> [4]:  Chan, et al., "Causal Scrubbing: a method for rigorously testing interpretability hypotheses", AI Alignment Forum, 2022.

---

> > ### Comment · Reviewer_ZhNY · 2024-08-11
> >
> > Thank you for clarifying the motivation of your study and conducting the experiments on larger input spaces. I appreciate the effort and believe that these results significantly strengthen the results of the paper. After reading the author rebuttals and other reviews, I have reconsidered my initial assessment. I now believe this paper would be an interesting contribution to the conference and will increase my score accordingly.

---

> > > ### Author Response · Authors · 2024-08-12
> > >
> > > Thank you!  Please let us know if there are any other questions we can clarify.

---

### Official Review · Reviewer_vjrV · 2024-07-12

**Soundness:** 3
**Presentation:** 1
**Contribution:** 3
**Rating:** 6
**Confidence:** 4

**Summary:**

The authors apply techniques from mechanistic interpretability to ``compact-ify'' the process required to prove model performance on the max-of-$k$ task, specifically where $k=4$. Through this procedure, the authors claim that (1) there is a positive relationship between mechanisitic understanding and the length of the proof required; (2) compact proofs provide better mechanistic understanding; (3) absence of known structure/interpretations make it difficult to compactify proofs in a non-vacuous manner.

**Strengths:**

- The proposed approach demonstrates connections between two important yet distinct subfields of deep learning---interpretability and robustness.
- For this specific max-of-$k$ task, the authors reverse-engineer the circuitry and algorithm implemented by transformers to perform this task. To the best of this reviewer's knowledge, this has not been done before in existing works.
- The authors introduce neat tricks such as mean+diff, max row-diff, as well as a nice convexity approximation in the appendix to achieve compact, subcubic proofs.
- The paper provides some insight into how theorists may utilize tools from mechanistic interpretability which so far has been a purely empirical endeavor.

**Weaknesses:**

- The task the authors choose seems very contrived even for mechanistic interpretability studies. I'm not sure why specifically the max-of-$k$ task was chosen or what its significance is in the formal verification literature and the authors do not provide this context.

- The model the authors use is also very contrived (one-layer, one attention-head transformer). Even though the results are interesting, it is unclear to this reviewer how any of these methods would generalize as the task/model changes in both complexity and size, respectively. If in fact they are not meant to generalize, it is unclear to this reviewer how the insights of this paper fit into the larger conversation of robustness and interpretability.

- The authors claim that Fig. 4 shows how less faithful interpretations lead to tighter bounds. It is extremely unclear to me that there is a positive correlation between the Singular ratio and accuracy bound. The authors should perform a statistical test here based on the $R^2$ to demonstrate that there is indeed a significant correlation.

- The paper is difficult to read in the sense that there are undefined/inconsistent terms and notations. For example,
- Line 16, ''we will need compact proofs of global robustness.'' The notion of a ''compact proof'' is never explicitly defined. I am not sure if this is jargon from formal verification. Even if it is, it should have citations and some explanation for audience.
- Line 55, ''we restrict $f$ to be the 0-1 loss so our theorems bound the accuracy of the model.'' In Eq. 1, we seek to lowerbound $f$. Based on my understanding of 0-1 loss, the model is penalized 1 for incorrect classification and 0 otherwise. Is this not the opposite of what we want?
- Line 87, Line 106, inconsistent bold-facing. What is difference between $\mathbf{E}$ and $E$. Moreover, what does $\mathbf{E}_q$ mean?
- Table 1, in the expressions of the complexity cost, specifically for rows ''Low-rank $QK$'', ``Low-rank $EU$'', and ''Low-rank $QK$ &$EU$'', are the terms (EU & OV), (QK & OV), (OV) annotations or a part of the mathematical expression?
- Line 501, citation deadlink.
- Fig. 6 ''Let $d = \sqrt{d}$''?
- Line 589 and beyond, notation $\mathbb{N}^{<n_{ctx}}$ is nonstandard. I think I understand from context, but it would be nice to clearly state what you mean by this.
- Algorithm 3, 5 is very difficult to parse, break this up into multiple subroutines?

**Questions:**

- Why did the authors choose to investigate a task like max-of-$k$ instead of modular addition (or IOI, colored-objects, etc.) where circuits for more general models have been discussed extensively in related literature?
- In lines 16-18, the authors emphasize that ``we need compact proofs of global robustness.'' Neither of these terms are rigorously defined in the paper nor is there reference to related literature. Since there are many working definitions of robustness what exactly do you mean by this? Based on Eq. 1, it does not seem that these guarantees fend against out-of-distribution or adversarial examples?
- In mechanistic interpretability, researchers generally choose a specific task, then reverse engineer the mechanisms of models that perform this task well to uncover the algorithms driving these models. The authors also follow this principle when reverse engineering the max-of-$k$ task in their toy transformer. Given that these mechanistic interpretability techniques require models with almost perfect performance a priori why is this not tautological with respect to their formal verification?
- How would this method generalize to different models that implement different circuits or algorithms on the same task? It seems that the inductive biases from the model used to compactify the proofs would not hold if a different model was using a different algorithm to complete the task (also perfectly).
- In line 73, why is pessimal ablation done by simply sampling from the data distribution instead of using carefully crafted counterfactuals that is common in mechanistic interpretability [1, 2, 3]
- Throughout the paper, the authors denote the residual from the rank-1 approximation of the EQKE matrix as ''unstructured noise.'' Could the authors elaborate on why they came to this conclusion? I understand in lines 1071-1077 the residuals follow ''pretty close'' to a normal distribution, did the authors run statistical tests to rigorize this claim? Moreover, why does this imply that the noise is unstructured?
- Why is the ratio of the largest to second-largest singular value a good metric for the faithfulness of the interpretation?

[1] Jack Merullo, Carsten Eickhoff, and Ellie Pavlick. Circuit Component Reuse Across Tasks in Transformer Lan-
guage Models. In The Twelfth International Conference on Learning Representations, September 2023. URL
https://openreview.net/forum?id=fpoAYV6Wsk.

[2] Neel Nanda, Lawrence Chan, Tom Lieberum, Jess Smith, and Jacob Steinhardt. Progress measures for grokking
via mechanistic interpretability. In The Eleventh International Conference on Learning Representations,
September 2022. URL https://openreview.net/forum?id=9XFSbDPmdW.

[3] Kevin Ro Wang, Alexandre Variengien, Arthur Conmy, Buck Shlegeris, and Jacob Steinhardt. Interpretability in
the Wild: a Circuit for Indirect Object Identification in GPT-2 Small. In The Eleventh International Conference
on Learning Representations, September 2022. URL https://openreview.net/forum?id=NpsVSN6o4ul.

**Limitations:**

I am unconvinced that the procedure the authors propose can be generalized to different tasks (even of the same complexity). I think the large amount manual quantitative/qualitative analysis/engineering required to derive the compact proofs for even this trivial task should be addressed as limitations. Other than this, the limitations section is well-written.

---

> ### Author Rebuttal · Authors · 2024-08-07
>
> Thank you very much for the extremely detailed review!
> We also appreciate the list of typographic errors, which we’ve corrected in a revised draft of the paper.
>
> We’d like to start by defining what we mean by “compact proof of global robustness” by clarifying our definition on lines 48-69 of Section 2.
>
> * By “compact proof”, we’re referring to a proof that is short.
> Specifically, in our paper, we measure the length of proofs by considering the length of a trace of the computational component of the proof (as described in lines 56-61).
> We agree that our use of the concept “compactness” could be made more explicit in the paper; we’ll add additional clarification.
> As for why we want compact proofs: proofs that are too long are both infeasible to find and also infeasible to verify – hence our interest in using mechanistic understanding to generate shorter/more compact proofs.
> * By “global robustness”, we mean that the model performs correctly in expectation over an input distribution $\mathcal D$ (lines 48-52).
> This is in contrast to much of the prior work, which focuses on establishing correctness of model behavior for points close to a particular input $x$ (which we refer to as local robustness).
> We agree that the use of the word “global robustness” in the introduction is confusing, because we use “global performance” to refer to the desired form of our verification target in the title, on line 49 in Section 2, and on line 294 and 301 in Section 6.
> We’ll edit the paper to be more clear and consistent.
>
> > Based on Eq. 1, it does not seem that these guarantees fend against out-of-distribution or adversarial examples?
>
> In addition, we’d like to clarify the flexibility inherent in equation 1.
> Normally, by OOD or adversarial inputs, we suppose that there is a distribution $\mathcal D_{\textrm{in}}$ that’s used for training and (in-distribution) validation, and another distribution $\mathcal D’$ that is the deployment distribution or generated by an adversary.
> If we had knowledge of $\mathcal D’$, we could compute the expected performance from inputs sampled from $\mathcal D’$.
> Even if we don’t have exact knowledge of $\mathcal D’$, we can still define a very broad distribution $\mathcal D$ that covers possible $\mathcal D’$s.
>
> In our work, $\mathcal D$ is the distribution of all $64^4$ possible valid input sequences.
> In addition, as our proofs partition $\mathcal D$ into subdistributions, and bound the performance on each subdistribution, we can bound the model’s performance on any possible distribution over valid input sequences.
>
> We agree that we don’t adequately discuss the flexibility inherent in our definition; we’ll add clarification to sec 2 as a well as a section to the appendix discussing this in more detail.
>
> ----
>
> We believe that much of the motivation for the work (and indeed the answers to your questions) follow from the goal of producing compact proofs.
>
> > Why did the authors choose to investigate a task like max-of-$k$ instead of modular addition (or IOI, colored-objects, etc.) \[...\] ?
>
> We picked a very simple task because formal verification is quite challenging, even for Max-of-$k$, and believe that Max-of-$k$ captures some of the core difficulties that we might expect to see.
> But to address some of your specific examples:
>
> 1. **Modular addition:** In Nanda et al 2023, the authors don’t provide a mechanistic understanding of *how* the MLP layer performs multiplication – instead, they check this via brute force.
> This lack of understanding prevents us from compactifying our proof of the model’s correctness.
>
>    In follow-up work, we extend our methodology to the modular addition task.
>    However, compactifying the proof required us to reverse engineer and provide a mechanistic description of how the MLP computes its output – which is, as far as we know, not previously known in the relevant literature.
>
> 1. **IOI/colored-objects/other small circuits in LMs:** One of the concerns with using these small circuits is that they tend to only be valid on a very restricted subdistribution.
> For example, the original IOI paper considers a small set of sentences with names ordered ABBA, and the circuit metrics become invalid on as small a distributional shift as ordering the names ABAB[1].
> If we chose to study these circuits, we’d run into a dilemma: either we restrict our distribution to be small enough to check feasibly with brute force (as the authors tend to do), or we’d have to perform additional interp that advances SOTA.
> As our focus is on the formal verification side, and not on the novel interpretability side, we chose instead to pick a task where the mech interp is relatively straightforward.
>
> > Given that these mechanistic interpretability techniques require models with almost perfect performance a priori why is this not tautological with respect to their formal verification?
>
> We agree that on models where we can evaluate the behavior on *all* possible input-output pairs, we can indeed guarantee performance via the brute-force proof (lines 151-4).
> We study such a case because it allows us to have “ground truth” on the performance of the model.
>
> In general, however, mech interp is often studied with samples from a distribution – for example, in IOI, the mean ablation is computed with respect to a small handful of other sequences, and in modular addition the authors use sampling to evaluate the 5-digit addition, repeated subsequence, and skip trigram tasks.
> In cases where the bad behavior is heavy tailed, sampling may not provide a viable estimate of performance – our hope is that understanding derived from studying the model with samples will allow us to construct formal guarantees that hold in the worst case.
> We prototype this problem by providing formal guarantees on a max-of-20 transformer, where brute forcing the input distribution is infeasible and our understanding is derived via samples.
>
> [1]: Miller et al. Transformer Circuit Faithfulness Metrics are not Robust.

---

> ### Author Response · Authors · 2024-08-07
> **Rebuttal by Authors (cont.)**
>
> > How would this method generalize to different models that implement different circuits or algorithms on the same task?
>
> We agree that guarantees require taking into account the specific model – this is part of the mechanistic understanding that interp should provide.
> That being said, we believe that this specificity can be useful: for example, if a proof strategy derived for one model does not apply to a second model, this provides evidence that the two models are using different algorithms.
>
> > In line 73, why is pessimal ablation done \[...with samples\]?
>
> We apologize for the typo.
> In line 73, “with values drawn from $\mathcal{D}$” should instead read “to values over $X$”.
> That is, instead of sampling, which might not find the actual worst-case, we use the actual global worst-case values as the counterfactuals for ablation.
> The counterfactuals commonly constructed in mechanistic interpretability are attempts to estimate the typical (/average-case) behavior of a component, which is insufficient for getting a worst-case guarantee on the overall behavior of the model.
> We’ll edit the paper to fix the typo and further clarify this point.
>
> > \[T\]he authors denote the residual from the rank-1 approximation of the EQKE matrix as ''unstructured noise.'' \[...\]
>
> We apologize for our lack of clarity.
> First, by “unstructured”, we mean that insofar as structure exists, it doesn’t help the model’s performance.
> Second, “If we sample elements randomly” should read “If we replace the entries of $E_{q,2}^\perp$, $E_{k,2}^\perp$, $Q^\perp$, and $K^\perp$ with randomly sampled values”.
> In order to validate our claim that the noise contains no helpful structure, we performed this replacement 100 times and compared the max-row-diff of these samples with the diff in the original model (lines 1074-7), and found that the diff of the original model is *worse* than the mean value by about 4σ.
> (This is analogous to the standard resampling or mean ablations that mech interp work uses to check for lack of structure.)
>
> > Why is the ratio of the largest to second-largest singular value a good metric for the faithfulness of the interpretation?
>
> Our interpretation claims that $EQKE$ is approximately rank 1 (the size direction).
> If the first singular value is much larger than all other singular values, this suggests that the matrix is well-approximated by a rank-1 approximation.
> We’ll clarify this in the text.
>
> Responses to more minor points:
>
> > Line 55, ''we restrict $f$ to be the 0-1 loss so our theorems bound the accuracy of the model.'' In Eq. 1, we seek to lowerbound $f$.
>
> We had this terminology wrong, and indeed were intending to speak of 1 minus the 0-1 loss.
> We’ll correct this, and thank you for pointing it out.
>
> > Line 87, Line 106, inconsistent bold-facing.
> What is difference between $\textbf{E}$ and $E$.
> Moreover, what does $\mathbf{E}_q$ mean?
>
> The boldfacing is significant and is defined on line 107:
>
> $$\hat{\mathbf{P}} = P - \overline{\mathbf{P}} \quad \text{and} \quad \mathbf{x}\overline{\mathbf{E}} = \mathbf{x}E + \overline{\mathbf{P}} \quad \text{and} \quad \mathbf{x}\mathbf{E}_q = \mathbf{x}E + \mathbf{P}_q \quad (\text{since } h^{(0)} = \mathbf{x}\overline{\mathbf{E}} + \hat{\mathbf{P}}).$$
>
> That is, we use the boldfaced versions of $\mathbf{E}$ to denote $E$ summed together with positional embeddings.
> We’ll try to clarify this more in the text.
>
> > The authors claim that Fig. 4 shows how less faithful interpretations lead to tighter bounds.
> It is extremely unclear to me that there is a positive correlation between the Singular ratio and accuracy bound.
> The authors should perform a statistical test here based on the $R^2$ to demonstrate that there is indeed a significant correlation.
>
> We deeply apologize for not catching the bug in our code that led to an incorrect version of this figure being included in the original submission.
> We’ve included a corrected version of this figure in the accompanying pdf.
> From this figure, it is apparent that for the smallest values of $\sigma_1/\sigma_2$ (the three values below 350), this ratio is strongly correlated with proof bound.
> For the “svd” proof strategy, where $\sqrt{2}\sigma_1(\mathrm{EQKE\_err})$ is used to bound the error term, a linear regression gives an $R^2$ of about 0.71.
>
> Please don't hesitate to ask any further questions! We appreciate the thoughtful engagement, and would love the chance to clarify results that might encourage you to increase your score.

---

> > ### Comment · Reviewer_vjrV · 2024-08-12
> >
> > Thank you for your detailed clarifications. I have read both the authors' global response and the individual responses.
> >
> > The authors adequately addressed my concerns relating to (1) the motivation of studying the max-of-$k$ task, (2) the typographical errors, and (3) OOD/adversarial distributions. However, I still have some further questions:
> >
> > > In Nanda et al 2023, the authors don’t provide a mechanistic understanding of how the MLP layer performs multiplication – instead, they check this via brute force. This lack of understanding prevents us from compactifying our proof of the model’s correctness.
> >
> > I do not really understand why this would prevent you from compactifying your proofs. Based on my understanding, understanding the underlying algorithm implemented by the model through mechanistic interpretability should only give more insight into the specific relaxations that can be made. In the case of modular arithmetic (specifically see [1]), as some algorithms implemented by the models are permutation invariant with respect to the input, one could potentially ignore those during the brute-force proof: shortening your proofs.
> >
> > Why is it the case that the role of every component in the network needs to be known for its proof to be compactified?
> >
> > > For example, the original IOI paper considers a small set of sentences with names ordered ABBA, and the circuit metrics become invalid on as small a distributional shift as ordering the names ABAB[1]. If we chose to study these circuits, we’d run into a dilemma: either we restrict our distribution to be small enough to check feasibly with brute force (as the authors tend to do)
> >
> > Why would you need to restrict the distribution to be small enough to check with brute-force? Isn't the contribution of the paper the ability to achieve tight lower bounds without this type of brute-force verification?
> >
> > > We prototype this problem by providing formal guarantees on a max-of-20 transformer,
> > where brute forcing the input distribution is infeasible and our understanding is derived via samples.
> >
> > This is also mentioned in the global response. Where are the results for these experiments? Also, as $k$ or $d$ changes how are the authors verifying that the underlying circuit implemented by the model is the same, as this could affect the assumptions of all subsequent theorems/algorithms?
> >
> > Looking forward to your clarifications.
> >
> > [1] Zhiqian Zhong, Ziming Liu, Max Tegmark, and Jacob Andreas. The Clock and the Pizza: Two Stories in
> > Mechanistic Explanation of Neural Networks. In Advances in Neural Information Processing Systems, November
> > 2023. URL https://arxiv.org/pdf/2306.17844.

---

> ### Author Response · Authors · 2024-08-12
>
> Thank you for your time, effort, and engagement.
>
> > Why is it the case that the role of every component in the network needs to be known for its proof to be compactified?
>
> To be more precise: the role of every component must be known for the proof to be *asymptotically* compactified.
>
> When measuring asymptotic behavior, it is standard to consider the leading order term(s) only. If the number of layers and attention heads is fixed, the number of parameters in a component is a constant fraction of (that is, linear in) the number of parameters of the overall network. Hence if we have to use the brute-force proof on any component, we will have an asymptotic cost equal to that for performing a brute-force analysis of the entire network.
>
> As we are not able to compress the proof of a component's behavior in the absence of mechanistic understanding of the role and functioning of that component, the leading contribution to the asymptotic size will always be the components we don't understand at all.
>
> Moreover, we want to clarify that compressing components using mechanistic understanding is how we obtain compact proofs with non-vacuous bounds *at all*, even if approximation errors from the compression can eventually tank the hoped-for bound. Without integrating mechanistic understanding, we are not able to compactify the brute-force proof.
>
> > Where are the results for these experiments?
>
> Sorry, “attached” was left-over from an earlier draft of the response, and we meant to say “included”: our results were included in the sentence “the cubic proof achieves bounds of 94% (Max-of-5), 93% (Max-of-10) and 83% (Max-of-20) in under two minutes”.
>
> However, we can provide some additional results for Max-of-10: The equivalent graph to Fig. 3 for Max-of-10 has the following points:
> - importance sampling (acc: 0.9988 ± 0.0013), $1.6 \cdot 2^{85}$ FLOPs for brute force
> - cubic (rel acc: 0.915 ± 0.021), $1.4 \cdot 2^{27}$ FLOPs
> - subcubic (rel acc: 0.734 ± 0.019) $(1.49 ± 0.10) \cdot 2^{22}$ FLOPs
> - attention-$d_{\\mathrm{vocab}}d_{\\mathrm{model}}^2$ (rel acc: 0.718 ± 0.019), $(1.609 ± 0.090) \cdot 2^{22}$ FLOPs
> - direct-quadratic (rel acc: 0.657 ± 0.032), $(1.392 ± 0.013) \cdot 2^{22}$ FLOPs
> - attention-quadratic (rel acc: 0.437 ± 0.028), $(1.384 ± 0.013) \cdot 2^{22}$ FLOPs
> - attention-quadratic, direct-quadratic (rel acc: 0.398 ± 0.028) $(1.318 ± 0.014) \cdot
> 2^{22}$ FLOPs
>
> The equivalent graph to Fig. 4 has ratios $\sigma_1 / \sigma_2$ starting at around 400, and the best-fit line for the SVD proof approach has $bound ≈ 0.56 + 8.3\cdot 10^{-5} (σ₁/σ₂)$, with $r^2 \approx 0.5$.
>
> > Also, as $k$ or $d$ changes how are the authors verifying that the underlying circuit implemented by the model is the same, as this could affect the assumptions of all subsequent theorems/algorithms?
>
> We note that:
> 1. We are working with proofs as computations (lines 56--59), i.e. the theorem statement doesn't exist independently of the proof, it doesn’t presuppose a lower bound $b$ that we then prove.
> 2. Whatever bound $b$ we are able to compute is the theorem statement we have proven. Thus, the theorem template is valid regardless of what circuit is implemented by the model.
>
> The faithfulness of the interpretation used in the proof mediates the bound (Section 5.2). In this way, the tightness of bound becomes a metric for how faithful the interpretation is. Thus, the recovered bounds can be seen as a measurement of how well the interpretation for Max-of-4 generalized to the new models trained to compute Max-of-5, Max-of-10, and Max-of-20. We are clarifying this further in our writeup, and this is valuable feedback on what concepts are worth highlighting from verification.
>
> Note that if the EVOU circuit did not perform copying, or if the model did not compute its answer by paying more attention to the correct token, the bound would be vacuous. We want to highlight that in the Fig. 4 equivalent for Max-of-10, the smallest ratio $\sigma_1/\sigma_2$ that we saw was over 400, which is evidence that the interpretation that EQKE is approximately low-rank remains valid. We are making this point more clear in a new section on applicability of results to other models.

---

> > ### Author Response · Authors · 2024-08-12
> >
> > > Why would you need to restrict the distribution to be small enough to check with brute-force? Isn't the contribution of the paper the ability to achieve tight lower bounds without this type of brute-force verification?
> >
> > We want to clarify that the paper's contribution is demonstrating how mechanistic understanding (aka compression of model internals) permits us to avoid brute force enumeration. Without such understanding, obtaining a more compact proof is not feasible.
> >
> > Here are examples illustrating how existing circuits are developed on small distributions, and why extending these results to larger distributions would require additional mechanistic interpretation:
> > 1. Names: The interpretation uses a small fixed dataset of names, lacking a broader interpretation to show that all tokens in specified positions would be parsed as names. For intuition: While we might expect a model to handle "John → Jon", it might struggle with "Mary → marry". Properly enumerating the set of names would require either proving that all elements in the dataset are equivalent, or resorting to brute force evaluation (which is the limiting behavior of their random sampling approach).
> > 2. Patterns: The interpretation is restricted to the ABBA case. This means either limiting ourselves to the ABBA case similarly, or conducting additional interpretation to show how answers on the ABBA case apply to other cases like ABAB.
> >
> > If we had taken on a project performing such interpretations, that would have been very cool! However, such further interpretation would be a separate contribution from the main objectives of our study.
> >
> > Please let us know if you have any further questions or if this fails to clarify some question that you’ve asked.

---

> > > ### Comment · Reviewer_vjrV · 2024-08-13
> > >
> > > Thank you to the authors for their time and detailed response. My concerns have been addressed, and I think the methods proposed in this paper are a solid contribution. However, I believe that the presentation of the manuscript needs significant improvement including the many typographical errors, consistency of notation, explanation of the role of the data distribution, etc.
> > >
> > > I have raised my scores accordingly.

---

### Official Review · Reviewer_exsQ · 2024-07-13

**Soundness:** 3
**Presentation:** 2
**Contribution:** 2
**Rating:** 6
**Confidence:** 2

**Summary:**

The paper aims to generate formal guarantees certifying model’s performance on max of K task. Using brute force gives the bound close to true performance of the model but the complexity cost i.e. compactness is exponential in the size of the vocabulary. The paper then aims to use understanding derived from mechanistic interpretability to reduce the complexity cost while ensuring the bound remains tight. For this, they first use the fact that model is composed of OV, QK circuits and a direct path. This aids in reducing the complexity to cubic in size of vocabulary. The bound still remains tight. Then the authors utilize few observations from their setup, for instance, low rank of EQKE matrices to further reduce the complexity to sub-cubic. However, there is a sharp drop seen in the tightness of the bound as these insights are used.

**Strengths:**

1) This work can motivate future works to try to generate more compact bounds on model’s performance. This is especially important in context of alignment, where these bounds could help in providing certificates for safe performance. Although this is a very preliminary study, it attempts to make a contribution in this direction.

2) It is interesting to see how authors were able to achieve better trade-offs between complexity and tightness of bounds on model performance by using insights from mechanistic interpretability. This work is very original in this regard.

**Weaknesses:**

1) The tightness of bounds seem to degrade very quickly as additional insights from mechanistic interpretability are used. Thus it seems difficult if future works would be able to use insights from mechanistic interpretability to derive tight bounds on performance of the model.

2) The author analyze a very simple task and techniques used by authors to decrease computational complexity are specifically tailored for this task. It therefore remains unclear how insights from this work can be applied to any other tasks. It would have been great if the authors would have attempted to investigate a few other tasks which are simple (fundamental) but still very different in nature, e.g., counting, sorting, etc.

3) There isn't enough evidence to adequately support the noise hypothesis presented by the authors i.e. the noise in estimating rank-1 approximations of E, Q, K results in larger magnitude in the error term. Currently this seems based on speculation and needs more detailed investigation. Therefore it would be great if the authors could try to artificially inject noise (of varying magnitude) in these matrices and see its effect. Else, the authors should not claim this being the major cause for not so tight bounds obtained on using insights from mechanistic interpretability.

**Questions:**

I request the authors to kindly address the questions in the weaknesses section.

**Limitations:**

Yes, the authors have addressed the limitations.

---

> ### Author Rebuttal · Authors · 2024-08-07
>
> Thank you for your review!
>
> > The tightness of bounds seem to degrade very quickly as additional insights from mechanistic interpretability are used.
> Thus it seems difficult if future works would be able to use insights from mechanistic interpretability to derive tight bounds on performance of the model.
>
> We agree that the degrading tightness of bounds is an issue.
> However, note that we evaluate proofs on the additional criteria of compactness of proof length, and using information about model internals is how we obtain shorter proofs than brute force.
> Thus, some degree of mechanistic interpretation is necessary for finding compact proofs in future work.
> We address this question in more detail in the top-level rebuttal.
>
> One claim that we will add more clearly to the paper is that proof length can be used as an evaluation metric on the quality of mechanistic interpretation (see Section 5).
> The interpretations in our different proofs all appear to be sound and use the same high-level abstractions.
> However, by pushing them to rigorous formal proofs we get to see the various degrees of freedom that remain in the different interpretations.
> Therefore, we expect that future work will need mechanistic interpretations to get short proofs, and can use the gains in proof reduction as a way of distinguishing what interpretation techniques to use.
>
>
> > The author analyze a very simple task and techniques used by authors to decrease computational complexity are specifically tailored for this task.
> It therefore remains unclear how insights from this work can be applied to any other tasks.
> It would have been great if the authors would have attempted to investigate a few other tasks which are simple (fundamental) but still very different in nature, e.g., counting, sorting, etc.
>
> Verification projects are quite intensive in human-labor, and our goal with this project was to prototype verification for transformer models, and focus on studying the core challenges to its viability.
> Thus, we limited ourselves in the number of different tasks we worked on, and instead focused on deriving multiple proofs that varied the two core desiderata: compactness and bound tightness.
>
> However, we want to highlight that it should be straightforward (if labor-intensive) to generalize our work to other toy models where the model computes the answer by paying attention to a single token or position, i.e. general element retrieval, example tasks include: first element, last element, binding adjective to key phrase, etc.
> This is because our theorems in the appendices prove in full generality that any computation performed by a single attention head, when only one of the position and the token value matters, is extremized by pure sequences.
> This means that the worst-case behavior of the computation at a single token can always be found by setting all other tokens to be identical, regardless of what the task is.
> Furthermore, generalizing our convexity arguments to multiple attention heads *whose error terms can be treated independently without degrading the bound too much* is also straightforward.
>
> We address the high-level version of this question in the top-level rebuttal where we talk about the general applicability of our approach to projects on larger input distributions, different tasks, more complex architectures, and larger models.
>
>
> > There isn't enough evidence to adequately support the noise hypothesis presented by the authors i.e. the noise in estimating rank-1 approximations of E, Q, K results in larger magnitude in the error term.
> Currently this seems based on speculation and needs more detailed investigation.
> Therefore it would be great if the authors could try to artificially inject noise (of varying magnitude) in these matrices and see its effect.
> Else, the authors should not claim this being the major cause for not so tight bounds obtained on using insights from mechanistic interpretability.
>
> Thank you for pointing out our confusing use of terminology! By “noise” we meant error terms in our approximations of the model that we use to generate more compact proofs, not error terms that the model learned.
> So, we are not referring to model weights that we can ablate to improve performance bounds of our proofs.
> In the improved version of Table 1 (included in the accompanying pdf, and which will replace Table 1 in the paper), consider the difference in bound between “Low-rank QK” (0.806) and “Quadratic QK” (0.407).
> The only change between these results is in the strategy for bounding the error that comes from approximating QK as low rank.
> We provide a more detailed clarification in the top-level rebuttal.
>
> Please don't hesitate to ask any further questions! We would love the chance clarify our approach, and provide further details that might encourage you to increase your score.

---

> > ### Comment · Reviewer_exsQ · 2024-08-13
> >
> > I thank the authors for their rebuttal. Their response helps me clarify my questions, therefore I will increase my score.

---

### Official Review · Reviewer_KZFC · 2024-07-13

**Soundness:** 4
**Presentation:** 4
**Contribution:** 4
**Rating:** 7
**Confidence:** 4

**Summary:**

This paper does a detailed and careful case study of the trade-offs and design space of formal verifications for meaningful lower bounds of the accuracy of a transformer model on a chosen task:
- Specifically, a one-layer, one-head, no-MLP, no-layernorm transformer is studied (akin to the model studied in detail in "A Mathematical Framework for Transformer Circuits", https://transformer-circuits.pub/2021/framework/index.html). A crucial difference from "A mathematical framework..." is that the model here also includes positional encodings.
- The task is to output the maximal element from a context of length 4, where tokens are synthetically generated to correspond to integers in a limited range, and the model is trained (successfully) solely on this task.
- Then, several verification approaches are considered:
	- Brute force: this means running the model on every possible input and calculating the accuracy
	- "Cubic": this proof uses some ingenious ideas to avoid iterating over all 4-tuples of possible inputs, and instead only over a particular set of triples $\Xi$. Specifically, they show that given an input $x=(t_1, t_2, t_3, t_4)$, then, *if* the model is 100% accurate on a certain subset of $\Xi$ derived from $x$, the model will also be correct on $x$.
	- "Sub-cubic": even more ingenious/compact proof strategies are possible if one uses mechanistic knowledge of how the chosen model architecture tends to solve the given task. For instance, a major part of the algorithm seems to be: the queries and keys combine to put attention on the token representing the largest integer in the input, and the output and value matrix then combine to copy this to the last token position. By using this and other properties (e.g. that the important structure of the query/key/value/output matrices is low-rank), it becomes possible to shorten the proof further.
- The main result is that approaches using more mechanistic knowledge of the
approximate algorithm implemented by the transformer require less computation for verification, but also lead to less faithful/tight bounds on the accuracy.

**Strengths:**

- this paper really engages with the computations of a transformer on a very fundamental level, and presents some ingenious ways to combine high level and heuristic approaches from the mechanistic interpretability literature with a fully verified method of deriving correctness.
- This is to my knowledge the first work to carry out such a complete analysis. One may hope that in the (not too distant) future, similar methods can be scaled up via LLM automation to apply to more interesting, non-toy reasoning tasks, and deliver ground-truth insights about the structure and weaknesses of LLM computations.
- the paper is very clear and rigorous about what it achieves, and all claims are backed up by evidence and proof.

**Weaknesses:**

- the setting is quite toy, though this is probably very hard to overcome for such an approach with human labor.

**Questions:**

- this is very interesting work! You probably have a good intuition for the most promising/low-hanging future directions in which it can be taken - can you get into specifics about this in the conclusion?
- how can this method be carried over to more realistic tasks? If I understand correctly, much of the theory in the appendices relies on a careful analysis of the structure of this particular transformer model **and** the particular (arithmetic) task in question.
- you mention: "models with similar training procedure and performance may still
have learned significantly different weights" - I believe a standard reference is "Breiman, Leo. Statistical modeling: The two cultures (with comments and a rejoinder by the author).", cf. "Rashomon"

**Limitations:**

- it is quite unclear how such results, based on careful analysis of the structure of the particular task, will carry over to more realistic tasks.
- similarly, it is unclear whether re-introducing transformer components, and especially MLP layers, won't obliterate the accuracy bounds derived here to be practically useless (e.g. below chance levels, which would be 25% here)

---

> ### Author Rebuttal · Authors · 2024-08-07
>
> Thank you for your kind review!
>
> > how can this method be carried over to more realistic tasks?
>
> It would be relatively straightforward to carry over the methodology to more realistic tasks if the following conditions were met:
>
> 1. A sufficiently detailed mechanistic understanding of the components of the model not being brute-force evaluated.
> 2. A model that is interpretable enough that the (not-understood) remainder of the model can be treated as independent error terms without blowing up the bound.
>
> If these conditions were met, then the interpretation would broadly proceed as follows:
> Every expression $x$ in the model with interpretation $y$ can be re-expressed as $x = y + (x - y)$.
> The error terms $(x - y)$ must then be given worst-case bounds.
>
> As far as we can tell, most realistic models do not meet these conditions.
> Consequently, in the project, we devote significant effort to getting reasonable bounds with minimal understanding (and it is surprising that, e.g., we can get a non-vacuous bound on the model that, modulo our lack of understanding of low-rank approximations to the identity matrix, is linear in the parameter count of the model!).
> In follow-up work, we are making progress on applying the approach to models with MLPs, to models with with multiple attention heads, to models with layer norm, and to two-layer models. We see three big obstacles. We believe all the obstacles are important for the field of mech interp to address (with the possible exception of the third one):
>
> 1. Unstructured approximation error terms (“noise”) accumulate quickly, necessitating either models with significantly less noise (more interpretable models) or more sophisticated error term analysis.
> In follow-up work, we have preliminary results suggesting that fine-tuning models can make them sufficiently more interpretable as to bypass this problem.
> 2. Path decomposition is exponential in the number of layers, necessitating adequate mechanistic understanding of how to simultaneously argue that multiple paths contribute negligibly.
> Existing approaches to solving instances of this problem in finding mechanistic interpretations -- such as presuming independence of irrelevant paths and evaluating them all simultaneously, or using gradient-based search methods that leave the path decomposition implicit -- unfortunately do not yield valid proof techniques.
> This problem is especially pressing for us, as it interacts with the central problem of compounding error approximations (1).
> However, we are currently investigating the possibility of matching prefixes of paths to allow mechanistic understanding to be used to simultaneously argue that multiple paths contribute negligibly.
> 3. Worst-case error bounds are both significantly looser by default and significantly more expensive to compute than average-case error bounds.
> We would be excited to see a formalization and analysis of compact guarantees of average-case performance!  (Unfortunately, we are not aware of any adequate formalism of average-case performance that is independent of training method.)
>
> > If I understand correctly, much of the theory in the appendices relies on a careful analysis of the structure of this particular transformer model **and** the particular (arithmetic) task in question.
>
> That is correct.
> However, we would like to highlight that it should be straightforward (if labor-intensive) to generalize our work to other toy models where the model computes the answer by paying attention to a single token or position, i.e. general element retrieval, example tasks include: first element, last element, binding adjective to key phrase, etc.
> This is because our theorems in the appendices prove in full generality that any computation performed by a single attention head, when only one of the position and the token value matters, is extremized by pure sequences.
> So the worst-case behavior of the computation at a single token can always be found by setting all other tokens to be identical, regardless of what the task is.
> Furthermore, generalizing our convexity arguments to multiple attention heads *whose error terms can be treated independently without degrading the bound too much* is also straightforward.
>
> Generalizing the theory to handle coupled attention heads or more complex uses of attention will require additional theoretical work.
>
> We address the high-level version of this concern in the top-level rebuttal where we talk about the general applicability of our approach to projects on larger input distributions, different tasks, more complex architectures, and larger models.
>
>
> > similarly, it is unclear whether re-introducing transformer components, and especially MLP layers, won't obliterate the accuracy bounds derived here to be practically useless (e.g. below chance levels, which would be 25% here)
>
> In follow-up work, we conduct analysis of a modular addition model with an MLP where we achieve a bound of roughly 80%--90% on a sub-brute-force proof.
> However, it is indeed the case that integrating all the mechanistic understanding we have of that obliterates the accuracy bound.
> Ultimately, we hope that fine-tuning will allow bypassing this issue.
>
> Separately, we want to note that 25% would be *average case* chance levels for *argmax* (predicting the position of the maximum).
> Average-case chance levels on max are actually $1/d_{\mathrm{vocab}} = 1/64 = 1.56\%$ for this task, since we are predicting tokens not positions.
> Furthermore, it's not entirely clear that the baseline for proofs should be average-case bounds rather than worst-case bounds.
> Indeed we typically see bound performance drop off to floating point error, not average-case chance, when error terms accumulate too quickly.
>
> > you mention: [...] I believe a standard reference is [...]
>
> Thank you!
>
> Please don't hesitate to ask any further questions!
> We'd love the chance to highlight strengths of our submission that might encourage you to further increase your score!

---

> > ### Comment · Reviewer_KZFC · 2024-08-13
> > **Thank you for the detailed clarifications**
> >
> > I thank the authors for their very thorough followup to my questions. Also, thanks for pointing out my mistake about the chance-level performance on the max task!
> >
> > I am looking forward to future work in this direction, and I remain strongly in favor of accepting this paper to the conference.

---

### Author Rebuttal · Authors · 2024-08-07

Thank you to the reviewers for their comments and feedback!
We’re happy that reviewers agree that our approach to verification is novel and that our approach to mech interp is rigorous.

We address the five common questions that reviewers raised.

**(1) Motivation for picking a toy setting (vjrV, KZFC, and ZhNY)**

Formal reasoning is computationally expensive; very few large software projects have ever been verified [1][2], none of them comparable to large transformer models [3][4].
Separately, there is a high fixed cost to taking on any verification project, regardless of computational efficiency of the verification itself.
Thus, we picked the simplest setting to study the question of interest:
Is it even possible to reason more efficiently than by brute force about model behavior?
We did not adequately explain this motivation in the submission, and will add a section.

**(2) Applicability of results to other settings (exsQ, vjrV, KZFC, and ZhNY)**

1. **Larger input spaces:**
We applied our proof strategies to models trained for Max-of-10 and Max-of-20, and have attached our results.
While running the brute force proof on Max-of-20 would require $64^{20} \cdot 20 \cdot 64 \cdot 32 \approx 10^{40}$ FLOPs, which is about $10^{17} \times$ the cost of training GPT-3, our cubic proof achieves bounds of 94% (Max-of-5), 93% (Max-of-10) and 83% (Max-of-20) in under two minutes, demonstrating that proof strategies can be reused on larger input spaces (and indeed, scales better than the brute force or sampling approach do).
2. **Different tasks:**
In this work, we worked on highly optimized relaxations to make our bounds as tight as possible when incorporating as little understanding as possible.
This is not necessary for deriving proofs.
Our general formalization of mech interp is replicable: (1) theorem statements are exact expressions for the difference between the actual behavior of the model and the purported behavior, and (2) proofs are computations that bound the expression.
Furthermore, our convexity theorems and proofs are applicable much more generally generally to element retrieval tasks.
3. **More complicated architectures:** We worked on a simple model studied in A Mathematical Framework for Transformer Circuits.
In follow-up work, we extend this approach to proving bounds on 1L transformers with ReLU MLP trained on modular addition.
4. **Larger models:** We agree that it’s an open question whether or not the mech interp approach to proofs can scale to larger models.
However, a large part of this question lies in the feasibility of deriving a high degree of faithful mechanistic understanding from large models (i.e. whether mech interp itself will scale).
This is widely recognized in the field of mech interp, and scaling interp approaches while getting both a high degree of mechanistic understanding and assurances that said understanding is faithful to the model is an active area of research in mech interp in general.

**(3) Why use mech interp at all, especially given that it “results” in worse bounds? (exsQ and ZhNY)**

First, we agree that ideally, we could maintain the tightness of the bound even as we decrease the length of the proof.
However, there’s some fundamental reasons to expect there to be a trade off: for example, from the compression perspective, any proof strategy that does not run the true model on all inputs needs to use a compressed approximation to the model, the input distribution, or both.
So unless the model weights are losslessly compressible, shorter proofs using lossy approximations will likely cause weaker bounds.
Consequently, the results of proofs derived via mech interp should be compared to other proofs of comparable length.
Indeed, we find better mech interp can give us tighter bounds at same proof length (figure 3), which provides evidence that better mech interp can be used to improve proofs.

Conversely, we can think of the quality of proofs (the combination of tightness of bound, and length of proof) as a metric for how good our mechanistic understanding is.
From this perspective, the fact that mech interp derived bounds are worse suggests gaps in our mechanistic understanding.
As mech interp matures as a field and we develop tools that enable more faithful and complete understanding of model behavior, we expect that the quality of bounds we derive from mechanistic understanding will improve.

**(4) Less faithful interpretations produce better bounds (exsQ and vjrV)**

We have corrected the plot attachment, which was a genuine mistake on our part.
The plot is more visually clear in the point it makes.

**(5) Validity of the noise hypothesis (exsQ and vjrV)**

We want to clarify that we use “noise” to refer to error terms in our approximation, which is confusing terminology that we will change.
In our work, we use approximations to model components in order to shorten proofs.
While we were able to approximate individual matrices with small error terms, when we compose the approximations, the estimate for the error term on the matrix composition is much larger than the empirical error term.
This is because lower-bounding (aka worst-case bounds) requires that we pessimize over the error terms.
Thus, we are not referring to noise in models (which we might ablate to improve our bounds or introduce to check our hypothesis).

Additionally, reviewers point out issues in our presentation.
We have incorporated feedback from this review process, and have made significant progress since the submission of the manuscript, including standardizing language and adding definitions, simplifying notation, and improving copyediting.
We address remaining issues in individual rebuttals.

[1]: Leroy. “A Formally Verified Compiler Back-end”

[2]: Klein et al. “seL4: Formal Verification of an OS Kernel”

[3]: Clarke. “State Explosion Problem in Model Checking”

[4]: Gross. “Performance Engineering of Proof-Based Software Systems at Scale”

---

### Decision · Program_Chairs · 2024-09-25

**Decision:**

Accept (poster)

**Comment:**

The paper proposes strategies for verifying the performance of a simplified transformer model (one layer, one head, attention only) on a toy task (max of $k$ integers in a bounded range). The authors use clever tricks to reduce the complexity from brute force $d_{vocab}^{context}$ by making use of the precise computations of the transformer, and further use mechanistic interpretability techniques on the learned models to get worse lower bounds at faster computation.

The main weakness of the paper is that the techniques are tailored to the simplified problem at hand and it seems to be very challenging to extend them efficiently to more complex models where we lack any mechanistic understanding (or can't really estimate the faithfulness of these). This seems like a severe limitation to me, though I agree with the other reviewers that the study is interesting, pretty rigorous (experimentally and theoretically), and could spark interesting future work. So I recommend accepting the paper. I encourage the authors to add a section about the applicability to other settings that they discussed in the rebuttal.